# Shuffling the Data, Stretching the Step-Size: Sharper Bias in Constant Step-Size SGD

**Konstantinos Emmanouilidis**[*], **Emmanouil-Vasileios Vlatakis-Gkaragkounis**[#], **René Vidal**[*]
Department of Computer Science
[*]University of Pennsylvania, [#]University of Wisconsin-Madison

## Abstract

From adversarial robustness to multi-agent learning, many machine learning tasks can be cast as finite-sum min–max optimization or, more generally, as variational inequality problems (VIPs). Owing to their simplicity and scalability, stochastic gradient methods with constant step size are widely used, despite the fact that they converge only up to a constant term. Among the many heuristics adopted in practice, two classical techniques have recently attracted attention to mitigate this issue: *Random Reshuffling* of data and *Richardson–Romberg extrapolation* across iterates. Random Reshuffling sharpens the mean-squared error (MSE) of the estimated solution, while Richardson-Romberg extrapolation acts orthogonally, providing a second order reduction in its bias. In this work, we show that their composition is strictly better than both, not only maintaining the enhanced MSE guarantees but also yielding an even greater cubic refinement in the bias. To the best of our knowledge, our work provides the first theoretical guarantees for such a synergy in structured non-monotone VIPs. Our analysis proceeds in two steps: (i) we smooth the discrete noise induced by reshuffling and leverage tools from continuous-state Markov chain theory to establish a novel law of large numbers and a central limit theorem for its iterates; and (ii) we employ spectral tensor techniques to prove that extrapolation debiases and sharpens the asymptotic behavior even under the biased gradient oracle induced by reshuffling. Finally, extensive experiments validate our theory, consistently demonstrating substantial speedups in practice.

## 1 Introduction

Variational inequality problems (VIPs) (Stampacchia, 1964) are a unifying framework that extends beyond classical loss minimization to encompass min–max optimization, complementarity problems (Dantzig & Cottle, 1968; Facchinei & Pang, 2003), equilibrium computation in games, and general fixed-point formulations (Bauschke & Combettes, 2017). In recent years, VIPs have gained significant traction in ML and data science, especially due to their broad potential applicability in domains where minimizing a single empirical loss is insufficient. Notable examples include generative adversarial networks (Goodfellow et al., 2014; Arjovsky et al., 2017), multi-agent and robust reinforcement learning (Namkoong & Duchi, 2016; Wang et al., 2021; Giannou et al., 2022), and auction theory (Syrgkanis et al., 2015).

In practice, many of these tasks reduce to finite-sum formulations, where the objective depends on a large collection of data samples or agents. In such settings, *stochastic gradient methods* have become the workhorse of large-scale learning (Bottou et al., 2018). By exploiting the finite-sum structure, stochastic gradient descent (SGD) and its variants replace expensive full-gradient computations with inexpensive updates on a few components, enabling scalability to massive datasets.

While the theoretical underpinnings of SGD date back to the seminal work of Robbins & Monro (1951) and Kiefer & Wolfowitz (1952), the initial results involved an asymptotic analysis under vanishing step size. Consequent works (Ljung, 1978; Benaïm, 2006; Rakhlin et al., 2011; Raginsky et al., 2017; Azizian et al., 2024; Malick & Mertikopoulos, 2024) have extended the theoretical guarantees of SGD under different design choices, as much of its practical success can be traced to a handful of seemingly "low-level" heuristics (Bottou, 2012b): step-size schedules (constant vs. decaying), data ordering (with vs. without resampling), and iterate selection (average vs. last iterate).

To facilitate analysis, the community has typically adopted a *ceteris paribus* perspective—isolating one design choice at a time while holding the rest fixed—an approach that clarifies individual effects but obscures their interaction.

A particularly important case is the use of a *constant step size $\gamma$*, which is popular in practice since it simplifies tuning, quickly erases the dependence on initialization, and yields fast early progress (Yu et al., 2021). However, its fundamental drawback is that convergence halts at a non-vanishing error. Even in strongly convex problems with unique solution $x^*$, the last iterate of SGD typically satisfies:

$$\text{MSE}(\text{SGD}) = \limsup_{k \to \infty} \mathbb{E}[\|x_k - x^*\|^2] = \mathcal{O}(\gamma) \text{ and bias}(\text{SGD}) = \limsup_{k \to \infty} \|\mathbb{E}[x_k] - x^*\| = \mathcal{O}(\gamma).$$

That is, the iterates stabilize at a distance from the optimum on the order of the step size.

▶ To mitigate this limitation, practitioners often turn to debiasing heuristics. A prominent example is *random reshuffling* ($\text{RR}_1$), or *without-replacement* sampling, where each data point is visited exactly once per epoch. Unlike classical *with-replacement* SGD, which may resample or skip points, $\text{RR}_1$ enforces a random full pass that closely mirrors large-scale training in practice (Bottou et al., 2018). Despite the dependence it induces across samples, recent work has established faster convergence guarantees for $\text{RR}_1$ in both minimization (Ahn et al., 2020; Gürbüzbalaban et al., 2021; Cai et al., 2023) and VIPs (Mishchenko et al., 2020b; Emmanouilidis et al., 2024), along with sharper *MSE* bounds from $\mathcal{O}(\gamma)$ to $\mathcal{O}(\gamma^2)$. However, the question of whether the bias term improves has remained open. Indeed, recall that for any estimator $\hat{x}$, the mean squared error decomposes as

$$\text{MSE}(\hat{x}) = \mathbb{E}[\|\hat{x} - x^*\|^2] = \|\mathbb{E}[\hat{x}] - x^*\|^2 + \text{Var}(\hat{x}),$$

so that $\text{bias}(\hat{x}) \leq \sqrt{\text{MSE}(\hat{x})}$. Under this trivial bound, SGD–$\text{RR}_1$ guarantees improved MSE compared to vanilla SGD, but does not necessarily yield smaller bias.

▶ Orthogonal to reshuffling, another classical idea from numerical analysis has recently re-emerged in stochastic optimization: *Richardson–Romberg* ($\text{RR}_2$) *extrapolation*. Its principle is simple yet powerful: *Run the algorithm of your choice with two different step sizes and combine their outputs so that the leading bias term cancels.* Concretely, whenever the bias admits an expansion of the form $\text{bias}(\gamma) = \Delta\gamma + \mathcal{O}(\gamma^\kappa)$ with $\kappa > 1$, running the stochastic approximation at two step sizes gives:

$$x_\infty^\gamma - x^* = \Delta\gamma + \mathcal{O}(\gamma^\kappa) \text{ and } x_\infty^{2\gamma} - x^* = 2\Delta\gamma + \mathcal{O}(\gamma^\kappa).$$

Extrapolating these iterates then yields:

$$x_{\text{extr}} - x^* = 2x_\infty^\gamma - x_\infty^{2\gamma} - x^* = 2\Delta\gamma - 2\Delta\gamma + \mathcal{O}(\gamma^\kappa) = \mathcal{O}(\gamma^\kappa).$$

Originally introduced for accelerating discretization schemes in stochastic differential equations (Hildebrand, 1987; Talay & Tubaro, 1990; Bally & Talay, 1996), $\text{RR}_2$ has since been applied to optimization, improving constant-step methods from SGD (Durmus et al., 2016; Dieuleveut et al., 2020; Mangold et al., 2024; Sheshukova et al., 2024) to Q-learning and two-timescale stochastic approximation methods (Huo et al., 2023; Kwon et al., 2024; Zhang & Xie, 2024; Allmeier & Gast, 2024). However, despite its conceptual simplicity and empirical success, the theoretical foundations of $\text{RR}_2$ for stochastic VIPs remain nascent (Vlatakis-Gkaragkounis et al., 2024).

In summary, the known *bias* rates of the heuristics remain limited when they are applied in isolation: For unconstrained strongly monotone VIPs, $\text{RR}_1$ is known to sharpen *MSE* bounds (from $\mathcal{O}(\gamma)$ to $\mathcal{O}(\gamma^2)$) but does not, in general, guarantee an improved bias order; on the other hand $\text{RR}_2$ alone attains $\mathcal{O}(\gamma^{3/2})$ bias (Vlatakis-Gkaragkounis et al., 2024)[1]. However, it remains unclear whether one can combine effectively the aforementioned heuristics in a unified method that outperforms both, ideally reaching $\mathcal{O}(\gamma^3)$ bias. This raises the natural question:

> *What new phenomena arise when these heuristics*
> *— constant step sizes, random reshuffling, and Richardson extrapolation—*     (★)
> *interact simultaneously?*

Addressing this question is delicate, because reshuffling introduces a biased stochastic oracle whose discrete, permutation-driven noise structure is not covered by existing analyses of extrapolation,

---

[1] The $\mathcal{O}(\gamma^2)$ rate in Vlatakis-Gkaragkounis et al. (2024, Sec. 5, Thm. 6) is obtained via a reduction to Dieuleveut et al. (2020, Sec. 3, Thm. 4), which requires additional noise assumptions not met in our setting.

which predominantly assume unbiased or continuously distributed perturbations (Dieuleveut et al., 2020; Sheshukova et al., 2024; Vlatakis-Gkaragkounis et al., 2024).

**Our contributions.** In this work, we unify the $\text{RR}_1$ and $\text{RR}_2$ heuristics into a principled algorithmic framework, showing that their composition yields a level of bias reduction unattainable by either heuristic alone. Our main result (Theorem 4.6) shows that for quasi-strongly monotone smooth VIPs, our combined method (SGD-$\text{RR}_2\oplus\text{RR}_1$, Algorithm 1) cancels all lower-order terms in the bias expansion, yielding an asymptotic bias of order $\mathcal{O}(\gamma^3)$.

Our contributions are as follows:

⋆ *Convergence of SGD-$RR_1$ under perturbations.* We first analyze the $\text{RR}_1$ component, proving exponential convergence with a bias that is linear (quadratic) under weak (quasi-strong) monotonicity, robust even to preprocessing perturbations.

⋆ *Epoch-level Markov chain viewpoint of SGD-$RR_1$.* We prove that the dynamics of $\text{RR}_1$ admit an invariant stationary distribution and establish a law-of-large-numbers (LLN) and a central-limit-theorem (CLT) for the per-epoch iterates of SGD-$\text{RR}_1$.

⋆ *Cubic Bias Refinement through the synergy of $RR_2\oplus RR_1$.* We extrapolate the iterates produced from SGD-$\text{RR}_1$and analyze the combination of the heuristics. Through a spectral analysis of the reshuffling-induced biased oracle, we prove that the first $\mathcal{O}(\gamma^3)$ bias guarantee for our algorithm in quasi strong monotone VIPs.

## 2 PROBLEM SETUP AND ASSUMPTIONS

**Variational inequalities.** Let's recall first the basic framework of finite-sum variational inequalities (VIs), which will underlie our analysis. Let $X \subseteq \mathbb{R}^d$ be a nonempty closed convex set and $F : \mathbb{R}^d \to \mathbb{R}^d$ a single-valued operator. The variational inequality problem $\text{VI}(X, F)$ asks for a point $x^\star \in X$ such that

$$\langle F(x^*), x - x^* \rangle \geq 0, \quad \forall x \in X \tag{VIP}$$

In our setting, we focus on the unconstrained finite-sum case, i.e. $X = \mathbb{R}^d$ and $F(x) = \frac{1}{n}\sum_{i=0}^{n-1} F_i(x)$, respectively, where each $F_i : \mathbb{R}^d \to \mathbb{R}^d$ typically represents the gradient of a loss function for a data point in some dataset $\mathcal{D}$. To build intuition, we illustrate the framework through a few canonical examples below:

**Example 2.1: Solving Non-linear equations.** A solution $x^*$ to the (VIP) corresponds to a root of the equation $F(x) = \mathbf{0}$. Therefore, we can cast any non-linear equation as a specific instantiation of the Variational Inequality framework. Well-known examples include the Navier-Stokes equations in computational dynamics (Hao, 2021).

**Example 2.2: Empirical Risk Minimization.** For any $\mathcal{C}^1-$smooth loss function $\ell : \mathbb{R}^d \to \mathbb{R}$, a solution $x^*$ to the (VIP) with $F(x) = \nabla\ell(x)$ is a critical point (KKT solution) to the associated empirical risk minimization problem, consisting the cornerstone of machine learning objectives.

**Example 2.3: Nash Equilibria & Saddle-point Problems.** Consider $N$ players, each having an action set in $\mathbb{R}^d$ and a convex cost function $c_i : \mathbb{R}^d \to \mathbb{R}$. A Nash Equilibrium (NE) is a joint-action profile $x^* = (x_i^*)_{i=1}^N$ that satisfies

$$c_i(x^*) \leq c_i(x_i; x_{-i}^*), \quad \forall i, x_i \in \mathbb{R}^d. \tag{NE}$$

For convex cost functions $c_i : \mathbb{R}^d \to \mathbb{R}$, a (NE) coincides with the solution of a (VIP) with operator $F(x) = (\nabla_{x_i} c_i(x))_{i=1}^N$. In the particular case of two players and a (quasi) convex-concave objective $\mathcal{L} : \mathbb{R}^d \times \mathbb{R}^d \to \mathbb{R}$, the solution $x^* = (x_1^*, x_2^*)$ to the (VIP) with $F(x) = (\nabla\mathcal{L}(x), -\nabla\mathcal{L}(x))$ is a saddle point of $\mathcal{L}$ satisfying

$$\mathcal{L}(x_1^*, x_2) \leq \mathcal{L}(x_1^*, x_2^*) \leq \mathcal{L}(x_1, x_2^*), \quad \forall x_1, x_2 \in \mathbb{R}^d.$$

Saddle-point problems and applications of (NE) are ubiquitous, from training Generative Adversarial Networks (GANs) to multi-agent reinforcement learning and auction/bandit problems (Daskalakis et al., 2017; Zhang et al., 2021; Pfau & Vinyals, 2016).

**Blanket assumptions.** We now state the standing assumptions for our analysis, beginning with the existence of a solution $x^\star$ to (VIP).

**Assumption 2.1.** The solution set $\mathcal{X}^*$ of (VIP) is nonempty and there exists $x^* \in \mathcal{X}^*, R \in \mathbb{R}$ such that $\|x^*\|_2 \leq R$.

The next assumption introduces the class of operators $F$ of the associated (VIP) for which our stochastic gradient algorithms will be analyzed.

**Assumption 2.2** ($\lambda$-weak $\mu$-quasi strong monotonicity). The operator $F$ is $\lambda$-weak $\mu$-quasi strongly monotone, i.e. there exist $\lambda \geq 0, \mu > 0$ such that for some $x^* \in \mathcal{X}^*$ it holds that

$$\langle F(x), x - x^* \rangle \geq \mu\|x - x^*\|^2 - \lambda, \quad \forall x \in \mathbb{R}^d. \tag{1}$$

Assumption 2.2 for $\lambda = 0$ coincides with the well-known notions of quasi-strong monotonicity (Loizou et al., 2020), strong stability condition (Mertikopoulos & Zhou, 2019), and strongly coherent VIPs (Song et al., 2020) in the optimization literature. It can be seen as a relaxation of the classical notion of strong monotonicity/convexity, which requires $\langle F(x) - F(x'), x - x' \rangle \geq \mu\|x - x'\|^2, \forall x, x' \in \mathbb{R}^d$. For $\lambda > 0$, Assumption 2.2 represents a further relaxation, motivated by dissipative dynamical systems and weakly convex optimization (Raginsky et al., 2017; Erdogdu et al., 2018), and it encompasses non-monotone games as well as a variety of problems in statistical learning theory (Tan & Vershynin, 2023).

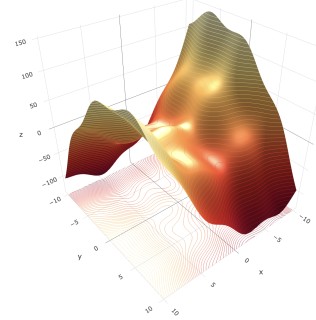

Figure 1: A simple example of a function satisfying Assumption 2.2 is $f(x, y) = (x^2 + 7\sin(x)) + xy - (y^2 - 7\cos(y))$, where the assumption holds with $(\mu, \lambda) = (1, 25)$.

A common assumption in the literature of smooth optimization that we will utilize is that the operators in the finite-sum structure of the (VIP) are Lipschitz continuous.

**Assumption 2.3** (Lipschitz continuity). Each $F_i$ is $L_i$-Lipschitz:

$$\|F_i(x_1) - F_i(x_2)\| \leq L_i\|x_1 - x_2\|, \qquad \forall x_1, x_2 \in \mathbb{R}^d, \ i \in [n],$$

with $L_{\max} = \max_{i \in [n]} L_i$.

Unlike standard analyses assuming unbiased oracles with bounded variance, e.g., (Loizou et al., 2021; Hsieh et al., 2019; Lin et al., 2020; Mishchenko et al., 2020b), random reshuffling induces bias via inter-step dependence. Such conditions may fail even for simple quadratics. Instead, we work directly with Lipschitz continuity and impose only a mild moment bound:

**Assumption 2.4** (Bounded moments at the solution). At some $x^* \in \mathcal{X}^*$, the oracle values have finite second and fourth moments:

$$\sigma_*^2 := \frac{1}{n}\sum_{i=1}^n \|F_i(x^*)\|^2 < \infty, \qquad \sigma_*^4 := \frac{1}{n}\sum_{i=1}^n \|F_i(x^*)\|^4 < \infty.$$

Assumption 2.4 is mild: it does not require global boundedness of gradients, but only that the oracle values $F_i(x^*)$ admit finite second and fourth moments at the solution. Building on this, we extend the variance bound of Emmanouilidis et al. (2024, Prop. A.2, p. 16) to higher-order moments.

**Proposition 2.5.** Let Assumptions 2.1–2.3 hold. Then, for any $x \in \mathbb{R}^d$ it holds that

$$\text{(i)} \quad \frac{1}{n}\sum_{i=1}^n \|F_i(x) - F(x)\|^2 \leq \frac{2}{n}\sum_{i=1}^n L_i^2\|x - x^*\|^2 + 2\sigma_*^2,$$

$$\text{(ii)} \quad \frac{1}{n}\sum_{i=1}^n \|F_i(x) - F(x)\|^4 \leq \frac{128}{n}\sum_{i=1}^n L_i^4\|x - x^*\|^4 + 128\sigma_*^4.$$

Having introduced the necessary assumptions, we, next, provide the proposed algorithm that will achieve the higher order bias refinement in our results.

## 3 THE ALGORITHMIC FRAMEWORK

While there are many conceivable ways to interleave $RR_2$ and $RR_1$, both intra- and inter-epoch, we adopt the most natural and practically motivated design. In modern pipelines, $RR_1$ is the workhorse at the low-level training stage, while $RR_2$ is often employed as a black-box refinement at a higher level, allowing parallelization and modular integration.

Accordingly, we study stochastic gradient algorithms that sample via random reshuffling to generate stochastic oracles of gradients/operators. At the start of each epoch $k > 0$, a random permutation $\omega_k$ of $[n]$ is drawn, prescribing the order in which data points are processed. The algorithm then performs the classical SGD update:

$$x_k^{i+1} = x_k^i - \gamma \operatorname{PreProcess}\big[\operatorname{StochOracle}(x_k^i; \omega_k^i)\big], \qquad \text{(SGD-}RR_2 \oplus RR_1\text{(inner-loop))}$$

where $\operatorname{StochOracle}(x_k^i; \omega_k^i)$ denotes either the stochastic gradient (in minimization problems) or the operator value $F_{\omega_k^i}(x_k^i)$ (in the general VI case), indexed by the $\omega_k^i$ data point and $\operatorname{PreProcess}[\cdot]$ is a preprocessing routine implementing calibrated Gaussian smoothing to the input. Then, the final iterate of each epoch becomes the starting point of the next, and the procedure repeats.

---

**Algorithm 1** SGD$-RR_2 \oplus RR_1$

---

**Require:** Initial point $x_0 \in \mathbb{R}^d$; step size $\gamma > 0$; epochs $I$; dataset size $n$;
       STOCHORACLE$(x; i)$ returns $F_i(x)$ (minimization) or operator value (VI);
       PREPROCESS$(g; i)$ adds calibrated Gaussian smoothing on $g$ (e.g., $\ U_k \sim \mathcal{N}(0, \gamma^2 n \sigma_*^2 I)$).
1: **for** $k = 0, 1, \dots, I - 1$ **do**                                      $\triangleright$ epoch $k$
2:      **for** $\eta = \gamma, 2\gamma$ **do**                        $\triangleright$ Parallel iterations with two step-sizes
3:          Draw a random permutation $\omega_k$ of $[n]$
4:          **for** $i = 0, 1, \dots, n - 1$ **do**                   $\triangleright$ inner loop (reshuffled pass)
5:              $x_{k,[\eta]}^{i+1} \leftarrow x_{k,[\eta]}^i - \eta \operatorname{PREPROCESS}(\operatorname{STOCHORACLE}(x_{k,[\eta]}^i, \omega_k[i]))$
6:          **end for**
7:          $x_{k+1,[\eta]}^0 \leftarrow x_{k,[\eta]}^n$                        $\triangleright$ baseline next-start (used for analysis)
8:      **end for**
9:      $\hat{x}_{k+1} \leftarrow 2\, x_{k,[\gamma]}^n - x_{k,[2\gamma]}^n$             $\triangleright$ outer loop ( extrapolation at epoch end)
         OR
10:     $\hat{x}_{k+1} \leftarrow (2 \sum_{m \in [k]} x_{k,[\gamma]}^n - x_{m,[2\gamma]}^n)/k$      $\triangleright$ Alternative: ( extrapolation at epoch's averages)
11: **end for**
12: **return** $\hat{x}_I$                                 $\triangleright$ (optionally average $\{\hat{x}_k\}$ across epochs)

---

*On the necessity of smoothing.* A key challenge with reshuffling is that, after one epoch, the cumulative gradient estimator is biased, unlike sampling with replacement, which is unbiased and analytically simpler. The induced noise is also discrete, tied to permutations. To handle this, we introduce a calibrated Gaussian perturbation that smooths the discrete reshuffling noise into a well-behaved proxy while preserving variance, moments, and bias order. In practice, the perturbation has negligible effect across datasets; clarifying its precise dependence on dataset size is an interesting direction for future work. For completeness, the supplement also includes a brief sketch showing how our results extend even without this step.

Finally, at the end of each epoch we apply $RR_2$, yielding the extrapolated update:

$$\hat{x}_{I+1}^N = 2\, x_{I,[\gamma]}^N - x_{I,[2\gamma]}^N. \qquad \text{(SGD-}RR_2 \oplus RR_1\text{(outer-loop))}$$

In Section 4, we prove that this combination achieves a provable $\mathcal{O}(\gamma^3)$ bias— to the best of our knowledge, the first such result.

## 4 THEORETICAL GUARANTEES

We present the theoretical guarantees for the synergy of both heuristics in our algorithm, establishing a cubic refinement in the bias attained. We start by analyzing the $RR_1$ component (inner loop of

| Aspect | Emmanouilidis et al. (2024) | Our work |
|---|---|---|
| Baseline Algorithm | **SEG** | **SGDA** |
| Model Assumptions (Smoothness) | $F_i$–$L_i$ Lipschitz | $F_i$–$L_i$ Lipschitz |
| Model Assumptions (Drift) | $F$ $\mu$–strongly monotone | $F$ *quasi*–strongly monotone |
| Main heuristic | **RR$_1$ only** | **RR$_1$ $\oplus$ RR$_2$** |
| Asymptotic Bias order | $\mathcal{O}(\gamma + \gamma^3)$ | $\mathcal{O}(\gamma^3)$ |
| Asymptotic MSE order | $\mathcal{O}(\gamma^2)$ | $\mathcal{O}(\gamma^2)$ |
| Condition number | **Worse than vanilla-SEG** | **Same as vanilla-SGDA** |
| Mechanism | EG-structure + RR$_1$ | Bias cancellation (RR$_1$ $\oplus$ RR$_2$) |

Table 1: Summary of key differences between Emmanouilidis et al. (2024) and our results.

Algorithm 1), which serves as the basis for performing RR$_2$ (outer loop of Algorithm 1) on the produced iterates and describe the interplay of both heuristics in the bias of Algorithm 1.

## 4.1 CONVERGENCE GUARANTEES FOR RR$_1$

Our first result concerns the convergence of the RR$_1$ component for $\lambda$-weak $\mu$-quasi strongly monotone VIPs. More specifically, in order to effectively utilize Markov Chain tools we study the RR$_1$ variant under preprocessing perturbations[2] (inner loop of Algorithm 1), attributing for simplicity of exposition of our results the name Perturbed SGD–RR$_1$.

**Theorem 4.1.** Let Assumptions 2.1-2.3 hold. If $\gamma \leq \gamma_{max}$, then the iterates of Perturbed SGD–RR$_1$ satisfy

$$\mathbb{E}\left[\|x_{k+1}^0 - x_*\|^2\right] \leq \left(1 - \frac{\gamma n \mu}{2}\right)^{k+1} \|x_0^0 - x^*\|^2 + \frac{8n\gamma^2 L_{max}^2}{\mu^2}\sigma_*^2 + \frac{8\lambda}{\mu}$$

where $\sigma_*^2 = \frac{1}{n}\sum_{i=0}^{n-1}\|F_i(x^*)\|^2$ and $\gamma_{max} = \min\left\{\frac{1}{3nL_{max}}, \frac{\sqrt{1+6\mu^2 L_{max}^2}-1}{12nL_{max}^2}\right\}$.

**Remark 1.** *Theorem 4.1 establishes exponential convergence up to a bias of $\mathcal{O}(\gamma^2\sigma_*^2 + \frac{\lambda}{\mu})$, where the $\frac{8\lambda}{\mu}$ term is inherent (Yu et al., 2021). For fair comparison we focus on the quasi-strongly monotone case ($\lambda = 0$), which already generalizes strong convexity. Our rate recovers known results for strongly monotone operators (Das et al., 2022; Emmanouilidis et al., 2024) and extends them to weak monotonicity. In this regime, reshuffling attains a much smaller bias than the $\mathcal{O}(\gamma\sigma_*^2)$ of with-replacement SGD (Loizou et al., 2020; Gower et al., 2019), converging to a tighter neighborhood. This sharper bias also yields faster accuracy rates: with $\gamma = 1/(nK)$, with-replacement SGD reaches $\mathcal{O}(1/(nK))$ accuracy (Das et al., 2022; Mishchenko et al., 2020a), while reshuffling accelerates to $\mathcal{O}(1/(nK^2))$, a further support for its empirical success.*

In the sequel, we view the algorithmic trajectory through the prism of Markov chain theory. This perspective enables a finer dissection of the reshuffling bias and, mutatis mutandis, equips us with the machinery to construct consistent estimators for performance statistics. The Markovian framework arises naturally, as the method progresses from $x_k$ to $x_{k+1}$ in a state-dependent fashion. The connection between stochastic approximation and Markov processes—traced back to early works such as Robbins & Monro (1951); Pflug (1986)—has fueled a rich literature for algorithms with unbiased oracles. Random reshuffling, however, generates systematically biased oracles, necessitating a genuine departure from this canonical line of analysis.

For readers accustomed only to classical finite-state Markov chains, the transition mechanism is usually represented by a directed graph with fixed transition probabilities. In our setting, the analogue is the transition kernel $P(x, A) = \Pr\left[x_{\text{next}} \in A \mid x_{\text{now}} = x\right], A \in \mathcal{B}(\mathbb{R}^d)$, where $\mathcal{B}(\mathbb{R}^d)$ denotes the Borel sets of $\mathbb{R}^d$. As in the finite-state case, it is highly desirable that this kernel remain invariant over time—this is the property of time-homogeneity.[3]

---

[2]Empirically, for sufficiently large datasets the effect of discrete noise in smooth problems is negligible, making the preprocessing step unnecessary. A detailed study of this effect lies beyond the scope of this paper, whose focus is instead the first systematic treatment of the interaction between Random reshuffling and Richardson extrapolation.

[3]If time-homogeneity fails, a process can be still Markovian in the sense that the future depends only on the present, but its statistical regularity vary with time, complicating both analysis and long-run guarantees.

At the step level, reshuffling destroys homogeneity: the transition kernel varies with the permutation index, making the process non-stationary. Fortunately, this irregularity vanishes at the epoch scale:

> *"After one reshuffled pass, the law of the next iterate depends only on the epoch's starting point and the drawn permutation, but not its position within the permutation."*

Thus, the sequence of epoch-level iterates $(x_k^{[0]})_{k\geq0}$ forms a bona fide time-homogeneous Markov chain, forming the basis for the asymptotic analysis of the $\text{RR}_2$ extrapolation component [4]

**Lemma 4.2** (Epoch-level homogeneity and kernel). Fix $\gamma > 0$ and $n \in \mathbb{N}$. Then the *Perturbed SGD-RR$_1$* can be described at each epoch $k$ as: *Draw $\omega_k$ uniformly from $\mathfrak{S}_n$ and set*

$$x_{k+1} = H(x_k, \omega_k) + U_k, \quad U_k \sim \mathcal{N}(0, \Sigma),$$

where $H(x, \omega)$ denotes the endpoint of one reshuffled pass started at $x$ with permutation $\omega$ (i.e., the map induced by $n$ inner updates with step size $\gamma$). Then $(x_k)_{k\geq0}$ is a time-homogeneous Markov chain on $\mathbb{R}^d$ with transition kernel

$$P(x, A) = \frac{1}{n!} \sum_{\omega \in \mathfrak{S}_n} \int_A \phi(y; H(x, \omega), \Sigma)\, dy, \qquad A \in \mathcal{B}(\mathbb{R}^d),$$

where $\phi(\cdot; m, \Sigma)$ is the $d$-variate Gaussian density with mean $m$ and covariance $\Sigma$.

By verifying irreducibility, aperiodicity, and *positive Harris recurrence* (Meyn & Tweedie, 2012), we establish a unique invariant distribution $\pi_\gamma$, geometric convergence in total variation to it, and concentration of scalar observables (admissible test functions) around $x^*$.

**Theorem 4.3.** Under Assumptions 2.1–2.3, run Perturbed SGD-RR$_1$ with $\gamma \leq \gamma_{\max}$. Then $(x_k)_{k\geq0}$ admits a unique stationary distribution $\pi_\gamma \in \mathcal{P}_2(\mathbb{R}^d)$, and additionally:

(i) $|\mathbb{E}[\ell(x_k)] - \mathbb{E}_{x\sim\pi_\gamma}[\ell(x)]| \leq c(1-\rho)^k \qquad \forall \ell : |\ell(x)| \leq L_\ell(1 + \|x\|),$

(ii) $|\mathbb{E}_{x\sim\pi_\gamma}[\ell(x)] - \ell(x^*)| \leq L_\ell\sqrt{C} \qquad \forall \ell : L_\ell - \text{Lipschitz functions},$

for some $c < \infty$, $\rho \in (0,1)$, $C = \Theta(\text{MSE}(\text{SGD}-\text{RR}_1))$ and $\gamma_{\max}$ defined in Theorem 4.1

**Remark 2.** *Item (i) of Theorem 4.3 shows that Perturbed SGD-RR$_1$ converges geometrically in total variation to $\pi_\gamma$. Item (ii) bounds the gap between the expectation of a measurement under $\pi_\gamma$ and its value at the solution $x^*$. Intuitively, if the method converged exactly to $x^*$, these expectations would coincide.*

The result of Theorem 4.3 follows from a Foster–Lyapunov drift condition combined with a minorization argument, showing that the induced Markov chain satisfies the standard ergodicity criteria in the spirit of Yu et al. (2021); Vlatakis-Gkaragkounis et al. (2024). Beyond geometric ergodicity, one may also ask whether the chain admits asymptotic statistical estimation of functionals of its trajectory. By invoking the Birkhoff–Khinchin ergodic theorem for continuous-state Markov chains, we establish both a Law of Large Numbers (LLN) and a Central Limit Theorem (CLT) for empirical averages of test functions evaluated along the epoch iterates.

**Theorem 4.4** (LLN and CLT for Perturbed SGD–RR$_1$). Suppose Assumptions 2.1–2.3 hold and run Perturbed SGD–RR$_1$ with $\gamma \leq \gamma_{\max}$, (cf. Theorem 4.1). Let $\ell : \mathbb{R}^d \to \mathbb{R}$ be any test function such that $|\ell(x)| \leq L_\ell(1 + \|x\|^2)$ and $\mathbb{E}_{x\sim\pi_\gamma}[\ell(x)] < \infty$. Then for the epoch-level iterates, it holds that:

$$\underbrace{\frac{1}{T} \sum_{t=0}^{T-1} \ell(x_t) \xrightarrow{\text{a.s.}} \mathbb{E}_{x\sim\pi_\gamma}[\ell(x)]}_{\text{(LLN)}} \qquad \underbrace{T^{-1/2} \sum_{t=0}^{T-1} \Big(\ell(x_t) - \mathbb{E}_{x\sim\pi_\gamma}[\ell(x)]\Big) \xrightarrow{d} \mathcal{N}(0, \sigma_{\pi_\gamma}^2(\ell))}_{\text{(CLT)}},$$

where $\sigma_{\pi_\gamma}^2(\ell) = \lim_{T\to\infty} \frac{1}{T} \mathbb{E}_{\pi_\gamma}[S_T^2]$ and $S_T^2 = \sum_{t=0}^{T-1} \big(\ell(x_t) - \mathbb{E}_{x\sim\pi_\gamma}[\ell(x)]\big)^2$.

---

[4] On the augmented space $\mathbb{R}^d \times \mathfrak{S}_n$, the chain $((x_k, \omega_k))_{k\geq0}$ is also time-homogeneous with kernel $K\big((x,\omega), A \times B\big) = \int_A \phi(y; H(x,\omega), \Sigma I_d)\, dy \cdot \frac{|B|}{n!}$. The above formulation is convenient for verifying Lyapunov–Foster and minorization criteria, since the coupling with uniform perturbation remains independent.

## 4.2 Convergence Guarantees for $\mathrm{RR}_2 \oplus \mathrm{RR}_1$

Having established the role of $\mathrm{RR}_1$ within stochastic algorithms, we now examine its interplay with $\mathrm{RR}_2$ and the effect of combining these heuristics on the bias term. The previous results hold for the full class of weakly quasi-strongly monotone problems with $\lambda > 0$. To sharpen our understanding, we focus on the quasi-strongly monotone case ($\lambda = 0$ in Assumption 2.2), which already covers a broad range of non-monotone regimes (Loizou et al., 2020). A key step in our analysis is to bound higher-order moments of the deviation between $\mathrm{RR}_1$ iterates and the solution of (VIP), thereby showing that the bias of Perturbed SGD–$\mathrm{RR}_1$ is linear in the step size with quadratic corrections.

Technically, our analysis relies on two delicate ingredients that go beyond straightforward generalizations. (i) A *spectral study of the full-pass operator* (Lemma F.2), which approximates the underlying map $F$ of the VIP. This connection between $\mathrm{RR}_1$ and the multi-step extragradient literature may be of independent interest, but its proof requires a nontrivial handling of spectral properties across reshuffled passes. (ii) A *combinatorial lemma* (Lemma E.2) that bounds fourth order moments of finite-sum subsets of vectors. While reminiscent of Mishchenko et al. (2020b, Lemma 1, Sec. 7), our result demands substantially more intricate manipulations to accommodate the dependencies introduced by sampling without replacement.

> **Lemma 4.5.** Let $\lambda = 0$ and Assumptions 2.1–2.4 hold. If $\gamma \leq \gamma_{\max}$ (cf. Lemma E.4), then
> $$\mathrm{bias}(\texttt{Perturbed SGD-RR}_1) = \limsup_{k \to \infty} \| \mathbb{E}[x_k] - x^* \| = C(x^*)\gamma + \mathcal{O}(\gamma^3).$$

**Remark 3.** *For classical* SGD, *the bias takes the form* $\mathrm{bias}(\texttt{SGD}) = C(x^*)\gamma + \mathcal{O}(\gamma^{1.5})$ *(Dieuleveut et al., 2020). Hence, while* $\mathrm{RR}_1$ *retains the same first-order term, it improves the higher-order contribution and simultaneously yields sharper mean-squared error guarantees.*

Building on this fact, we construct a refined trajectory via the debiasing scheme $\mathrm{RR}_2$. Our final result shows that the combined scheme attains exponentially fast a provable asymptotic $\mathcal{O}(\gamma^3)$ bias:

> **Theorem 4.6.** Under the assumptions of Lemma 4.5, the output of Algorithm 1 satisfies
>
> Last-iterate version (line 9): $\qquad \| \mathbb{E}[x_k] - x^* \| \ \leq \ c(1 - \rho)^k + \mathcal{O}(\gamma^3),$
>
> Averaged-iterate version (line 10): $\qquad \left\| \mathbb{E}\left[ \frac{1}{k} \sum_{m=1}^{k} x_m \right] - x^* \right\| \ \leq \ \frac{c/\rho}{k} + \mathcal{O}(\gamma^3).$
>
> where $\rho \in (0,1)$, $c < \infty$ (cf. Theorem 4.3).

**Remark 4.** *Although the* last-iterate *estimator is often preferred in theory, in practice a trade-off emerges vs* ergodic-average*: full-epoch or tailed averaging (the Polyak–Ruppert scheme (Polyak & Juditsky, 1992)) achieves improved variance properties, asymptotically captured by Theorem 4.4.*

## 5 Experiments

In this section, we conduct a series of experiments demonstrating the benefits from the synergy of the two heuristics empirically. More specifically, we focus in the strongly monotone setting and compare the relative error and bias attained by 4 variants: the classical SGD(A) algorithm using uniform with-replacement sampling (denoted as *SGDA* in the plots), the one equipped with $\mathrm{RR}_1$, the one equipped with $\mathrm{RR}_2$ and the method utilizing both heuristics. For each experiment, we report the average of 5 trials/runs and plot the relative error $\log \left( \frac{\|x_k - x^*\|^2}{\|x_0 - x^*\|^2} \right)$ with respect to the iterations of the algorithm.

***Two-player Zero-Sum Games.*** In the strongly monotone case, we consider the two-player zero-sum game from Emmanouilidis et al. (2024) and Loizou et al. (2021), consisting a strongly convex -strongly concave quadratic of the form

$$\min_{x_1 \in \mathbb{R}^d} \max_{x_2 \in \mathbb{R}^d} f(x_1, x_2) = \frac{1}{n} \sum_{i=1}^{n} \frac{1}{2} x_1^T A_i x_1 + x_1^T B_i x_2 - \frac{1}{2} x_2^T C_i x^2 + \alpha_i^T x_1 - c_i^T x_2.$$

For the interested reader, additional details on the experimental setup and the procedure used to sample the matrices $A_i, B_i, C_i$ are provided in Appendix G.

**On the Rate of Convergence.** In the first set of experiments, we aim to validate empirically the result of Theorem 4.1 by running SGDA with $\mathrm{RR}_1$ and using the step sizes described by theory. We conduct experiments for multiple conditions $\kappa = \frac{L}{\mu}$ with value $\kappa = \{1, 5, 10\}$ and $\mu = 1$. In Figure 2, we observe that the algorithm with $\mathrm{RR}_1$ converges linearly to a neighbourhood around the solution $x^*$ and the neighbourhood depends on the step size used, validating in this way the results of Theorem 4.1. We have run experiments also for stepsizes that are larger than the ones predicted in theory, observing similar behaviour of the optimization algorithm. Additionally, we have performed an ablation study in Wasserstein GANs (Emmanouilidis et al., 2024; Daskalakis et al., 2017), showing that the performance benefit of the proposed heuristic is universal in many other common optimization algorithms used in VIs (Appendix G).

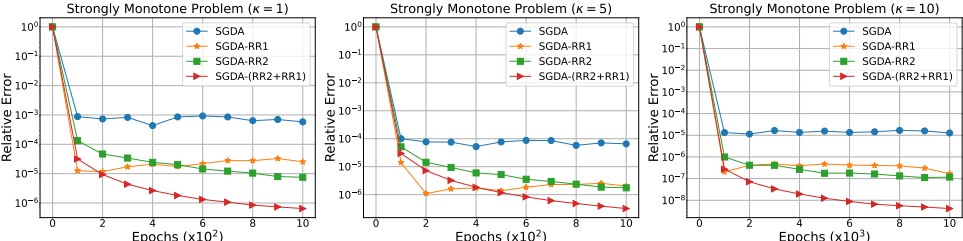

Figure 2: Comparison of different heuristics. The combination of $\mathrm{RR}_2 \oplus \mathrm{RR}_1$ converges linearly to neighborhood of the solution, validating the established theoretical results (Theorem 4.1). Even when we are using the last iterates, the combination of $\mathrm{RR}_2 \oplus \mathrm{RR}_1$ converges to a smaller relative error in comparison to the other variants (classical SGDA, $\mathrm{RR}_1$, $\mathrm{RR}_2$). This validates that the bias of Algorithm 1 is improved even when $\mathrm{RR}_1$-last iterates are used.

**Efficient Statistics & Empirical Concentration.** This set of experiments examines the central limit theorem (CLT) and aims to validate empirically the theoretical results established in Theorem 4.4. The value of the game, which is zero, is used as the test value for which we observe the averaged evaluations after $T = \{100, 500, 1000\}$ iterations respectively. In particular, we run the algorithm with the step size suggested by Theorem 4.4 and maintain for the total number of iterations the sum of the evaluations, normalized with $\sqrt{T}$. We run the experiment for $T = 2000$ trials/runs and plot the corresponding histograms. In Figure 3, we observe that the histograms tend to concentrate to the value of the game as the number of iterations increase. Additionally, we examine the effect of the step size to concentration of the observed distributions.

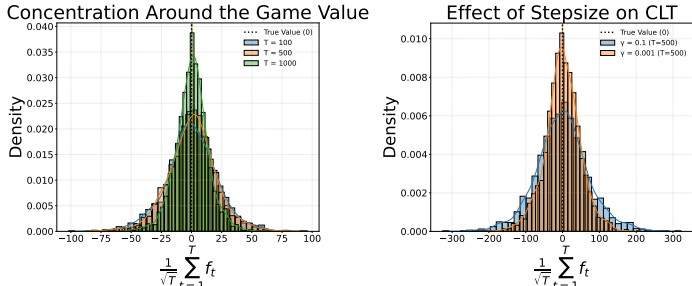

Figure 3: Validation of concentration around the mean and effect of the number of iterations and step sized selected. The average of the game values tend to concentrate more around the mean for larger number of iterations and smaller step sizes. The right plot indicates the effect of two different step sizes $\gamma \in \{0.1, 0.001\}$, showing that for smaller step sizes the corresponding distribution attains higher concentration around the mean of the values.

## 6 CONCLUSION

In summary, our work establishes that the synergy of random reshuffling and extrapolation yields a principled reduction of bias, culminating in accelerated convergence guarantees for structured non-monotone VIPs. By combining Markov chain techniques, spectral analysis, and higher-moment bounds, we provide the first rigorous evidence that these heuristics can be synergistically integrated rather than studied in isolation. This perspective bridges a long-standing gap between practice and

theory, offering a systematic framework that extends naturally to a broad class of constant step-size stochastic methods. We view this as a foundation for a new generation of analyses where practical heuristics are not only empirically verified but also theoretically grounded to deliver provable performance improvements in complex stochastic optimization landscapes.

ACKNOWLEDGMENTS

Konstantinos Emmanouilidis and René Vidal acknowledge support from the NSF-Simons grant 2031985 "Research Collaborations on the Mathematical and Scientific Foundations of Deep Learning". Emmanouil-Vasileios Vlatakis-Gkaragkounis acknowledges support from the startup funding at Wisconsin Madison university under the grant (233-AAN5596 - GF000020971).

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

# Supplemental Material

## Contents

## A   ADDITIONAL RELATED LITERATURE

Owing to its central role in optimization and machine learning, stochastic gradient descent (SGD) and its numerous variants have generated an extensive body of literature that spans several decades. A complete survey is well beyond the scope of this paper, so we restrict ourselves to highlighting the most relevant threads and pointers.

The only aspects we emphasize at this point are the phenomena most pertinent to our analysis:

1. classical stochastic approximation and asymptotic normality results;
2. constant step-size schemes viewed through the lens of Markov chains and diffusion approximations;
3. the widespread use of random reshuffling and its still-developing theoretical guarantees; and
4. challenges that arise in min–max and variational inequality settings.

These themes form the backbone of our extended discussion in the appendix, where we provide a more comprehensive account of prior work.

**From classical stochastic approximation to modern SGD.**   The study of stochastic approximation predates machine learning by decades, beginning with the foundational work of Robbins & Monro (1951) and Kiefer & Wolfowitz (1952). Early analyses focused on vanishing step-sizes obeying the classic $L^2$–$L^1$ summability rules, and developed the ODE method to describe limiting dynamics; see, e.g., Ljung (1978; 2003); Benaïm (2006); Bertsekas & Tsitsiklis (2000b). In parallel, a rich line of results examined the almost-sure behavior of stochastic approximation, including avoidance of saddle points and convergence to locally stable equilibria (Pemantle, 1990; Brandière & Duflo, 1996; Benaïm & Hirsch, 1995; Hsieh et al., 2021; 2023; Jordan et al., 1998; Mertikopoulos et al., 2020; 2024; Staib et al., 2019; Antonakopoulos et al., 2022).

**Asymptotic normality and statistical inference.**   A complementary thread established central limit theorems for stochastic approximation: classical milestones include Chung (1954); Sacks (1958); Fabian (1968); Ruppert (1988); Shapiro (1989), culminating in the Polyak–Juditsky averaging principle (Polyak & Juditsky, 1992). Under suitable decaying step-sizes, the averaged SGD iterate is asymptotically normal and attains the Cramér–Rao optimal variance. This statistical perspective has been leveraged to construct confidence intervals and inference procedures for SGD-based estimators (Tripuraneni et al., 2018; Su & Zhu, 2018; Toulis & Airoldi, 2017; Fang et al., 2018).

**Constant step sizes: bias, speed, and Markovian viewpoints.**   Constant step-size policies, now standard in large-scale learning, trade a nonvanishing asymptotic error for fast initial progress and robust practical performance. Their benefits in over-parameterized regimes are well documented (Schmidt & Roux, 2013; Needell et al., 2014; Ma et al., 2018; Vaswani et al., 2019). The Markov chain viewpoint provides a unifying language for analyzing such constant-step schemes: early developments used dynamical-systems and Markov-process techniques to establish stability and ergodic properties (Kifer, 1988; Benaim, 1996; Priouret & Veretenikov, 1998; Fort & Pages, 1999; Aguech et al., 2000), with recent refinements quantifying convergence behavior and variance (Dieuleveut et al., 2020; Chee & Toulis, 2018; Tan & Vershynin, 2023). In parallel, diffusion-based analyses and Langevin-type discretizations connect SGD to MCMC methodology, yielding non-asymptotic guarantees and sharp mixing rates in log-concave and beyond-log-concave settings (Dalalyan, 2017; Durmus & Moulines, 2017; Cheng et al., 2018b; Dalalyan & Karagulyan, 2019; Brosse et al., 2017; Cheng et al., 2018a; Bubeck et al., 2018; Dwivedi et al., 2019; Dalalyan & Riou-Durand, 2020; Li et al., 2019; Shen & Lee, 2019; Erdogdu & Hosseinzadeh, 2021).

**Random reshuffling vs. with-replacement sampling.**   Among finite-sum methods, *random reshuffling* (RR) occupies a special place: each epoch processes every component exactly once in a random order, in contrast to classical with-replacement SGD. RR is ubiquitous in practice— it improves cache locality (Bengio, 2012), often converges faster than with-replacement sampling (Bottou, 2009; Recht & Ré, 2013), and is the default in deep learning pipelines (Sun, 2020). The

success of RR contrasts with the mature theory for with-replacement SGD, which enjoys tight upper/lower bounds in many regimes (Rakhlin et al., 2012; Drori & Shamir, 2019; Nguyen et al., 2019). A key analytical hurdle is bias: within an epoch, conditional expectations are no longer unbiased gradients, so classical SGD proofs do not transfer verbatim. Early attempts leveraging the noncommutative arithmetic–geometric mean conjecture (Recht & Ré, 2012) were later undermined when the conjecture was disproved (Lai & Lim, 2020). More recent works establish rates for twice-smooth and smooth objectives and highlight gaps between theory and prevalent heuristics (Gürbüzbalaban et al., 2019; Haochen & Sra, 2019; Nagaraj et al., 2019; Safran & Shamir, 2020; Rajput et al., 2020).

**Incremental/ordered passes and sensitivity to permutations.** Before RR became the default, incremental gradient (IG) methods with fixed orderings were widely used in neural network training (Luo, 1991; Grippo, 1994), with asymptotic convergence known since early work (Mangasarian & Solodov, 1994; Bertsekas & Tsitsiklis, 2000a). However, their performance can depend strongly on the chosen ordering (Nedić & Bertsekas, 2001; Bertsekas, 2011). By randomizing the order every epoch, RR mitigates this sensitivity and—under smoothness—can outperform both with-replacement SGD and deterministic IG (Gürbüzbalaban et al., 2019; Haochen & Sra, 2019), with refined lower/upper bounds developed in follow-up studies (Nagaraj et al., 2019; Safran & Shamir, 2020; Rajput et al., 2020).

**Min–max problems and variational inequalities.** In large-scale saddle-point and VIP settings, most theoretical analyses assume with-replacement sampling for convenience, whereas implementations overwhelmingly adopt without-replacement sampling (Bottou, 2012a). A growing literature is closing this gap: for minimization problems, several works show (sometimes provably faster) RR rates in finite-sum regimes (Mishchenko et al., 2020a; Ahn et al., 2020; Gürbüzbalaban et al., 2021; Cai et al., 2023). For min–max and VIPs, guarantees remain comparatively sparse: Chen & Rockafellar (1997) and Korpelevich (1976) initiated the study of stochastic and extragradient-type methods, with modern analyses for SEG and optimistic variants (Gorbunov et al., 2022a;b; Hsieh et al., 2019; Choudhury et al., 2023). For RR specifically, Das et al. (2022) derive guarantees for SGDA and PPM under strong structural conditions, and Cho & Yun (2023) extend to certain non-monotone settings. Nevertheless, classical SGDA can diverge even in simple monotone bilinear games, while proximal methods are implicit and less practical; filling this theoretical–practical gap remains an active direction.

**Overparameterization and global convergence phenomena.** Finally, SGD training dynamics in overparameterized neural networks reveal regimes where global convergence can emerge from structural properties such as width, depth, and initialization (Du et al., 2019; Zou et al., 2020; Nguyen & Mondelli, 2020; Liu et al., 2023). These results are powerful but specialized: they rely on problem-specific structure (e.g., width scaling or tailored initializations). Our focus is orthogonal—we seek guarantees for general non-convex or non-monotone landscapes under stochastic approximation, independent of architectural assumptions. For completeness, we refer the reviewer for the related work of the aforementioned work for further surveys about these SGD & overparameterization results in more detail.

**Comparison to Prior Work and Overview of Our Contributions.** Before introducing the intuition behind our algorithmic design, we briefly contrast our results with those of Emmanouilidis et al. (2024), who study $RR_1$-based improvements for the Stochastic Extragradient (SEG) method. Our analysis uncovers a fundamentally different phenomenon: the joint use of $RR_1 \oplus RR_2$ produces a *bias cancellation mechanism* that eliminates the leading $\mathcal{O}(\gamma)$ term while preserving the condition number and asymptotic behavior of SGDA. The key distinctions are summarized in Table 1. Achieving the best of both worlds—optimal bias order together with a tight condition number, as SEG without reshuffling enjoys—remains an interesting direction for future work.

*Summary.* To summarize, there is a mature theory for with-replacement SGD (both asymptotic and non-asymptotic), well-developed statistical limits via averaging, and powerful diffusion/Markov perspectives for constant-step schemes. RR, despite being the practical default, poses distinctive analytical challenges due to its within-epoch bias, especially in min–max and VIP settings. Recent advances begin to bridge this gap, but a comprehensive understanding of how classical heuris-

tics (constant steps, reshuffling, extrapolation) interact remains incomplete—precisely the juncture where our work contributes.

## OUR MODEL'S ASSUMPTIONS.

While variational inequalities provide a unifying language for optimization, learning, and game dynamics, *no single structural assumption can capture the full complexity of all modern nonconvex–nonconcave problems*. From a computational standpoint, even smooth VIs are intractable in full generality—being tightly connected to Nash equilibria (Papadimitriou et al., 2022; Goldberg & Katzman, 2022), linear complementarity (IEOR, 2011), and constrained saddle-point problems (Daskalakis et al., 2021). Consequently, much of the theoretical literature adopts *structured* assumptions (strong convexity, quasi-strong monotonicity, quasar or weak convexity, PL/KL conditions, Minty conditions, error bounds, etc.), each expressive in specific regimes but not universal.

Our work is based on *quasi-strong monotonicity* which falls squarely within this class: it captures stabilizing behaviors of many smooth systems, while remaining far more permissive than strong convexity or global monotonicity. At the same time, it is helpful to clarify that this assumption is not meant as a universal model for all adversarial or fully nonconvex–nonconcave settings. Certain modern ML applications— including GANs, adversarial robustness, and multi-agent RL—can exhibit fundamentally unstable or rotational dynamics (Jin et al., 2020; Han et al., 2023; Kim & Seo, 2022; Bukharin et al., 2023), where even *local* monotonicity surrogates fail. As such, our theoretical guarantees should be viewed as pertaining to regimes where a minimum amount of local structure is present, rather than to the most adversarial or unstructured cases. [5]

## LIMITATIONS

**Limitations of the structural assumption.** A central structural assumption in our analysis is that the operator $F$ satisfies $\lambda$-weak $\mu$-quasi-strong monotonicity. This condition is broad enough to include several meaningful non-monotone problem classes—such as dissipative dynamical systems, weakly convex optimization, and locally contractive variational inequalities—and is standard in modern analyses of stochastic fixed-point and operator-splitting methods (e.g., Hsieh et al., 2019; Mertikopoulos and Zhou, 2019; Chavdarova et al., 2021). However, it is important to emphasize the following limitations.

1. **Not applicable to general nonconvex–nonconcave min–max problems.** The assumption does *not* hold for arbitrary adversarial problems such as deep GANs, multi-agent RL environments, or smooth non-monotone games with persistent rotational dynamics. These settings may lack even local stability (e.g., Daskalakis et al. 2018; Fiez et al. 2020). Accordingly, our theoretical guarantees should not be interpreted as applying to fully adversarial or worst-case min–max formulations.

2. **Local nature of the assumption.** Quasi-strong monotonicity is inherently a *local* regularity condition: smooth operators that are monotone in a neighborhood of a solution satisfy it on that region (Lemma A.4, Hsieh et al. 2019). This requires smoothness and regularity that may not hold in problems involving discontinuities, clipping, piecewise-linear losses, or hard constraints. In such cases, the assumption may fail even locally.

3. **Not capturing highly oscillatory or anti-monotone operators.** Allowing $\lambda > 0$ permits controlled violation of monotonicity, but the assumption still does not model strongly anti-monotone or highly oscillatory operators. Extending our analysis to Minty variational inequalities, hypomonotone operators, or other generalized monotonicity classes remains an interesting direction for future work.

---

[5] A complementary and key fact for our setting, established in (Hsieh et al., 2019, Lemma A.4), is that *any smooth VI operator is locally quasi-strongly monotone in a neighborhood of a regular solution*. Combined with our Markov-chain recurrence result—which ensures that the iterates remain in such neighborhoods with probability 1—this provides a natural and widely adopted stability regime in which the $RR_1$ and $RR_2$ debiasing mechanisms are both theoretically justified and practically meaningful.

Despite these limitations, we believe the assumption remains meaningful for a broad set of structured, smooth VIPs where local stability is present. We hope that this explicit discussion helps avoid any misunderstanding about the scope of applicability of our results.

## B  Proof Roadmap

Our main theorem relies on several technical components, developed across different parts of the appendix. In Appendix C, we establish the mean-squared convergence rate of Perturbed SGD–$RR_1$. While this result is of independent interest—as it extends prior analyses to a noisy setting—it primarily serves to construct the Lyapunov function that underpins our Markov chain treatment of the algorithm. Armed with this Lyapunov function, the properties of the perturbation, and the epoch-level viewpoint, Appendix D shows that SGD–$RR_1$ forms a geometrically ergodic Markov chain with all standard consequences: existence of an invariant measure, a law of large numbers, and a central limit theorem.

Appendix E develops higher-moment control. In particular, part E.2 introduces a new combinatorial lemma on fourth moments of finite-sum subsets of vectors—a technically challenging step, motivated by the fact that most extrapolation analyses (e.g., Dieuleveut et al. (2020)) require bounded fourth moments of the reshuffling estimator. With this tool in hand, Appendix F shows that no change in the step-size order is required to accommodate the extrapolation trick: we are able to control the higher moments of the Jacobian of the reshuffled biased gradient estimator. To the best of our knowledge, this is the first such result. The last parts of Appendix F then contain the full proofs of our main theorem.

Finally, Appendix G presents additional experiments demonstrating the practical gains of our method, which originally motivated this study.

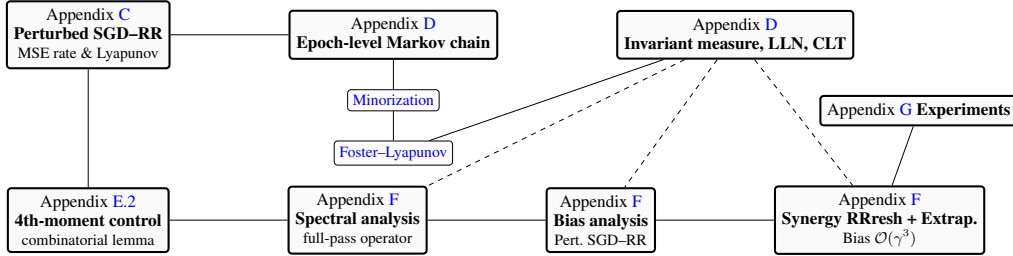

Figure 4: Dependency graph of Appendix results. Solid lines: main logical flow. Dashed lines: auxiliary inputs.

### B.1  Warm-Up: Useful Inequalities

We start our technical appendix by providing inequalities that will be useful in our proofs

$$\left\| \sum_{i=1}^{n} x_i \right\|^2 \leq n \sum_{i=1}^{n} \|x_i\|^2 \tag{2}$$

$$\left\| \sum_{i=1}^{n} x_i \right\|^4 \leq n^3 \sum_{i=1}^{n} \|x_i\|^4 \tag{3}$$

$$\|a - b\|^2 \geq \frac{1}{2} \|a\|^2 - \|b\|^2 \tag{4}$$

$$\langle a, b \rangle = \frac{1}{2} \left[ \|a\|^2 + \|b\|^2 - \|a - b\|^2 \right] \tag{5}$$

$$e^{-x} \geq 1 - x, \forall x \geq 0 \tag{6}$$

$$\|a + b\|^2 \leq \frac{1}{t} \|a\|^2 + \frac{1}{1-t} \|b\|^2, \forall t \in (0, 1) \tag{7}$$

## C  PROOF OF CONVERGENCE RATE (MSE) OF PERTURBED SGD–RR₁ (THEOREM 4.1)

Our first result concerns the *Perturbed SGD–RR₁* variant for $\lambda$-weak $\mu$-quasi strongly monotone VIPs.

**Theorem C.1** (Restatement of Theorem 4.1). Let Assumptions 2.1-2.3 hold. Then the iterates of Perturbed SGD–RR₁ satisfy for $\gamma \leq \gamma_{max}$,

$$\mathbb{E}\left[\|x_{k+1}^0 - x_*\|^2\right] \leq \left(1 - \frac{\gamma n \mu}{2}\right)^{k+1} \|x_0^0 - x^*\|^2 + \frac{8n\gamma^2 L_{max}^2}{\mu^2}\sigma_*^2 + \frac{8\lambda}{\mu}$$

where $\sigma_*^2 = \frac{1}{n}\sum_{i=0}^{n-1}\|F_i(x^*)\|^2$ and $\gamma_{max} = \min\left\{\frac{1}{3nL_{max}}, \frac{\sqrt{1+6\mu^2 L_{max}^2}-1}{12nL_{max}^2}\right\}$.

We first provide some notation that will be necessary for establishing the proof of Theorem 4.1. Consider the epoch-wise update rule of Perturbed SGD–RR₁

$$x_{k+1}^0 = x_k^n \quad = \quad x_k^0 - \gamma\sum_{i=0}^{n-1} F_{\omega_k^i}(x_k^i) - \gamma U_k$$
$$= \quad x_k^0 - \gamma G_{\omega_k}(x_k^0) - \gamma U_k \tag{8}$$

where $G_{\omega_k}(x_k^0) := \sum_{i=0}^{n-1} F_{\omega_k^i}(x_k^i)$ denotes the epoch-wise operator used to update the epoch-level iterates $(x_k)_{k \geq 0}$.

### C.1  PREPARATORY LEMMAS & PROPOSITIONS

With this notation at hand, we proceed in proving two Lemmas that are necessary for deriving the rate of convergence of the Theorem 4.1. In the first lemma, following the high-level intuition that one epoch of random reshuffling with step size $\gamma$ progresses the underlying dynamics approximately equal to one step of the deterministic GD with step size $\gamma' = n\gamma$, in the first lemma we bound the "progress" that the deterministic algorithm makes in one step.

**Lemma C.2.** Let Assumptions 2.1-2.3 hold. For any $x^* \in \mathcal{X}^*$, the iterates of Perturbed SGD–RR₁ satisfy that

$$\mathbb{E}\left[\|x_{k+1} - x^* - \gamma n F(x_k)\|^2 \mid \mathcal{F}_k\right] \leq \left[(1 - \gamma n \mu)^2 + \gamma^2 n^2 L_{max}^2\right]\|x_k - x^*\|^2 + 2\gamma n \lambda$$

*Proof.* For any fixed $x^* \in \mathcal{X}^*$, it holds that

$$\|x_{k+1} - x^* - \gamma n F(x_k)\|^2 \quad = \quad \|x_k - x^*\|^2 - 2\gamma n \langle x_k - x^*, F(x_k)\rangle + \gamma^2 n^2 \|F(x_k)\|^2$$
$$\leq \quad \|x_k - x^*\|^2 - 2\gamma n \mu \|x_k - x^*\|^2 + 2\gamma n \lambda + \gamma^2 n^2 \|F(x_k)\|^2$$
$$\leq \quad (1 - 2\gamma n \mu)\|x_k^0 - x^*\|^2 + 2\gamma n \lambda + \gamma^2 n^2 \|F(x_k)\|^2$$
$$\overset{\text{Assumption 2.3}}{\leq} \quad (1 - 2\gamma n \mu + \gamma^2 n^2 L_{max}^2)\|x_k - x^*\|^2 + 2\gamma n \lambda \tag{9}$$

Taking expectation condition on the filtration $\mathcal{F}_k$, gives

$$\mathbb{E}\left[\|x_{k+1} - x^* - \gamma n F(x_k)\|^2 \mid \mathcal{F}_k\right] \leq (1 - 2\gamma n \mu + \gamma^2 n^2 L_{max}^2)\|x_k - x^*\|^2 + 2\gamma n \lambda$$
$$\leq \left[(1 - \gamma n \mu)^2 + \gamma^2 n^2 L_{max}^2\right]\|x_k - x^*\|^2 + 2\gamma n \lambda$$
□

Having an expression for the progress made by the deterministic counterpart of Perturbed SGD–RR₁, we next aim to bound how large the deviation of the two algorithms becomes inside an epoch. To do so, we bound the sum of the distances of the iterates obtain by Perturbed SGD–RR₁ from the start of the epoch, which corresponds to the fictitious iterate of our comparator deterministic counterpart. The following lemma provides an upper bound dependent on the distance of the current epoch-level iterate from the solution and the variance at the optimum.

**Lemma C.3.** Let Assumptions 2.1, 2.3 hold. If Perturbed SGD–RR$_1$ is run with step size $\gamma \leq \frac{1}{\sqrt{3n(n-1)L_{max}}}$, then it holds that

$$\mathbb{E}\Big[\sum_{i=1}^{n-1}\big\|x_k^i - x_k^0\big\|^2 \mid \mathcal{F}_k\Big] \leq 6n^3\gamma^2 L_{max}^2\big\|x_k^0 - x^*\big\|^2 + 2n^2\gamma^2\sigma_*^2$$

*Proof.* From the epoch-level update (8), it holds

$$\|x_k^i - x_k^0\|^2 = \gamma^2 i^2 \left\|\frac{1}{i}\sum_{j=0}^{i-1} F_{\omega_k^j}(x_k^j)\right\|^2$$

$$\overset{(2)}{\leq} 3\gamma^2 i\sum_{j=0}^{i-1}\left\|F_{\omega_k^j}(x_k^j) - F_{\omega_k^j}(x_k^0)\right\|^2 + 3\gamma^2 i^2\left\|\frac{1}{i}\sum_{j=0}^{i-1} F_{\omega_k^j}(x_k^0) - F(x_k^0)\right\|^2$$

$$+ 3\gamma^2 i^2\big\|F(x_k^0)\big\|^2$$

$$\overset{\text{Assumption } 2.3}{\leq} 3\gamma^2 L_{max}^2 i\sum_{j=0}^{i-1}\left\|x_k^j - x_k^0\right\|^2 + 3\gamma^2 i^2\left\|\frac{1}{i}\sum_{j=0}^{i-1} F_{\omega_k^j}(x_k^0) - F(x_k^0)\right\|^2$$

$$+ 3\gamma^2 i^2\big\|F(x_k^0)\big\|^2$$

where at the last step we have used the Lipschitz property of the operators $F_i, \forall i \in [n]$. Taking expectation condition on the filtration $\mathcal{F}_k$, we get

$$\mathbb{E}\left[\big\|x_k^i - x_k^0\big\|^2\Big|\mathcal{F}_k\right] \leq 3\gamma^2 L_{max}^2 i\, \mathbb{E}\left[\sum_{j=0}^{i-1}\big\|x_k^j - x_k^0\big\|^2\Big|\mathcal{F}_k\right]$$

$$+ 3\gamma^2 i^2\, \mathbb{E}\left[\left\|\frac{1}{i}\sum_{j=0}^{i-1} F_{\omega_k^j}(x_k^0) - F(x_k^0)\right\|^2\Big|\mathcal{F}_k\right] + 3\gamma^2 i^2\big\|F(x_k^0)\big\|^2 \quad (10)$$

From Lemma A.3 in (Emmanouilidis et al., 2024), it holds for $A = \frac{2}{n}\sum_{i=0}^{n-1} L_i^2$, $\sigma_*^2 = \frac{1}{n}\sum_{i=0}^{n-1}\|F_i(x^*)\|^2$ and $\forall i \in [n]$ that

$$i^2\, \mathbb{E}\left[\left\|\frac{1}{i}\sum_{j=0}^{i-1} F_{\omega_k^j}(x_k^0) - F(x_k^0)\right\|^2\Big|\mathcal{F}_k\right] \leq \frac{i(n-i)}{n-1}\left(A\|x_k^0 - x^*\|^2 + 2\sigma_*^2\right) \quad (11)$$

From inequality (11) and (10), thus, we obtain

$$\mathbb{E}\left[\big\|x_k^i - x_k^0\big\|^2\Big|\mathcal{F}_k\right] \leq 3\gamma^2 L_{max}^2 i\, \mathbb{E}\left[\sum_{j=0}^{i-1}\big\|x_k^i - x_k^0\big\|^2\Big|\mathcal{F}_k\right] + 3\gamma^2\frac{i(n-i)}{n-1}A\big\|x_k^0 - x^*\big\|^2$$

$$+ 6\gamma^2\frac{i(n-i)}{n-1}\sigma_*^2 + 3\gamma^2 i^2\big\|F(x_k^0)\big\|^2 \quad (12)$$

By summing over $0 \leq i \leq n-1$ we have that

$$\sum_{i=0}^{n-1}\mathbb{E}\left[\big\|x_k^i - x_k^0\big\|^2\Big|\mathcal{F}_k\right] \leq 3\gamma^2 L_{max}^2\frac{n(n-1)}{2}\sum_{i=0}^{n-1}\mathbb{E}\left[\big\|x_k^i - x_k^0\big\|^2\Big|\mathcal{F}_k\right] + \gamma^2 A\frac{n(n+1)}{2}\big\|x_k^0 - x^*\big\|^2$$

$$+ \gamma^2 n(n+1)\sigma_*^2 + \frac{\gamma^2 n(n-1)(2n-1)}{2}\big\|F(x_k^0)\big\|^2, \quad (13)$$

where we used the facts

$$\sum_{i=0}^{n-1} i = \frac{n(n-1)}{2}, \quad \sum_{i=0}^{n-1} i^2 = \frac{n(n-1)(2n-1)}{6}, \quad \sum_{i=0}^{n-1}\frac{i(n-i)}{n-1} = \frac{n(n+1)}{6}.$$

For $\gamma \leq \frac{1}{\sqrt{3n(n-1)L_{max}}}$, rearranging the terms in (13) we obtain

$$
\begin{aligned}
\sum_{i=0}^{n-1} \mathbb{E}\left[\left\|x_k^i - x_k^0\right\|^2 \middle| \mathcal{F}_k\right] &\leq \gamma^2 n(n+1)A\left\|x_k^0 - x^*\right\|^2 + 2n(n+1)\gamma^2\sigma_*^2 \\
&\quad + n(n-1)(2n-1)\gamma^2\left\|F(x_k^0)\right\|^2 \\
&\overset{\text{Assumption 2.3}}{\leq} 2\gamma^2 n^2(A + nL^2)\left\|x_k^0 - x^*\right\|^2 + 2n^2\gamma^2\sigma_*^2 \\
&\overset{A \leq 2L_{max}^2}{\leq} 6n^3\gamma^2 L_{max}^2\left\|x_k^0 - x^*\right\|^2 + 2n^2\gamma^2\sigma_*^2
\end{aligned}
$$

$\square$

In the preceding subsection, we established a series of preparatory lemmas. We now combine these ingredients into a unified elementwise argument to prove Theorem 4.1.

## C.2 ASSEMBLING THE LEMMAS: PROOF OF THEOREM 4.1

In this section, we provide the proof of Theorem 4.1, establishing linear convergence of Perturbed SGD–RR$_1$ to a neighbourhood of the solution. The proof technique leverages the interpretation that one epoch of Perturbed SGD–RR$_1$ with sufficiently small step size $\gamma > 0$ is equivalent to one step of the gradient descent with step size $\gamma' = n\gamma$, as the iterates of Perturbed SGD–RR$_1$ inside the epoch do not change drastically. To account for the deviation of the iterates from the initial state $x_k^0$ inside the epoch, we have upper bounded the sum of the corresponding distances in Lemma C.3. Thus, using the combining the bound on the progress made by gradient descent from Lemma C.2 with the potential "deviation" between the two algorithms we establish the rate of convergence of the method.

*Proof.* Using the update rule of Perturbed SGD–RR$_1$, we have that:

$$
\begin{aligned}
x_{k+1}^0 &= x_k^{n-1} - \gamma F_{\omega_{n-1}^k}(x_k^{n-1}) - \gamma \mathbb{U}_k \\
&= x_k^0 - \gamma \sum_{i=0}^{n-1} F_{\omega_k^i}(x_k^i) - \gamma \mathbb{U}_k \\
&= x_k^0 - \gamma n F(x_k^0) - \gamma \sum_{i=0}^{n-1}\left(F_{\omega_k^i}(x_k^i) - F_{\omega_k^i}(x_k)\right) - \gamma \mathbb{U}_k
\end{aligned}
\tag{14}
$$

where the last step we used the fact that $\gamma n F(x_k^0) = \gamma \sum_{i=0}^{n-1} F_{\omega_{i-1}^k}(x_k^0)$ and the finite-sum structure of the operator $F$. It holds, thus, that

$$
\left\|x_{k+1}^0 - x^*\right\|^2 = \left\|x_k^0 - x^* - \gamma n F(x_k^0) - \gamma \sum_{i=0}^{n-1}(F_{\omega_k^i}(x_k^i) - F_{\omega_k^k}(x_k^0)) - \gamma \mathbb{U}_k\right\|^2
\tag{15}
$$

From Young's inequality, the right-hand side (RHS) of (15) can be bounded as follows

$$
\begin{aligned}
\left\|x_{k+1}^0 - x^*\right\|^2 &\leq \frac{\left\|x_k^0 - x^* - \gamma n F(x_k^0)\right\|^2}{1 - \gamma n\mu} + \frac{\gamma}{n\mu}\left\|\sum_{i=0}^{n-1}(F_{\omega_k^i}(x_k^i) - F_{\omega_k^k}(x_k^0)) + \mathbb{U}_k\right\|^2 \\
&\overset{(2)}{\leq} \frac{\left\|x_k^0 - x^* - \gamma n F(x_k^0)\right\|^2}{1 - \gamma n\mu} + \frac{2\gamma}{n\mu}\left\|\sum_{i=0}^{n-1} F_{\omega_k^i}(x_k^i) - F_{\omega_k^i}(x_k^0)\right\|^2 + \frac{2\gamma}{n\mu}\|\mathbb{U}_k\|^2 \\
&\overset{(2)}{\leq} \frac{\left\|x_k^0 - x^* - \gamma n F(x_k^0)\right\|^2}{1 - \gamma n\mu} + \frac{2\gamma}{\mu}\sum_{i=0}^{n-1}\left\|F_{\omega_k^i}(x_k^i) - F_{\omega_k^i}(x_k^0)\right\|^2 + \frac{2\gamma}{n\mu}\|\mathbb{U}_k\|^2
\end{aligned}
$$

Applying the Lipschitz property of the operators, we obtain

$$\left\|x_{k+1}^0 - x^*\right\|^2 \leq \frac{\left\|x_k^0 - x^* - \gamma n F(x_k^0)\right\|^2}{1 - \gamma n \mu} + \frac{2\gamma L_{max}^2}{\mu} \sum_{i=1}^{n-1} \left\|x_k^i - x_k^0\right\|^2 + \frac{2\gamma}{n\mu} \left\|\mathbb{U}_k\right\|^2 \quad (16)$$

Taking expectation condition on the filtration $\mathcal{F}_k$ (history of $x_k^0$) and using the fact that the noise $\mathbb{U}_k \sim \mathcal{N}\left(0, \gamma^2 n^2 \sigma_*^2 \mathbb{I}\right)$, we get

$$\mathbb{E}\left[\left\|x_{k+1}^0 - x^*\right\|^2 \mid \mathcal{F}_k\right] \leq \frac{\mathbb{E}\left[\left\|x_k^0 - x^* - \gamma n F(x_k^0)\right\|^2 \mid \mathcal{F}_k\right]}{1 - \gamma n \mu} + \frac{2\gamma L_{max}^2}{\mu} \mathbb{E}\left[\sum_{i=1}^{n-1} \left\|x_k^i - x_k^0\right\|^2 \mid \mathcal{F}_k\right]$$
$$+ \frac{2n\gamma^3 \sigma_*^2}{\mu} \quad (17)$$

To complete the proof, it suffices to bound each term on the right-hand side of (17). From Lemmas C.2, C.3, it holds for $\gamma \leq \frac{1}{\sqrt{3n(n-1)}L_{max}}$ that

$$\mathbb{E}\left[\left\|x_k^0 - x^* - \gamma n F(x_k^0)\right\|^2 \mid \mathcal{F}_k\right] \leq \left[(1 - \gamma n \mu)^2 + \gamma^2 n^2 L_{max}^2\right] \left\|x_k - x^*\right\|^2 + 2\gamma n \lambda \quad (18)$$

$$\mathbb{E}\left[\sum_{i=1}^{n-1} \left\|x_k^i - x_k^0\right\|^2 \mid \mathcal{F}_k\right] \leq 4\gamma^2 n^3 L^2 \left\|x_k^0 - x^*\right\|^2 \quad (19)$$

Substituting (18) and (19) into (17), we obtain

$$\mathbb{E}\left[\left\|x_{k+1}^0 - x^*\right\|^2 \Big| \mathcal{F}_k\right] \leq \left(1 - \gamma n \mu + \frac{\gamma^2 n^2 L_{max}^2}{1 - \gamma n \mu} + \frac{8n^3 \gamma^3 L^2 L_{max}^2}{\mu}\right) \left\|x_k^0 - x^*\right\|^2$$
$$+ \frac{4n^2 \gamma^3 L_{max}^2}{\mu} \sigma_*^2 + \frac{2n\gamma\lambda}{1 - \gamma n \mu} \quad (20)$$

Selecting the stepsize $\gamma \leq \min\left\{\frac{1}{2n\mu}, \frac{\sqrt{1 + 6L_{max}^2 \mu^2} - 1}{12n L_{max}^2}\right\}$, we have that

$$\frac{1}{1 - \gamma n \mu} \leq 2$$

$$\text{and } \left(1 - \gamma n \mu + \frac{\gamma^2 n^2 L_{max}^2}{1 - \gamma n \mu} + \frac{12n^3 \gamma^3 L_{max}^4}{\mu}\right) \leq \left(1 - \frac{\gamma n \mu}{2}\right)$$

and thus substituting in (20) we get

$$\mathbb{E}\left[\left\|x_{k+1}^0 - x^*\right\|^2 \Big| \mathcal{F}_k\right] \leq \left(1 - \frac{\gamma n \mu}{2}\right) \left\|x_k^0 - x^*\right\|^2 + \frac{4n^2 \gamma^3 L_{max}^2}{\mu} \sigma_*^2 + 4n\gamma\lambda \quad (21)$$

Taking expectation on both sides and using the tower property of expectations, we have that:

$$\mathbb{E}\left[\left\|x_{k+1}^0 - x^*\right\|^2\right] \leq \left(1 - \frac{\gamma n \mu}{2}\right) \left\|x_k^0 - x^*\right\|^2 + \frac{4n^2 \gamma^3 L_{max}^2}{\mu} \sigma_*^2 + 8n\gamma\lambda$$
$$\leq \left(1 - \frac{\gamma n \mu}{2}\right)^{k+1} \left\|x_k^0 - x^*\right\|^2 + \sum_{i=1}^{k} (1 - \gamma n \mu)^i \left(\frac{4n^2 \gamma^3 L_{max}^2}{\mu} \sigma_*^2 + 8n\gamma\lambda\right)$$
$$\leq \left(1 - \frac{\gamma n \mu}{2}\right)^{k+1} \left\|x_0^0 - x^*\right\|^2 + \frac{8n\gamma^2 L_{max}^2}{\mu^2} \sigma_*^2 + \frac{8\lambda}{\mu}$$

$$\square$$

# D   PROOF OF ERGODIC PROPERTIES AND LIMIT THEOREMS (THEOREM 4.3) & (THEOREM 4.4)

We start by proving a series of properties that the induced Markov Chain satisfies, that will be necessary for proving the Theorem 4.3.

> **Proposition D.1** (Proposition 5.5.3 (Meyn & Tweedie, 2012)). *If a set $C \in \mathcal{B}(\mathbb{R}^d)$ is $\nu_m$-small, then it is $\nu_{\delta_m}$-petite for some $\delta_m > 0$.*

*Intuition.* The notions of *small* and *petite* sets are technical tools in Markov chain theory that help verify stability properties. A set $C$ is called *small* if, starting from $C$, the chain has a uniform positive chance of reaching any region of the state space within a fixed number of steps. A *petite* set is a weaker concept: instead of requiring such uniformity in a single time step, it allows the chance of hitting any region to be distributed over a random number of steps (via a probability distribution over times). Thus, every small set is automatically petite, but the reverse is not true. Intuitively, small sets guarantee "uniform mixing after a fixed horizon," while petite sets guarantee the same effect "on average over time."

## D.1   PROOF OF CONTINUOUS STATE TIME HOMOGENIOUS MARKOV CHAIN (LEMMA 4.2)

> **Lemma D.2** (Epoch-level homogeneity and kernel). Fix $\gamma > 0$ and $n \in \mathbb{N}$. Then Perturbed-SGD can be described at each epoch $k$ as: *Draw $\omega_k$ uniformly from $\mathfrak{S}_n$ and set*
> $$x_{k+1} \;=\; H(x_k, \omega_k) \;+\; U_k, \quad U_k \sim \mathcal{N}(0, \Sigma),$$
> where $H(x, \omega)$ denotes the endpoint of one reshuffled pass started at $x$ with permutation $\omega$ (i.e., the map induced by $n$ inner updates with step size $\gamma$). Then $(x_k)_{k \geq 0}$ is a time-homogeneous Markov chain on $\mathbb{R}^d$ with transition kernel
> $$P(x, A) \;=\; \frac{1}{n!} \sum_{\omega \in \mathfrak{S}_n} \int_A \phi\big(y;\, H(x, \omega),\, \Sigma\big)\, dy, \qquad A \in \mathcal{B}(\mathbb{R}^d),$$
> where $\phi(\cdot; m, \Sigma)$ is the $d$-variate Gaussian density with mean $m$ and covariance $\Sigma$.

*Proof.* Fix $\gamma > 0$ and $n \in \mathbb{N}$. For any $x \in \mathbb{R}^d$ and $\omega \in \mathfrak{S}_n$, define the inner-epoch recursion
$$x^{[0]}(x, \omega) = x, \qquad x^{[j+1]}(x, \omega) = x^{[j]}(x, \omega) - \gamma\, F_{\omega[j]}\big(x^{[j]}(x, \omega)\big), \;\; j = 0, \ldots, n-1,$$
and the (measurable) epoch map
$$H(x, \omega) := x - \gamma \sum_{j=0}^{n-1} F_{\omega[j]}\big(x^{[j]}(x, \omega)\big).$$

By construction, at epoch $k$ the algorithm updates as
$$x_{k+1} = H(x_k, \omega_k) + U_k,$$
where $(\omega_k)_{k \geq 0}$ are i.i.d. uniform on $\mathfrak{S}_n$ and $(U_k)_{k \geq 0}$ are i.i.d. with law $\mathcal{N}(0, \Sigma I_d)$, independent of $(\omega_k)_{k \geq 0}$ and of $x_k$ given the present state.

*Markov property.* Let $A \in \mathcal{B}(\mathbb{R}^d)$. Using the tower property and the independence of $\omega_k, U_k$ from the past given $x_k$,
$$\Pr(x_{k+1} \in A \mid x_0, \ldots, x_k) = \mathbb{E}\Big[\Pr(H(x_k, \omega_k) + U_k \in A \mid x_k, \omega_k) \,\Big|\, x_k\Big] = \mathbb{E}[\Pr(H(x_k, \omega) + U \in A)],$$
where the outer expectation is over $\omega \sim \mathrm{Unif}(\mathfrak{S}_n)$ and $U \sim \mathcal{N}(0, \Sigma I_d)$, independent. Thus
$$\Pr(x_{k+1} \in A \mid x_0, \ldots, x_k) = \Pr(x_{k+1} \in A \mid x_k) =: P(x_k, A),$$
so $(x_k)_{k \geq 0}$ is a Markov chain.

*Time-homogeneity and kernel.* Since the joint law of $(\omega_k, U_k)$ does not depend on $k$, the transition kernel $P$ is time-invariant. By conditioning on $\omega$ and integrating over the Gaussian $U$,

$$P(x, A) = \frac{1}{n!} \sum_{\omega \in \mathfrak{S}_n} \Pr(H(x, \omega) + U \in A) = \frac{1}{n!} \sum_{\omega \in \mathfrak{S}_n} \int_A \phi(y;\, H(x, \omega),\, \Sigma I_d)\, dy,$$

where $\phi(\cdot;\, m, \Sigma I_d)$ denotes the $d$-variate Gaussian density with mean $m$ and covariance $\Sigma I_d$. This yields the stated expression for $P$ and establishes time-homogeneity on $\mathbb{R}^d$.

*Augmented formulation (for reference).* On the product space $\mathbb{R}^d \times \mathfrak{S}_n$, define the next permutation $\omega' \sim \mathrm{Unif}(\mathfrak{S}_n)$ independently of $(x, \omega)$ and $U$. Then the augmented chain $((x_k, \omega_k))_{k \geq 0}$ satisfies

$$(x', \omega') = \big(H(x, \omega) + U,\ \omega'\big),$$

and the associated kernel is

$$K\big((x, \omega), A \times B\big) = \int_A \phi(y;\, H(x, \omega),\, \Sigma I_d)\, dy \cdot \frac{|B|}{n!},$$

which is manifestly time-homogeneous. $\qquad\square$

We, next, show that there exists an energy function that describes the iterates of the Markov chain.

## D.2 PROOF OF FOSTER-LYUAPUNOV INEQUALITY

A central tool for proving stability and ergodicity of Markov chains is the *Foster–Lyapunov inequality*. The idea is to construct an "energy" or "Lyapunov" function $\mathcal{E}(x, x^*)$ that tracks the distance of the chain's state from equilibrium. If this function decreases on average outside a bounded region, it ensures that the process cannot drift to infinity and will instead return frequently to a compact set. This property, when combined with the minorization condition, implies positive Harris recurrence and geometric ergodicity (Meyn & Tweedie, 2012).

In our case, a natural candidate for such an energy is the squared distance to a solution $x^*$, up to an additive constant. The following corollary verifies that this choice indeed satisfies a Foster–Lyapunov inequality for Perturbed SGD–$\mathrm{RR}_1$, showing that the expected energy after one epoch contracts linearly up to a fixed additive term.

**Corollary D.3.** Let Assumptions 2.1-2.3 hold. The function $\mathcal{E}(x_0^k, x^*) = \|x_0^k - x^*\|_2^2 + 1$ satisfies for any $x^* \in \mathcal{X}^*$ the inequality

$$\mathbb{E}\Big[\mathcal{E}(x_0^{k+1}, x^*) \,|\, \mathcal{F}_k\Big] \quad \leq \quad c_1 \mathcal{E}(x_0^k, x^*) + c_2,$$

where $c_1 = 1 - \frac{\gamma n \mu}{2}$ and $c_2 = \frac{\gamma n \mu}{2} + \frac{8 n \gamma^2 L_{max}^2}{\mu^2} \sigma_*^2 + \frac{8\lambda}{\mu}$.

*Proof.* From inequality (21) of Theorem 4.1, we have that

$$\mathbb{E}\Big[\|x_{k+1}^0 - x^*\|^2 \,|\, \mathcal{F}_k\Big] \quad \leq \quad \Big(1 - \frac{\gamma n \mu}{2}\Big)^{k+1} \|x_0^0 - x^*\|^2 + \frac{8 n \gamma^2 L_{max}^2}{\mu^2} \sigma_*^2 + \frac{8\lambda}{\mu}$$

Adding in both sides one and using the definition of $\mathcal{E}(x_0^k, x^*)$, we obtain

$$\mathbb{E}\Big[\|x_{k+1}^0 - x^*\|^2 + 1 \,|\, \mathcal{F}_k\Big] \quad \leq \quad \Big(1 - \frac{\gamma n \mu}{2}\Big)\big(\|x_k - x^*\|^2 + 1\big) + \frac{\gamma n \mu}{2} + \frac{8 n \gamma^2 L_{max}^2}{\mu^2} \sigma_*^2 + \frac{8\lambda}{\mu}$$

$$\iff \mathbb{E}\Big[\mathcal{E}(x_0^{k+1}, x^*) \,|\, \mathcal{F}_k\Big] \quad \leq \quad c_1 \mathcal{E}(x_0^k, x^*) + c_2 \qquad\qquad (22)$$

where at the last step we have let $c_1 = 1 - \frac{\gamma n \mu}{2}$ and $c_2 = \frac{\gamma n \mu}{2} + \frac{8 n \gamma^2 L_{max}^2}{\mu^2} \sigma_*^2 + \frac{8\lambda}{\mu}$. $\qquad\square$

**Lemma D.4.** Let Assumptions 2.1-2.3 hold. If $\gamma \leq \gamma_{max}$, then for any fixed $x^* \in \mathcal{X}^*$ the functions $\mathcal{E}_1(x, x^*) = \mathcal{E}(x, x^*), \mathcal{E}_2(x, x^*) = \sqrt{\mathcal{E}(x, x^*)}$ satisfy the geometric drift property for the iterates of Perturbed SGD–RR$_1$, i.e., $\forall i \in \{1, 2\}$ there exist measurable set $C_i$, constants $\alpha_i > 0, \tilde{\alpha}_i < \infty$ such that $\forall x \in \mathbb{R}^d$

$$\Delta \mathcal{E}_i(x, x^*) = -\alpha \mathcal{E}_i(x, x^*) + \mathbb{1}_C \, \tilde{\alpha}, \tag{23}$$

where $\Delta \mathcal{E}_i(x, x^*) = \int_{x' \in \mathbb{R}^d} P(z, dx') \mathcal{E}_i(x') - \mathcal{E}_i(x)$ and the constant $\gamma_{max} = \min\left\{\frac{1}{3nL_{max}}, \frac{\sqrt{1+6\mu^2 L_{max}^2}-1}{12nL_{max}^2}\right\}$.

*Proof.* In order to prove that the geometric drift property is satisfied, we need to show that there exist function $\mathcal{E}_i : \mathbb{R}^d \to [1, +\infty]$, measurable set $C_i$ and constants $\alpha_i > 0, \tilde{\alpha}_i < \infty$ such that (23) holds. From Corollary D.3, we have that the function $\mathcal{E}_1 : \mathbb{R}^d \to [1, +\infty]$ with $\mathcal{E}_1(x, x^*) = \|x - x^*\|^2 + 1$ satisfies along the iterates of Perturbed SGD–RR$_1$ that

$$\mathbb{E}\left[\mathcal{E}_1(x_{k+1}, x^*) \,|\, \mathcal{F}_k : \{x_k = x\}\right] \leq c_1 \mathcal{E}_1(x, x^*) + c_2, \tag{24}$$

where $c_1 = 1 - \frac{\gamma n \mu}{2}$ and $c_2 = \frac{\gamma n \mu}{2} + \frac{8n\gamma^2 L_{max}^2}{\mu^2}\sigma_*^2 + \frac{8\lambda}{\mu}$. Additionally, for the epoch-level iterates $x_k$ of Perturbed SGD–RR$_1$ the definition of $\Delta \mathcal{E}$ is

$$\Delta \mathcal{E}_1(x, x^*) = \int_{x' \in \mathbb{R}^d} P(x, dx') \mathcal{E}_1(x', x^*) - \mathcal{E}_1(x, x^*)$$
$$= \mathbb{E}\left[\mathcal{E}_1(x_{k+1}, x^*) - \mathcal{E}_1(x_k, x^*) \,|\, \mathcal{F}_k : \{x_k = x\}\right]. \tag{25}$$

From (24) and (25), we have that

$$\mathbb{E}\left[\mathcal{E}_1(x_{k+1}, x^*) \,|\, \mathcal{F}_k : \{x_k = x\}\right] \leq c_1 \mathcal{E}_1(x) + c_2$$
$$\Rightarrow \quad \mathbb{E}\left[\mathcal{E}_1(x_{k+1}, x^*) - \mathcal{E}_1(x_k, x^*) \,|\, \mathcal{F}_k : \{x_k = x\}\right] \leq -(1 - c_1)\mathcal{E}_1(x, x^*) + c_2$$
$$\Rightarrow \quad \Delta \mathcal{E}_1(x, x^*) \leq -(1 - c_1)\mathcal{E}_1(x, x^*) + c_2 \tag{26}$$

Let $C_1 = \left\{x \in \mathbb{R}^d : \mathcal{E}_1(x, x^*) \leq \frac{2c_2}{(1-c_1)}\right\}$. We have that

$$\Delta \mathcal{E}_1(x, x^*) \leq -(1 - c_1)\mathcal{E}_1(x, x^*) + \mathbb{1}_C(x)c_2 + \mathbb{1}_{C^c}(x)\frac{1 - c_1}{2}\mathcal{E}_1(x, x^*)$$
$$\leq -\frac{1 - c_1}{2}\mathcal{E}_1(x, x^*) + \mathbb{1}_{C_1(x)} c_2 \tag{27}$$

where at the last step we used the fact that $\mathbb{1}_{C_1^c}(x) < 1$ and $c_1 \in (0, 1)$. From (27) we conclude that $\mathcal{E}_1(x, x^*)$ satisfies the geometric drift property for the set $C_1 = \left\{x \in \mathbb{R}^d : \mathcal{E}_1(x) \leq \frac{2c_2}{(1-c_1)}\right\}$ and with constants $\alpha = \frac{1-c_1}{2}, a = c_2$.

For the $\mathcal{E}_2(x, x^*) = \sqrt{\mathcal{E}(x, x^*)}$, by Jensen's inequality it holds that

$$\mathbb{E}\left[\sqrt{\mathcal{E}(x_{k+1}, x^*)} \,|\, \mathcal{F}_k : \{x_k = x\}\right] \leq \sqrt{\mathbb{E}\left[\mathcal{E}(x_{k+1}, x^*) \,|\, \mathcal{F}_k : \{x_k = x\}\right]}$$
$$\leq \sqrt{c_1 \mathcal{E}(x, x^*) + c_2}$$
$$\leq \sqrt{c_1}\sqrt{\mathcal{E}(x, x^*)} + \sqrt{c_2}$$

Thus, there exist constants $d_1 = \sqrt{c_1}, d_2 = \sqrt{c_2}$ such that it holds

$$\mathbb{E}\left[\mathcal{E}_2(x_{k+1}, x^*) \,|\, \mathcal{F}_k : \{x_k = x\}\right] \leq d_1 \mathcal{E}_2(x, x^*) + d_2, \tag{28}$$

Since it holds that

$$\Delta \mathcal{E}_2(x, x^*) = \int_{x' \in \mathbb{R}^d} P(x, dx') \mathcal{E}_2(x', x^*) - \mathcal{E}_2(x, x^*) = \mathbb{E}\left[\mathcal{E}_2(x_{k+1}, x^*) - \mathcal{E}_2(x_k, x^*) \,|\, \mathcal{F}_k : \{x_k = x\}\right], \tag{29}$$

we have that

$$\mathbb{E}\Big[\mathcal{E}_2(x_{k+1}, x^*) - \mathcal{E}_2(x_k, x^*) \,|\, \mathcal{F}_k : \{x_k = x\}\Big] \leq -(1 - d_1)\mathcal{E}_2(x, x^*) + d_2$$

$$\Rightarrow \quad \Delta\mathcal{E}_2(x, x^*) \leq -(1 - d_1)\mathcal{E}_2(x, x^*) + d_2 \tag{30}$$

Let $C_2 = \left\{ x \in \mathbb{R}^d : \mathcal{E}_2(x, x^*) \leq \frac{2d_2}{(1-d_1)} \right\}$. We have that

$$\begin{aligned}
\Delta\mathcal{E}_2(x, x^*) &\leq -(1 - d_1)\mathcal{E}_2(x, x^*) + \mathbb{1}_{C_2(x)}\, d_2 + \mathbb{1}_{C_2^c}(x)\frac{1 - d_1}{2}\mathcal{E}_2(x, x^*) \\
&\leq -\frac{1 - d_1}{2}\mathcal{E}_2(x, x^*) + \mathbb{1}_{C_2(x)}\, d_2
\end{aligned}$$

where at the last step we used the fact that $\mathbb{1}_{C_2^c}(x) < 1$ and $d_1 \in (0, 1)$. Hence, we conclude that $\mathcal{E}_2(x, x^*)$ satisfies the geometric drift property for the set $C_2 = \left\{ x \in \mathbb{R}^d : \mathcal{E}_2(x) \leq \frac{2d_2}{(1-d_1)} \right\}$ and with constants $\alpha_2 = \frac{1-d_1}{2}, \tilde{\alpha} = d_2$. $\qquad \square$

### D.3 Proof of Minorization Property

The next step in establishing ergodicity is to verify a *minorization condition*. Intuitively, this property guarantees that whenever the chain is in a certain "small set" $C$, its one-step transition kernel dominates a fixed nontrivial distribution $\nu$, uniformly with probability $\delta > 0$. In other words, starting from any $x \in C$, the algorithm has a baseline chance of moving into any region of the state space according to $\nu$. This is the key ingredient that, together with a Lyapunov–Foster drift condition, yields geometric ergodicity of the Markov chain. The following lemma formalizes this property for the iterates of Perturbed SGD–RR$_1$.

**Lemma D.5** (Minorization property). Let Assumptions 2.1–2.3 hold. If $\gamma \leq \gamma_{\max}$, then the iterates of `Perturbed SGD--RR`$_1$ satisfy the minorization condition: there exist a constant $\delta > 0$, a probability measure $\nu$ on $(\mathbb{R}^d, \mathcal{B}(\mathbb{R}^d))$, and a set $C \subseteq \mathbb{R}^d$ such that $\nu(C) = 1, \nu(C^\complement) = 0$, and

$$P(x, A) \geq \delta\, \mathbb{1}_C(x)\, \nu(A), \qquad \forall x \in \mathbb{R}^d, \ A \in \mathcal{B}(\mathbb{R}^d), \tag{31}$$

where $P(x, A) = \Pr(x_{k+1} \in A \mid x_k = x)$ and $\gamma_{\max} = \min\left\{ \frac{1}{3nL_{max}}, \frac{\sqrt{1+6\mu^2 L_{max}^2}-1}{12nL_{max}^2} \right\}$.

*Proof.* Consider the Lyapunov candidate $\mathcal{E}(x) = \|x - x^*\|^2 + 1$ for some $x^* \in \mathcal{X}^*$. Its sublevel sets

$$C(r) := \{x \in \mathbb{R}^d : \mathcal{E}(x) \leq r\} = \mathbb{B}(x^*, \sqrt{r - 1}), \qquad r > 1,$$

are bounded, hence suitable for applying small/petite set arguments.

At each epoch, the update of Perturbed SGD-RR$_1$ can be described by

$$x_{k+1} = H(x_k, \omega_k) + U_k,$$

where $\omega_k$ is uniform on $\mathfrak{S}_n$ and $U_k \sim \mathcal{N}(0, \Sigma I_d)$, independent of $\omega_k$ and $x_k$. Thus, for any $A \in \mathcal{B}(\mathbb{R}^d)$,

$$P(x, A) = \frac{1}{n!} \sum_{\omega \in \mathfrak{S}_n} \int_A \phi\big(y; H(x, \omega), \Sigma\big)\, dy,$$

Since $\phi(y; m, \Sigma I_d) > 0$ for all $y \in \mathbb{R}^d$, the kernel has strictly positive support everywhere.

Now fix $r_0 > 1$ and restrict to $C(r_0)$. Define the reference measures for any $A \in \mathcal{B}(\mathbb{R}^d)$

$$\nu(A) := \frac{\text{Leb}(A \cap C(r_0))}{\text{Leb}(C(r_0))} \text{ and } \text{Leb}(A) = \int_A \inf_{x \in C(r_0)} \frac{1}{n!} \sum_{\omega \in \mathfrak{S}_n} \phi\big(y; H(x, \omega), \Sigma\big)\, dy.$$

i.e., the uniform probability distribution over $C(r_0)$. Clearly $\nu(C(r_0)) = 1$ and $\nu(C(r_0)^\complement) = 0$.

Finally, by continuity of $\phi$ and compactness of $C(r_0)$, there exists $\delta \geq \text{Leb}(C(r_0)) > 0$ such that

$$P(x, A) \geq \delta\, \nu(A), \qquad \forall x \in C(r_0), \ A \subseteq C(r_0).$$

If $x \notin C(r_0)$ or $A \not\subseteq C(r_0)$, the right-hand side of (31) is zero and the inequality is trivially satisfied. Hence the minorization condition (31) holds. $\qquad \square$

## D.4 PROOF OF IRREDUCIBILITY, APERIODICITY AND HARRIS AND POSITIVE RECURRENCE

**Lemma D.6.** The Markov chain $(x_k)_{k\geq 0}$ of Perturbed SGD–RR$_1$ is

1. $\psi-$irreducible for some non-zero $\sigma$-finite measure $\psi$ on $\mathbb{R}^d$ over the Borel $\sigma$-algebra of $\mathbb{R}^d$.
2. strongly aperiodic.
3. Harris and positive recurrent with an invariant measure.

*Proof.* We prove each of the three properties in turn.

**Irreducibility.** From Lemma D.5, the Markov kernel of `Perturbed SGD--RR`$_1$ is

$$P(x, A) = \frac{1}{n!} \sum_{\omega \in \mathfrak{S}_n} \int_A \phi(y;\, H(x, \omega),\, \Sigma I_d)\ dy, \qquad A \in \mathcal{B}(\mathbb{R}^d),$$

where $\phi(\cdot; m, \Sigma I_d)$ is a Gaussian density with strictly positive support. Hence, for any measurable set $A$ of positive Lebesgue measure, $P(x, A) > 0$. Taking $\psi$ to be the Lebesgue measure, we conclude that the chain is $\psi$-irreducible.

**Strong Aperiodicity.** By Lemma D.5, there exist $\delta > 0$, a probability measure $\nu$, and a set $C \subseteq \mathbb{R}^d$ such that

$$P(x, A) \geq \delta\, \mathbb{1}_C(x)\nu(A), \qquad \forall x \in \mathbb{R}^d,\ A \in \mathcal{B}(\mathbb{R}^d).$$

Since $C$ has positive Lebesgue measure and $\nu(C) = 1$, $\nu(C^\circ) = 0$ and given that the sets $C(r)$ in the proof of the Lemma D.5 are small and of positive measure, we get that the Markov chain is strongly aperiodic.

**Harris and Positive Recurrence.** By Proposition D.1, the small set $C$ of Lemma D.5 is also petite. Combined with the Foster–Lyapunov drift condition of Lemma D.4, the Geometric Ergodic Theorem (Theorem 15.0.1 in (Meyn & Tweedie, 2012)) guarantees that the chain is positive recurrent and admits an invariant probability measure.

Finally, from Theorem 9.1.8 of (Meyn & Tweedie, 2012), the existence of a Lyapunov function unbounded off petite sets, satisfying $\Delta\mathcal{E} \leq 0$ together with $\psi$-irreducibility, implies Harris recurrence. $\qquad\square$

## D.5 PROOF OF EXISTENCE OF UNIQUE INVARIANT DISTRIBUTION AT EPOCH-LEVEL (THEOREM 4.3)

By verifying irreducibility, aperiodicity, and *positive Harris recurrence* (Meyn & Tweedie, 2012), we establish a unique invariant distribution $\pi_\gamma$, geometric convergence in total variation to it, and concentration of scalar observables (admissible test functions) around $x^*$.

**Theorem D.7** (Restatement of Theorem 4.3). Under Assumptions 2.1–2.3, run *Perturbed SGD-RR$_1$* with $\gamma \leq \gamma_{\max}$. Then $(x_k)_{k\geq 0}$ admits a unique stationary distribution $\pi_\gamma \in \mathcal{P}_2(\mathbb{R}^d)$, and additionally:

$$\begin{aligned}
&\text{(i)} \quad |\,\mathbb{E}[\ell(x_k)] - \mathbb{E}_{x\sim\pi_\gamma}[\ell(x)]\,| \ \leq\ c(1-\rho)^k && \forall \ell : |\ell(x)| \leq L_\ell(1 + \|x\|), \\
&\text{(ii)} \quad |\,\mathbb{E}_{x\sim\pi_\gamma}[\ell(x)] - \ell(x^*)\,| \ \leq\ L_\ell\sqrt{C} && \forall \ell : L_\ell - \text{Lipschitz functions},
\end{aligned}$$

for some $c < \infty$, $\rho \in (0,1)$, $C = \Theta(\mathrm{MSE}(\texttt{SGD}-\text{RR}_1))$ and $\gamma_{\max}$ defined in Theorem 4.1

*Proof.* From Lemma D.6, we have that the underlying Markov Chain has an invariant probability measure. Since from Lemma D.4 the induced Markov Chain satisfies the geometric drift property, according to the Strong Ergodic Theorem (Meyn & Tweedie, 2012) we conclude that the measure is finite and unique. From the invariant property of $\pi_\gamma$, we have that for $x_0 \sim \pi_\gamma$ the iterates satisfy also that $(x_k)_{k>0} \sim \pi_\gamma$. From Corollary D.3, we have that for an arbitrary fixed $x^*$ the iterates of Perturbed SGD–RR$_1$ with step size $\gamma \leq$ satisfy for $c_1 \in (0,1), c_2 > 0$ that

$$\mathbb{E}\Big[\|x_{k+1} - x^*\|_2^2 + 1 \,|\, \mathcal{F}_k\Big] \ \leq\ c_1\left(\|x_k - x^*\|_2^2 + 1\right) + c_2.$$

Taking expectation with respect to the invariant measure $\pi_\gamma$ and using the tower law of expectation, we get

$$\mathbb{E}_{x \sim \pi_\gamma}\left[\|x - x^*\|_2^2\right] \leq \frac{c_1 + c_2 - 1}{1 - c_1} = \mathcal{O}\left(\frac{\max(\gamma, \lambda)}{\mu}\right) < +\infty. \tag{32}$$

Combining the above inequality with the fact that $\|x_*\| \leq R$ by Assumption 2.1, we conclude that the invariant measure $\pi_\gamma \in \mathcal{P}_2(\mathbb{R}^d)$, where $\mathcal{P}_2(\mathbb{R}^d)$ is the set of distributions supported in $\mathbb{R}^d$ with finite second moment.

We, next, proceed with proving the second statement of the Theorem. By assumption, we have that the test function satisfies $\forall x \in \mathbb{R}^d_{\geq 0}$ that

$$
\begin{aligned}
|\ell(x)| &\leq L_\ell(1 + \|x\|) \\
&\leq L_\ell(1 + \|x^*\| + \|x - x^*\|) \\
&\leq L_\ell(1 + R + \|x - x^*\|) \\
&\leq (1 + R)L_\ell(1 + \|x - x^*\|) \tag{33}
\end{aligned}
$$

where we have used the triangle inequality and the fact that $\|x^*\| \leq R$. Applying Cauchy-Schwarz inequality, we can further upper bound $\|\ell(x)\|$

$$
\begin{aligned}
|\ell(x)| &\leq \sqrt{2}(1 + R)L_\ell\sqrt{1 + \|x - x^*\|} \\
&\leq \max\left(1, \sqrt{2}(1 + R)L_\ell\right)\sqrt{\mathcal{E}(x, x^*)} \tag{34}
\end{aligned}
$$

Letting $c = \max\left(1, \sqrt{2}(1 + R)L_\ell\right)$ and $\tilde{\mathcal{E}}(x, x^*) = c\sqrt{\mathcal{E}(x, x^*)}$, we have that

$$|\ell(x)| \leq \tilde{\mathcal{E}}(x, x^*)$$

From Lemma D.4 we have that $\mathcal{E}_1(x, x^*), \mathcal{E}_2(x, x^*)$ satisfy the geometric drift property and since $c \geq 1$ we have that $\tilde{\mathcal{E}}(x, x^*) = c\mathcal{E}_2(x, x^*)$ satisfies also the geometric drift property. According to Theorem 16.0.1 in (Meyn & Tweedie, 2012) Perturbed SGD–$\text{RR}_1$ is $\tilde{\mathcal{E}}$-uniformly ergodic and there exists $\rho \in (0, 1)$ and $R \in (0, +\infty)$ such that

$$\left|P^k\ell(x_0) - \mathbb{E}_{x \sim \pi_\gamma}\left[\ell(x)\right]\right| \leq R(1 - \rho)^k\left|\tilde{\mathcal{E}}(x_0, x^*)\right| \tag{35}$$

Letting $c = R\left|\tilde{\mathcal{E}}(x_0, x^*)\right|$, we have proven the inequality in the statement of the theorem. In order to show that the epoch-level iterates converge under the total variation distance it suffices to consider only functions $\ell : \mathbb{R}^d \to \mathbb{R}$ that are bounded by 1. In this case, there are constants $\tilde{\rho} \in (0, 1)$ and $\tilde{R} \in (0, +\infty)$ independent of $\ell$ such that it holds

$$\sup_{|\ell| \leq 1}\left|P^k\ell(x_0) - \mathbb{E}_{x \sim \pi_\gamma}\left[\ell(x)\right]\right| \leq \tilde{R}(1 - \tilde{\rho})^k\left|\tilde{\mathcal{E}}(x_0, x^*)\right|$$

implying according to the dual representation of Radon metric for bounded initial conditions (Wikipedia, Accessed: 2025-08-28) the geometric convergence under the total variation distance.

In order to prove the third statement of the theorem, we apply linearity of expectation and the Lipschitz property of the test function $\ell$ and obtain

$$
\begin{aligned}
\left|\mathbb{E}_{x \sim \pi_\gamma}\left[\ell(x)\right] - \ell(x^*)\right| &\leq \mathbb{E}_{x \sim \pi_\gamma}\left[|\ell(x) - \ell(x^*)|\right] \\
&\leq \mathbb{E}_{x \sim \pi_\gamma}\left[L_\ell\|x - x^*\|\right]
\end{aligned}
$$

Applying Cauchy-Schwarz inequality and using inequality (32), we obtain that

$$\left|\mathbb{E}_{x \sim \pi_\gamma}\left[\ell(x)\right] - \ell(x^*)\right| \leq L_\ell\sqrt{\mathbb{E}_{x \sim \pi_\gamma}\left[\|x - x^*\|\right]} \leq L_\ell\sqrt{D}$$

where $D \propto \frac{\max(\gamma, \lambda)}{\mu}$ according to (32). $\qquad\square$

We conclude with the establishment of a Law of Large Numbers (LLN) and the corresponding Centra limit Theorem (CLT) that describe the epoch-level iterates of Perturbed SGD–RR$_1$.

### D.6  PROOF OF LIMIT THEOREMS OF EPOCH-LEVEL ITERATES (THEOREM 4.3)

**Theorem D.8** (Restatement of Theorem 4.4). Suppose Assumptions 2.1–2.3 hold and run Perturbed SGD–RR$_1$ with $\gamma \leq \gamma_{\max}$, (cf. Theorem 4.1).

Let $\ell : \mathbb{R}^d \to \mathbb{R}$ be any test function such that $|\ell(x)| \leq L_\ell(1 + \|x\|^2)$ and $\mathbb{E}_{x \sim \pi_\gamma}[\ell(x)] < \infty$. Then for the epoch-level iterates, it holds that:

$$\underbrace{\frac{1}{T}\sum_{t=0}^{T-1} \ell(x_t) \xrightarrow{\text{a.s.}} \mathbb{E}_{x \sim \pi_\gamma}[\ell(x)]}_{\text{(LLN)}} \qquad \underbrace{T^{-1/2}\sum_{t=0}^{T-1}\Big(\ell(x_t) - \mathbb{E}_{x \sim \pi_\gamma}[\ell(x)]\Big) \xrightarrow{d} \mathcal{N}(0, \sigma_{\pi_\gamma}^2(\ell)),}_{\text{(CLT)}}$$

where $\sigma_{\pi_\gamma}^2(\ell) = \lim_{T \to \infty} \frac{1}{T}\mathbb{E}_{\pi_\gamma}[S_T^2]$ and $S_T^2 = \sum_{t=0}^{T-1}\big(\ell(x_t) - \mathbb{E}_{x \sim \pi_\gamma}[\ell(x)]\big)^2$.

*Proof.* We show that the Markov Chain induced by the epoch-level iterates of Perturbed SGD–RR$_1$ is Harris positive recurrent, it has an invariant measure and satisfies $\mathcal{E}$-uniform ergodicity, and hence by Theorem 17.0.1 in (Meyn & Tweedie, 2012) the stated Law of Large Numbers and Central Limit Theorem hold.

From Lemma D.6, we have that the Markov Chain is Harris positive recurrent with an invariant measure. It suffices, thus, to show that the chain is $\mathcal{E}$-uniform ergodic by proving that there exists a potential function $\mathcal{E}(\cdot)$ such that the chain satisfies the geometric drift property of Meyn & Tweedie (2012) and $\|\ell(x)\|^2 \leq \mathcal{E}(x)$. Let $\mathcal{E}(x, x^*) = \mathbb{E}\Big[\mathcal{E}(x_0^{k+1}, x^*) \,|\, \mathcal{F}_k\Big]$ for any fixed $x^* \in \mathcal{X}^*$. According to Lemma D.4, $\mathcal{E}(x, x^*)$ satisfies the geometric drift property. Additionally, since $\ell$ has a linear growth it holds that

$$\begin{aligned}
|\ell(x)|^2 &\leq& L_\ell^2(1 + \|x\|^2)^2 \\
&\leq& L_\ell^2(1 + \|x^*\| + \|x - x^*\|)^2 \\
&\leq& L_\ell^2(1 + R + \|x - x^*\|)^2 \\
&\leq& L_\ell^2(1 + R)^2(1 + \|x - x^*\|)^2
\end{aligned} \tag{36}$$

From Cauchy-Schwarz inequality, it holds that

$$\begin{aligned}
1 + \|x - x^*\| &\leq& \sqrt{2}\sqrt{1 + \|x - x^*\|^2} \\
\Rightarrow (1 + \|x - x^*\|)^2 &\leq& 2(1 + \|x - x^*\|^2) \\
\Rightarrow (1 + \|x - x^*\|)^2 &\leq& 2\mathcal{E}(x, x^*)
\end{aligned} \tag{37}$$

Thus, combining (37) and (36), we obtain

$$|\ell(x)|^2 \leq 2L_\ell^2(1 + R)^2\mathcal{E}(x, x^*) \tag{38}$$

Thus, $\mathcal{E}(x, x^*)$ satisfies the geometric drift property and it holds that $|\ell(x)|^2 \leq \mathcal{E}(x, x^*)$ and hence the chain is $\mathcal{E}$-uniform ergodic, completing the proof. $\square$

# E   BOUNDING 4TH-ORDER MOMENTS OF ERROR DISTANCE

## E.1   HIGHER ORDER VERSION OF PROPOSITION A.2 (EMMANOUILIDIS ET AL., 2024) - BOUND OF KYRTOSIS FOR LIPSCHTIZ OPERATORS

**Proposition E.1.** Let Assumption 2.3 hold. For any $x \in \mathbb{R}^d$ and any reference point $x^* \in \mathbb{R}^d$, it holds that

$$\frac{1}{n} \sum_{i=1}^{n} \|F_i(x) - F(x)\|^4 \leq 128 \left( \frac{1}{n} \sum_{i=1}^{n} L_i^4 \right) \|x - x^*\|^4 + 128\, \sigma_*^4,$$

where $\sigma_*^4 := \frac{1}{n} \sum_{i=1}^{n} \|F_i(x^*)\|^4$.

*Proof.* Define

$$\Delta_i := F_i(x) - F_i(x^*), \quad \Delta := \frac{1}{n} \sum_{j=1}^{n} \Delta_j = F(x) - F(x^*), \quad \xi_i := F_i(x^*) - F(x^*).$$

Then $F_i(x) - F(x) = (\Delta_i - \Delta) + \xi_i$. Using $(a+b)^4 \leq 8(a^4 + b^4)$, we have that

$$\|F_i(x) - F(x)\|^4 \leq 8\|\Delta_i - \Delta\|^4 + 8\|\xi_i\|^4.$$

Applying the same inequality once more to $\Delta_i - \Delta$, we obtain

$$\|\Delta_i - \Delta\|^4 \leq 8 \big( \|\Delta_i\|^4 + \|\Delta\|^4 \big).$$

Averaging over $i$ and using Jensen's inequality for the convex map $u \mapsto \|u\|^4$,

$$\frac{1}{n} \sum_{i=1}^{n} \|F_i(x) - F(x)\|^4 \leq 128 \left( \tfrac{1}{n} \sum_{i=1}^{n} \|\Delta_i\|^4 \right) + 8 \left( \tfrac{1}{n} \sum_{i=1}^{n} \|\xi_i\|^4 \right).$$

Moreover, $\|\xi_i\| = \|F_i(x^*) - F(x^*)\| \leq \|F_i(x^*)\| + \|F(x^*)\|$, hence by $(a+b)^4 \leq 8(a^4 + b^4)$ and Jensen,

$$\frac{1}{n} \sum_{i=1}^{n} \|\xi_i\|^4 \leq 16 \left( \tfrac{1}{n} \sum_{i=1}^{n} \|F_i(x^*)\|^4 \right) = 16\, \sigma_*^4.$$

By Lipschitz continuity, we have

$$\|\Delta_i\| = \|F_i(x) - F_i(x^*)\| \leq L_i \|x - x^*\| \quad \Rightarrow \quad \frac{1}{n} \sum_{i=1}^{n} \|\Delta_i\|^4 \leq \left( \tfrac{1}{n} \sum_{i=1}^{n} L_i^4 \right) \|x - x^*\|^4.$$

Combine the last two inequalities to obtain

$$\frac{1}{n} \sum_{i=1}^{n} \|F_i(x) - F(x)\|^4 \leq 128 \left( \tfrac{1}{n} \sum_{i=1}^{n} L_i^4 \right) \|x - x^*\|^4 + 128\, \sigma_*^4$$

$\square$

### E.2 A COMBINATORIAL TOOL FOR HIGHER-ORDER MOMENTS IN SAMPLING WITHOUT REPLACEMENT

As part of our analysis, we require bounds on the fourth moment of empirical averages when the underlying data are sampled *without replacement*. While this result is of independent combinatorial interest—appearing naturally in the study of randomization effects and variance reduction—it also plays a technical role in controlling higher-order error terms in our proofs. The following lemma provides a clean upper bound in terms of simple population statistics such as $\hat\Sigma$ and $S_4$.

**Lemma E.2** (Fourth moment of a sample mean; upper bound)**.** Let $X_1, \ldots, X_n \in \mathbb{R}^d$ be fixed vectors, let

$$\bar{X} := \frac{1}{n} \sum_{i=1}^n X_i, \qquad r_i := X_i - \bar{X}, \qquad \hat\Sigma := \frac{1}{n} \sum_{i=1}^n r_i r_i^\top,$$

and define the population sums

$$S_4 := \sum_{i=1}^n \|r_i\|^4, \qquad U_2 := \sum_{i\neq j} \|r_i\|^2 \|r_j\|^2, \qquad T_2 := \sum_{i\neq j} (r_i^\top r_j)^2.$$

Draw a size-$k$ simple random sample without replacement, with indices $(\omega_1, \ldots, \omega_n)$ uniformly chosen among all $k$-subsets, and set $\bar{X}_\omega = \frac{1}{k} \sum_{t=1}^k X_{\omega_t}$. Then

$$\mathbb{E} \|\bar{X}_\omega - \bar{X}\|^4 \leq \frac{1}{k^4} \left[ \frac{k}{n} S_4 + \frac{9k(k-1)}{n(n-1)} \left( n^2 (\operatorname{tr} \hat\Sigma)^2 - S_4 \right) \right]. \tag{39}$$

*Proof.* Write $S := \sum_{t=1}^k r_{\omega_t}$ so that $\bar{X}_\omega - \bar{X} = \frac{1}{k} S$ and $\|\bar{X}_\omega - \bar{X}\|^4 = \frac{1}{k^4} \|S\|^4$. Using the Frobenius inner product $\langle A, B \rangle_F = \operatorname{tr}(A^\top B)$,

$$\|S\|^2 = \left\| \sum_{t=1}^k r_{\omega_t} \right\|^2 = \sum_{t,u=1}^k r_{\omega_t}^\top r_{\omega_u} = \left\langle \sum_{t=1}^k r_{\omega_t} r_{\omega_t}^\top, \sum_{u=1}^k r_{\omega_u} r_{\omega_u}^\top \right\rangle_F,$$

and hence

$$\|S\|^4 = \left( \sum_{t,u=1}^k r_{\omega_t}^\top r_{\omega_u} \right)^2 = \sum_{t,u,s,v=1}^k (r_{\omega_t}^\top r_{\omega_u})(r_{\omega_s}^\top r_{\omega_v}).$$

Taking expectation and using the inclusion probabilities for simple random sampling without replacement,

$$\mathbb{P}(\omega_a = i) = \frac{1}{n}, \qquad \mathbb{P}(\omega_a = i, \, \omega_b = j) = \frac{1}{n(n-1)} \quad (i \neq j),$$

we may group terms by the equality pattern among the *positions* $(t, u, s, v)$ (Hoeffding/U-statistics enumeration). Only three patterns survive:

**(P1) Diagonal–diagonal:** $(t = u)$ and $(s = v)$. This contributes

$$\frac{k}{n} S_4 \quad + \quad \frac{k(k-1)}{n(n-1)} U_2.$$

**(P2) Diagonal–off-diagonal (or vice versa):** exactly one of the pairs $(t, u)$ or $(s, v)$ is diagonal and the other is off-diagonal. Counting gives a coefficient $4\binom{k}{1}\binom{k-1}{2}$, leading to the contribution

$$\frac{4\binom{k}{1}\binom{k-1}{2}}{\binom{n}{1}\binom{n-1}{2}} T_2 = \frac{6k(k-1)}{n(n-1)} T_2.$$

**(P3) Off-diagonal–off-diagonal:** both pairs are off-diagonal but correspond to the same unordered pair of distinct sampled units. This yields

$$\frac{2\binom{k}{2}}{\binom{n}{2}} U_2 = \frac{2 \cdot \frac{k(k-1)}{2}}{\frac{n(n-1)}{2}} U_2 = \frac{2k(k-1)}{n(n-1)} U_2.$$

Summing the three contributions we obtain the exact identity

$$\mathbb{E}\|\bar{X}_\omega - \bar{X}\|^4 = \frac{1}{k^4}\left[\frac{k}{n}S_4 + \frac{3k(k-1)}{n(n-1)}U_2 + \frac{6k(k-1)}{n(n-1)}T_2\right]. \tag{40}$$

Next, by Cauchy–Schwarz,

$$(r_i^\top r_j)^2 \leq \|r_i\|^2\,\|r_j\|^2 \quad \text{for all } i \neq j,$$

hence $T_2 \leq U_2$. Plugging this into (40) gives

$$\mathbb{E}\|\bar{X}_\omega - \bar{X}\|^4 \leq \frac{1}{k^4}\left[\frac{k}{n}S_4 + \frac{(3+6)k(k-1)}{n(n-1)}U_2\right] = \frac{1}{k^4}\left[\frac{k}{n}S_4 + \frac{9k(k-1)}{n(n-1)}U_2\right].$$

Finally, observe the exact identity

$$U_2 = \sum_{i\neq j}\|r_i\|^2\|r_j\|^2 = \Big(\sum_{i=1}^n\|r_i\|^2\Big)^2 - \sum_{i=1}^n\|r_i\|^4 = n^2\big(\operatorname{tr}\hat{\Sigma}\big)^2 - S_4,$$

because $\sum_i\|r_i\|^2 = n\operatorname{tr}\hat{\Sigma}$. Substituting this into the previous display yields the claimed bound (39). $\qquad\square$

---

**Lemma E.3.** Let Assumptions 2.1, 2.3 hold. If Perturbed SGD–RR$_\mathrm{I}$ is run with step size $\gamma \leq \frac{1}{3nL_{max}}$, then it holds that

$$\mathbb{E}\Big[\sum_{i=1}^{n-1}\big\|x_k^i - x_k^0\big\|^4 \mid \mathcal{F}_k\Big] \leq 54\gamma^4 C\|x_k - x^*\|^4 + 3456\gamma^4(n-1)\sigma_*^4 + 972\gamma^4\frac{n(n+1)}{n-1}(\sigma_*^2)^2$$

where $C = 64nL_{max}^2 + 3n^2(n+1)L_{max}^2 + \frac{n^2(n+1)^2(2n+1)L^4}{10}$.

---

*Proof.* From the epoch-level update (8), it holds

$$\big\|x_k^i - x_k^0\big\|^4 \quad = \quad \gamma^4 i^4\left\|\frac{1}{i}\sum_{j=0}^{i-1}F_{\omega_k^j}(x_k^j)\right\|^4$$

$$\overset{(3)}{\leq} \quad 27\gamma^4 i^3\sum_{j=0}^{i-1}\big\|F_{\omega_k^j}(x_k^j) - F_{\omega_k^j}(x_k^0)\big\|^4 + 27\gamma^4 i^4\left\|\frac{1}{i}\sum_{j=0}^{i-1}F_{\omega_k^j}(x_k^0) - F(x_k^0)\right\|^4$$

$$+27\gamma^4 i^4\big\|F(x_k^0)\big\|^4$$

$$\overset{\text{Assumption 2.3}}{\leq} \quad 27\gamma^4 i^3 L_{max}^4\sum_{j=0}^{i-1}\big\|x_k^j - x_k^0\big\|^2 + 27\gamma^4 i^4\left\|\frac{1}{i}\sum_{j=0}^{i-1}F_{\omega_k^j}(x_k^0) - F(x_k^0)\right\|^4$$

$$+27\gamma^4 i^4\big\|F(x_k^0)\big\|^4$$

where at the last step we have used the Lipschitz property of the operators $F_i, \forall i \in [n]$. Taking expectation condition on the filtration $\mathcal{F}_k$, we get

$$\mathbb{E}\Big[\big\|x_k^i - x_k^0\big\|^4\Big|\mathcal{F}_k\Big] \leq 27\gamma^4 i^3 L_{max}^4\,\mathbb{E}\left[\sum_{j=0}^{i-1}\big\|x_k^j - x_k^0\big\|^4\Big|\mathcal{F}_k\right]$$

$$+27\gamma^4 i^4\,\mathbb{E}\left[\left\|\frac{1}{i}\sum_{j=0}^{i-1}F_{\omega_k^j}(x_k^0) - F(x_k^0)\right\|^4\Big|\mathcal{F}_k\right] + 27\gamma^4 i^4\big\|F(x_k^0)\big\|^4 \tag{41}$$

Substituting in Lemma E.2 $X_{\omega_t} := F_{\omega_k^j}(x_k^0)$ and $k = n$, we get that

$$\mathbb{E}\left[\left\|\frac{1}{i}\sum_{j=0}^{i-1}F_{\omega_k^j}(x_k^0) - F(x_k^0)\right\|^4 \Big| \mathcal{F}_k\right] \leq \frac{1}{i^4}\left[\frac{i}{n}S_4 + \frac{9i(i-1)}{n(n-1)}\left(n^2(\operatorname{tr}\hat{\Sigma})^2 - S_4\right)\right]$$

$$\leq \frac{1}{i^4}\left[\frac{i}{n}S_4 + \frac{9in(i-1)}{n-1}(\operatorname{tr}\hat{\Sigma})^2\right] \tag{42}$$

where $S_4 = \sum_{j=0}^{i-1}\|F_{\omega_k^j}(x_k^0) - F(x_k^0)\|^4 \geq 0$ and $\operatorname{tr}\hat{\Sigma} = \frac{1}{n}\sum_{i=0}^{n-1}\|F_i(x_k^0) - F(x_k^0)\|^2$. We, next, upper bound the terms $S_4$ and $\operatorname{tr}\hat{\Sigma}$. From Lemma E.2, we have that

$$S_4 \leq 128\left(\frac{1}{n}\sum_{i=1}^{n}L_i^4\right)\|x_k - x^*\|^4 + 128\sigma_*^4. \tag{43}$$

Using Proposition A.2 (Emmanouilidis et al., 2024), we have that

$$\operatorname{tr}\hat{\Sigma} = \frac{1}{n}\sum_{i=0}^{n-1}\|F_i(x_k^0) - F(x_k^0)\|^2 \leq A\|x_k - x^*\|^2 + 2\sigma_*^2, \tag{44}$$

since each $F_i$ is Lipschitz. Substituting (43), (44) into (41), we obtain that

$$i^4 \mathbb{E}\left[\left\|\frac{1}{i}\sum_{j=0}^{i-1}F_{\omega_k^j}(x_k^0) - F(x_k^0)\right\|^4 \Big| \mathcal{F}_k\right] \leq \frac{128i}{n^2}\sum_{i=1}^{n}L_i^4\|x_k - x^*\|^4 + \frac{128i}{n}\sigma_*^4$$

$$+ \frac{9i(i-1)}{n-1}(A\|x_k - x^*\|^2 + 2\sigma_*^2)^2$$

$$\overset{(2)}{\leq} \frac{128i}{n^2}A_4\|x_k - x^*\|^4 + \frac{128i}{n}\sigma_*^4$$

$$+ \frac{18i(i-1)}{n-1}A\|x_k - x^*\|^4 + \frac{36i(i-1)}{n-1}(\sigma_*^2)^2 \tag{45}$$

where we have let $A_4 := \sum_{i=1}^{n}L_i^4$ for brevity. From inequality (45) and (41), thus, we obtain

$$\mathbb{E}\left[\|x_k^i - x_k^0\|^4 \Big| \mathcal{F}_k\right] \leq 27\gamma^4 i^3 L_{max}^4 \mathbb{E}\left[\sum_{j=0}^{i-1}\|x_k^i - x_k^0\|^4 \Big| \mathcal{F}_k\right]$$

$$+ 27\gamma^4\left[\frac{128i}{n^2}A_4\|x_k - x^*\|^4 + \frac{128i}{n}\sigma_*^4 + \frac{18i(i-1)}{n-1}A\|x_k - x^*\|^4 + \frac{36i(i-1)}{n-1}(\sigma_*^2)^2\right]$$

$$+ 27\gamma^4 i^4\|F(x_k^0)\|^4$$

$$\leq 27\gamma^4 i^3 L_{max}^4 \mathbb{E}\left[\sum_{j=0}^{i-1}\|x_k^i - x_k^0\|^4 \Big| \mathcal{F}_k\right]$$

$$+ 27\gamma^4\left[\left(\frac{128i}{n^2}A_4 + \frac{18i(i-1)}{n-1}A\right)\|x_k - x^*\|^4 + \frac{128i}{n}\sigma_*^4 + \frac{36i(i-1)}{n-1}(\sigma_*^2)^2\right]$$

$$+ 27\gamma^4 i^4\|F(x_k^0)\|^4 \tag{46}$$

By summing over $0 \leq i \leq n-1$, we have that

$$\sum_{i=0}^{n-1}\mathbb{E}\left[\|x_k^i - x_k^0\|^2 \Big| \mathcal{F}_k\right] \leq 27\gamma^4 L_{max}^4 \frac{n^2(n-1)^2}{4}\sum_{i=0}^{n-1}\mathbb{E}\left[\|x_k^i - x_k^0\|^2 \Big| \mathcal{F}_k\right]$$

$$+ 27\gamma^4\left(\frac{64(n-1)}{n}A_4 + 18n(n+1)A\right)\|x_k - x^*\|^4 + 1728\gamma^4(n-1)\sigma_*^4$$

$$+ \gamma^4\frac{972n(n+1)}{n-1}(\sigma_*^2)^2 + 27\gamma^4\frac{n(n+1)(2n+1)(3n^2+3n-1)}{30}\|F(x_k^0)\|^4, \tag{47}$$

where we used the facts

$$\sum_{i=0}^{n-1} i = \frac{n(n-1)}{2}, \quad \sum_{i=0}^{n-1} i(i-1) = n(n+1)(n-1), \quad \sum_{i=0}^{n-1} \frac{i(n-i)}{n-1} = \frac{n(n+1)}{6},$$

$$\sum_{i=0}^{n-1} i^3 = \left(\frac{n(n-1)}{2}\right)^2, \quad \sum_{i=0}^{n-1} i^4 = \frac{n(n+1)(2n+1)(3n2+3n-1)}{30}.$$

We, thus, have that by rearranging the terms in (47) it holds that

$$\left[1 - 27\gamma^4 L_{max}^4 \frac{n^2(n-1)^2}{4}\right] \sum_{i=0}^{n-1} \mathbb{E}\left[\left\|x_k^i - x_k^0\right\|^4 \Big| \mathcal{F}_k\right]$$

$$\leq 27\gamma^4 \left(\frac{64(n-1)}{n} A_4 + 18n(n+1)A\right) \|x_k - x^*\|^4 + 1728\gamma^4(n-1)\sigma_*^4$$

$$+ \gamma^4 \frac{972n(n+1)}{n-1}(\sigma_*^2)^2 + 27\gamma^4 \frac{n(n+1)(2n+1)(3n^2+3n-1)}{30}\left\|F(x_k^0)\right\|^4$$

For $\gamma \leq \frac{1}{3nL_{max}}$ we have that $(1 - 27\gamma^4 L_{max}^4 \frac{n^2(n-1)^2}{4}) \geq \frac{1}{2}$, and thus we obtain

$$\sum_{i=0}^{n-1} \mathbb{E}\left[\left\|x_k^i - x_k^0\right\|^4 \Big| \mathcal{F}_k\right] \leq 54\gamma^4 \left(\frac{64(n-1)}{n} A_4 + 18n(n+1)A\right) \|x_k - x^*\|^4$$

$$+ 3456\gamma^4(n-1)\sigma_*^4 + \gamma^4 \frac{972n(n+1)}{n-1}(\sigma_*^2)^2$$

$$+ 54\gamma^4 \frac{n(n+1)(2n+1)(3n^2+3n-1)}{30}\left\|F(x_k^0)\right\|^4$$

$$\overset{2.3}{\leq} 54\gamma^4 \left(64A_4 + 18n(n+1)A + \frac{n^2(n+1)^2(2n+1)L^4}{10}\right) \|x_k - x^*\|^4$$

$$+ 3456\gamma^4(n-1)\sigma_*^4 + \gamma^4 \frac{972n(n+1)}{n-1}(\sigma_*^2)^2$$

$$\leq 54\gamma^4 \left(64nL_{max}^2 + 36n^2(n+1)L_{max}^2 + \frac{n^2(n+1)^2(2n+1)L^4}{10}\right) \|x_k - x^*\|^4$$

$$+ 3456\gamma^4(n-1)\sigma_*^4 + \gamma^4 \frac{972n(n+1)}{n-1}(\sigma_*^2)^2$$

where at the last step we used the fact that $A_4 = \sum_{i=0}^{n-1} L_i^4 \leq nL_{max}^4$ and $A = 2\sum_{i=0}^{n-1} L_i^2 \leq 2nL_{max}^2$. □

To establish Theorem 4.6, we will first prove upper bounds on the higher moments of the distance of the epoch-level iterates from the optimum, as well as the connection of the Lipschitz property of the per-step operators $F_i, i \in [n]$, and the Lipschitz constant of the epoch-level operator $G_{\omega_k}$. We start by providing the bound on the higher moments in the following Lemma.

**Lemma E.4.** Let Assumption 2.1-2.3 hold and $\lambda = 0$. Then, the iterates of Perturbed SGD–RR$_1$ with stepsize $\gamma \leq \gamma_{max}$ satisfy

$$\mathbb{E}\left[\|x_{k+1}^0 - x_*\|^3\right] = \mathcal{O}\left(n^{\frac{3}{2}}\gamma^3\right)$$

$$\mathbb{E}\left[\|x_{k+1}^0 - x_*\|^4\right] = \mathcal{O}\left(\gamma^4\right)$$

where $\gamma_{max} = \min\left\{\frac{\mu}{3nL_{max}^2}, \frac{\sqrt{1+6\mu^2 L_{max}^2}-1}{12nL_{max}^2}, \frac{\mu^{3/5}}{8nL_{max}^{3/5}}\right\}$.

*Proof.* We, first, provide the bound on the third moment. From Holder's inequality, we have that

$$\mathbb{E}\left[\|x_{k+1}^0 - x_*\|^3\right] \leq \left(\mathbb{E}\left[\|x_{k+1}^0 - x_*\|^2 \Big| \mathcal{F}_k\right]\right)^{\frac{3}{2}}$$

From Theorem 4.1, it holds that

$$\mathbb{E}\left[\|x_0^k - x^*\|^2\right] = \mathcal{O}\left(n\gamma^2\right),$$

and thus substituting we obtain

$$\mathbb{E}\left[\|x_{k+1}^0 - x_*\|^3\right] \leq \mathcal{O}\left(n^{\frac{3}{2}}\gamma^3\right),$$

where we have used the fact that the fourth moment of the stochastic oracles is bounded from Assumption 2.4. Taking the limit of the Markov chain we obtain the bound on the third moment

$$\mathbb{E}\left[\|x_{k+1}^0 - x_*\|^3\right] \leq \mathcal{O}\left(n^{\frac{3}{2}}\gamma^3\right)$$

We, next, bound the fourth moment $\mathbb{E}\left[\|x_{k+1}^0 - x_*\|^4\right]$. We have that

$$\|x_0^{k+1} - x^*\|^4 = \left(\|x_0^{k+1} - x^*\|^2\right)^2$$

$$\overset{(16)}{\leq} \left(\frac{\|x_k^0 - x^* - \gamma n F(x_k^0)\|^2}{1 - \gamma n\mu} + \frac{2\gamma L_{max}^2}{\mu}\sum_{i=1}^{n-1}\|x_k^i - x_k^0\|^2 + \frac{2\gamma}{n\mu}\|\mathbb{U}_k\|^2\right)^2$$

Letting $T_1 = \|x_k - x^* - \gamma n F(x_k^0)\|^2$, $T_2 = \sum_{i=1}^{n-1}\|x_k^i - x_k^0\|^2$ and $T_3 = \|\mathbb{U}_k\|^2$ for brevity, we have

$$\|x_0^{k+1} - x^*\|^4 \leq \left(\frac{T_1}{1 - \gamma n\mu} + \frac{2\gamma}{\mu}T_2 + \frac{2\gamma}{\mu}T_3\right)^2 \tag{48}$$

From (9) in Lemma C.2, we can bound the term $T_1$ by

$$T_1 \leq \left[(1 - \gamma n\mu)^2 + \gamma^2 n^2 L_{max}^2\right]\|x_k - x^*\|^2 + 2\gamma n\lambda$$

Substituting the bound of $T_1$ into (48), we get

$$\|x_0^{k+1} - x^*\|^4 \leq \left(\frac{(1 - \gamma n\mu)^2 + \gamma^2 n^2 L_{max}^2}{1 - \gamma n\mu}\|x_k - x^*\|^2 + \frac{2\gamma L_{max}^2}{\mu}T_2 + \frac{2\gamma}{n\mu}T_3\right)^2$$

Letting $c_1 = \frac{(1 - \gamma n\mu)^2 + \gamma^2 n^2 L_{max}^2}{1 - \gamma n\mu}$ for brevity and performing Young's inequality (7) with $t = c_1$, we obtain that

$$\|x_0^{k+1} - x^*\|^4 \leq \frac{1}{c_1}\left(c_1\|x_k - x^*\|^2 + \frac{2\gamma}{n\mu}T_3\right)^2 + \frac{4\gamma^2 L_{max}^4}{c_1\mu^2}T_2^2$$

$$\leq c_1\|x_0^k - x^*\|^4 + \frac{4\gamma^2}{\mu^2 n^2 c_1}T_3^2 + \frac{4c_1\gamma}{n\mu}T_3\|x_k - x^*\|^2 + \frac{2\gamma^2 L_{max}^4}{\mu^2}T_2^2$$

Taking expectation with respect to the permutation $\omega_k$ and condition on the filtration $\mathcal{F}^k$ (history of $x_k^0$) as well as using the fact that the noise $U_k$ is independent of the stochasticity of $\omega_k$, we get

$$\mathbb{E}\left[\|x_{k+1}^0 - x^*\|^4\Big|\mathcal{F}_k\right] \leq c_1\|x_0^k - x^*\|^4 + \frac{4\gamma^2}{\mu^2 n^2 c_1}T_3^2 + \frac{4c_1\gamma}{n\mu}T_3\|x_k - x^*\|^2 + \frac{2\gamma^2 L_{max}^4}{\mu^2}\mathbb{E}\left[T_2^2\Big|\mathcal{F}_k\right] \tag{49}$$

Using Lemma E.3, we can bound the term

$$\mathbb{E}\left[T_2^2\Big|\mathcal{F}_k\right] \leq 54\gamma^4 C\|x_k - x^*\|^4 + 3456\gamma^4(n-1)\sigma_*^4 + 972\gamma^4\frac{n(n+1)}{n-1}(\sigma_*^2)^2$$

where $C = 64n L_{max}^2 + 3n^2(n+1)L_{max}^2 + \frac{n^2(n+1)^2(2n+1)L^4}{10}$ and thus substituting into (49) we obtain that

$$\mathbb{E}\left[\|x_{k+1}^0 - x^*\|^4\Big|\mathcal{F}_k\right] \leq \left(c_1 + \frac{108\gamma^6 L_{max}^4}{\mu^2}C\right)\|x_0^k - x^*\|^4 + \frac{4\gamma^2}{\mu^2 n^2 c_1}T_3^2 + \frac{4c_1\gamma}{n\mu}T_3\|x_k - x^*\|^2$$

$$+ \frac{6912\gamma^6 L_{max}^4}{\mu^2}(n-1)\sigma_*^4 + \frac{1944\gamma^6 L_{max}^4}{\mu^2}\frac{n(n+1)}{n-1}(\sigma_*^2)^2. \tag{50}$$

The next step involves taking expectation on both sides with respect to the randomness of the $U_k$ and will require a bound on the terms $\mathbb{E}\left[T_3\right], \mathbb{E}\left[T_3^2\right]$. Since $\mathbb{U}_k \sim \mathcal{N}\left(0, \frac{\gamma^2 n^2}{d}\sigma_*^2 \mathbb{I}_d\right)$, we obtain

$$\mathbb{E}\left[\|\mathbb{U}_k\|^2\right] = \mathrm{tr}\left(\frac{\gamma^2 n^2}{d}\sigma_*^2 I_d\right) = \gamma^2 n^2 \sigma_*^2.$$

and

$$\mathbb{E}\left[\|U_k\|^4\right] = d(d+2)\left(\frac{\gamma^2 n^2}{d}\sigma_*^2\right)^2 \leq \gamma^4 n^4 (\sigma_*^2)^2,$$

as $\|U_k\|^2 / \left(\frac{\gamma^2 n^2}{d}\sigma_*^2\right)^2 \sim \chi_d^2$ and $\mathbb{E}\left[(\chi_d^2)^2\right] = d^2 + 2d$.

Thus, taking expectation on both sides of (50) and substituting the bounds on $\mathbb{E}\left[T_3\right], \mathbb{E}\left[T_3^2\right]$, we have that

$$
\begin{aligned}
\mathbb{E}\left[\|x_{k+1}^0 - x^*\|^4\right] &\leq \left(c_1 + \frac{108\gamma^6 L_{max}^4}{\mu^2}C\right)\|x_0^k - x^*\|^4 + \frac{4\gamma^2}{\mu^2 n^2 c_1}\mathbb{E}\left[T_3^2\right] + \frac{4c_1\gamma}{n\mu}\mathbb{E}\left[T_3\right]\|x_k - x^*\|^2 \\
&\quad + \frac{6912\gamma^6 L_{max}^4}{\mu^2}(n-1)\sigma_*^4 + \frac{1944\gamma^6 L_{max}^4}{\mu^2}\frac{n(n+1)}{n-1}(\sigma_*^2)^2 \\
&\leq \left(c_1 + \frac{108\gamma^6 L_{max}^4}{\mu^2}C\right)\|x_0^k - x^*\|^4 + \frac{4\gamma^6 n^2}{\mu^2 c_1}(\sigma_*^2)^2 + \frac{4c_1\gamma^3 n}{\mu}\sigma_*^2\|x_k - x^*\|^2 \\
&\quad + \frac{6912\gamma^6 L_{max}^4}{\mu^2}(n-1)\sigma_*^4 + \frac{1944\gamma^6 L_{max}^4}{\mu^2}\frac{n(n+1)}{n-1}(\sigma_*^2)^2
\end{aligned}
$$

Taking expectation on both sides, using the tower law of expectation and the fact that from Theorem 4.1 $\mathbb{E}\left[\|x_0^k - x^*\|^2\right] = \mathcal{O}\left(n\gamma^2\right)\sigma_*^2$, we obtain

$$
\begin{aligned}
\mathbb{E}\left[\|x_{k+1}^0 - x^*\|^4\right] &\leq \left(c_1 + \frac{108\gamma^6 L_{max}^4}{\mu^2}C\right)\mathbb{E}\left[\|x_0^k - x^*\|^4\right] + \frac{4\gamma^6 n^2}{\mu^2 c_1}(\sigma_*^2)^2 \\
&\quad + \frac{1944\gamma^6 L_{max}^4}{\mu^2}(\sigma_*^2)^2 + \frac{4c_1\gamma^3 n}{\mu}\sigma_*^2\mathbb{E}\left[\|x_k - x^*\|^2\right] \\
&\quad + \frac{6912\gamma^6 L_{max}^4}{\mu^2}(n-1)\sigma_*^4 + \frac{1944\gamma^6 L_{max}^4}{\mu^2}\frac{n(n+1)}{n-1}(\sigma_*^2)^2 \\
&\leq \left(c_1 + \frac{108\gamma^6 L_{max}^4}{\mu^2}C\right)\mathbb{E}\left[\|x_0^k - x^*\|^4\right] + \frac{4\gamma^6 n^2}{\mu^2 c_1}(\sigma_*^2)^2 \\
&\quad + \frac{1944\gamma^6 L_{max}^4}{\mu^2}(\sigma_*^2)^2 + \frac{4c_1\gamma^3 n}{\mu}\mathcal{O}\left(n\gamma^2\right)(\sigma_*^2)^2 \\
&\quad + \frac{6912\gamma^6 L_{max}^4}{\mu^2}(n-1)\sigma_*^4 + \frac{1944\gamma^6 L_{max}^4}{\mu^2}\frac{n(n+1)}{n-1}(\sigma_*^2)^2 \\
&\leq \left(c_1 + \frac{108\gamma^6 L_{max}^4}{\mu^2}C\right)\mathbb{E}\left[\|x_0^k - x^*\|^4\right] + \mathcal{O}\left(\frac{\gamma^6 n^2}{c_1} + \frac{c_1 n\gamma^5}{\mu} + \gamma^6 n\right)(\sigma_*^2)^2 \\
&\quad + \frac{6912\gamma^6 L_{max}^4}{\mu^2}(n-1)\sigma_*^4
\end{aligned}
\tag{51}
$$

Selecting the stepsize $\gamma \leq \min\left\{\frac{1}{3nL_{max}}, \frac{\mu}{3nL_{max}^2}, \frac{\mu^{3/5}}{8nL_{max}^{3/5}}\right\}$, we have that

$$c_1 = 1 - \gamma n\mu + \frac{\gamma^2 n^2 L_{max}^2}{1 - \gamma n\mu} \leq 1 - \frac{\gamma n\mu}{2}$$

and

$$
\begin{aligned}
c_1 + \frac{108\gamma^6 L_{max}^4}{\mu^2}C &\leq 1 - \frac{\gamma n\mu}{2} + \frac{108\gamma^6 L_{max}^4}{\mu^2}C \\
&\overset{C \leq 64n^5 L_{max}^4}{\leq} 1 - \frac{\gamma n\mu}{2} + \frac{108\gamma^6 L_{max}^4}{\mu^2}64n^5 L_{max}^4 \\
&\leq 1 - \frac{\gamma n\mu}{4}
\end{aligned}
\tag{52}
$$

Thus, substituting (52) in (51)

$$\mathbb{E}\Big[\|x_{k+1}^0 - x^*\|^4\Big] \leq \left(1 - \frac{\gamma n \mu}{4}\right)\mathbb{E}\Big[\|x_0^k - x^*\|^4\Big] + \mathcal{O}\left(\gamma^5 n\right)(\sigma_*^2)^2 + \frac{6912\gamma^6 L_{max}^4}{\mu^2}(n-1)\sigma_*^4$$

Taking expectation with respect to the invariant measure $\pi_\gamma$ and using Assumption 2.4 and the fact that $\sigma_*^2 < +\infty$, we obtain

$$\frac{\gamma n \mu}{4}\,\mathbb{E}_{x \sim \pi_\gamma}\,\|x - x^*\|^4 \leq \mathcal{O}\left(\gamma^5 n\right)$$

and thus we have

$$\mathbb{E}_{x \sim \pi_\gamma}\,\|x - x^*\|^4 \leq \mathcal{O}\left(\gamma^4\right)$$

$\square$

# F    PROOF OF BIAS FOR $\mathrm{RR}_1$ AND $\mathrm{RR}_1 \oplus \mathrm{RR}_2$ METHODS (LEMMA 4.5) AND (THEOREM 4.6)

## F.1    JACOBIAN BOUND FOR ONE-PASS MAP

**Lemma F.1** (Jacobian bound for one-pass map). Assume each component $F_i : \mathbb{R}^d \to \mathbb{R}^d$ is $C^1$ near $x^*$ with $\|\nabla F_i(x^*)\|_{op} \leq L_i$ and let $L_{\max} = \max_i L_i$. For a permutation $\omega \in \mathfrak{S}_n$, define the inner one-step map

$$\Phi_{\omega,i}(x) := x - \gamma F_{\omega_i}(x), \qquad i = 0, \dots, n-1,$$

its composition over one reshuffled pass

$$\Phi_\omega^{(n)}(x) := \Phi_{\omega,n-1} \circ \cdots \circ \Phi_{\omega,0}(x),$$

and the epoch map

$$H(x,\omega) := x - \gamma \sum_{i=0}^{n-1} F_{\omega_i}\big(x^{[i]}(x,\omega)\big), \quad x^{[0]}(x,\omega) = x, \ \ x^{[i+1]}(x,\omega) = \Phi_{\omega,i}\big(x^{[i]}(x,\omega)\big).$$

Then, at $x^*$ it holds that

$$\|\nabla_x G(x^*,\omega)\|_{op} \leq 1 + \sum_{i=1}^{n}(\gamma L_{\max})^i.$$

Consequently, the spectral radius of $\nabla_x G(x^*,\omega)$ is at most $1 + \sum_{i=1}^{n}(\gamma L_{\max})^i$.

Our first lemma aims to bound the maximum eigenvalue of the matrix $\nabla_x H(\omega, x^*)$ with respect to the known Lipschitz constants of the operators $F_i, \forall i \in [n]$.

**Lemma F.2.** The maximum eigenvalue of the operator $\nabla_x G(x^*,\omega)$ is $L_{max}^G = 1 + \sum_{i=1}^{n}(\gamma L_{max})^i$.

*Proof.* Let $\phi_{\omega_i}(x,z) = x - \gamma F_{\omega_i}(z)$. Define, also, the $k-$step operator $\phi_\omega^{(k)}(x,z) = \phi_{\omega_k}\big(x, \phi_{\omega_{k-1}}(x, \dots \phi_{\omega_1}(x,z)\dots)\big)$ with $\phi_\omega^{(0)}(x,z) = z$ and obtain that

$$x_{k+1}^0 = H(x_k^0, \omega)$$
$$\nabla_x G(x_k^0, \omega) = I - \nabla \phi_\omega^{(n)}(x_k^0, x_k^0)$$

since $G(x_k^0, \omega) := \sum_{i=0}^{n-1} F_{\omega_k^i}\big(x_k^i\big)$.

The gradient of $G(\cdot, \omega)$ is computed by deriving first the partial derivatives of $\phi_\omega^{(n)}(x,z)$ with respect to $x$ and $z$. We prove by induction that

- $\nabla_z \phi_\omega^{(n)}(x, z) = (-\gamma)^n \nabla F_{\omega_n}\left(\phi_\omega^{(n-1)}(x, z)\right) \cdot \nabla F_{\omega_{n-1}}\left(\phi_\omega^{(n-2)}(x, z)\right) \cdot \ldots \cdot \nabla F_{\omega_1}\left(\phi_\omega^{(0)}(x, z)\right)$

- $\nabla_x \phi_\omega^{(n)}(x, z) = \sum_{j=0}^{n-1}(-\gamma)^j \nabla F_{\omega_{n-1}}(\phi_\omega^{(n)}(x, z))\nabla F_{\omega_{n-1}}\left(\phi_\omega^{(n-2)}(x, z)\right) \cdot \ldots \cdot \nabla F_{\omega_{n-j+1}}\left(\phi_\omega^{(n-j)}(x, z)\right)$

For $k = 1$, we have that $\phi_\omega^{(1)}(x, z) = \phi_{\omega_1}(x, z) = x - \gamma F_{\omega_1}(z)$ and it holds that

$$\begin{aligned}
\nabla_z \phi_\omega^{(1)}(x, z) &= -\gamma \nabla F_{\omega_0}(z) \\
\nabla_x \phi_\omega^{(1)}(x, z) &= I
\end{aligned}$$

thus the inductive hypothesis holds for $k = 1$. Assuming that it holds for $n - 1$, we, next, prove that it holds for $n$. We have that

$$\begin{aligned}
\nabla_z \phi_\omega^{(n)}(x, z) &= \nabla_z \phi_\omega\left(x, \phi_\omega^{(n-1)}(x, z)\right)\nabla_z\phi_\omega^{(n-1)}(x, z) \\
&= (-\gamma)\nabla F_{\omega_n}\left(\phi_\omega^{(n-1)}(x, z)\right)\cdot(-\gamma)^{n-1}\nabla F_{\omega_{n-1}}\left(\phi_\omega^{(n-2)}(x, z)\right)\cdot\ldots\cdot\nabla F_{\omega_1}\left(\phi_\omega^{(0)}(x, z)\right) \\
&= (-\gamma)^n\nabla F_{\omega_n}\left(\phi_\omega^{(n-1)}(x, z)\right)\cdot\nabla F_{\omega_{n-1}}\left(\phi_\omega^{(n-2)}(x, z)\right)\cdot\ldots\cdot\nabla F_{\omega_1}\left(\phi_\omega^{(0)}(x, z)\right)
\end{aligned}$$

We, next, compute the gradient with respect to $x$ and get

$$\nabla_x \phi_\omega^n(x, z) = \nabla_x \phi_\omega(x, \phi_\omega^{(n-1)}(x, z)) + \nabla_z \phi_\omega\left(x, \phi_\omega^{(n-1)}(x, z)\right)\nabla_x\phi_\omega^{(n-1)}(x, z) \quad (53)$$

Using the fact that $\nabla_x \phi_\omega(x, \phi_\omega^{(n-1)}(x, z)) = I, \nabla_z \phi_\omega\left(x, \phi_\omega^{(n-1)}(x, z)\right) = \nabla F_{\omega_n}(\phi_\omega^{(n-1)}(x, z))$ and the inductive hypothesis for $\nabla_x \phi_\omega^{(n-1)}(x, z)$, we obtain

$$\begin{aligned}
\nabla_z \phi_\omega^{(n)}(x, z) &= I - \gamma \nabla F_{\omega_n}(\phi_\omega^{(n-1)}(x, z))\sum_{j=0}^{n-2}(-\gamma)^j \nabla F_{\omega_{n-1}}\left(\phi_\omega^{(n-2)}(x, z)\right)\cdot\ldots\cdot\nabla F_{\omega_{n-j}}\left(\phi_\omega^{(n-1-j)}(x, z)\right) \\
&= I + \sum_{j=0}^{n-2}(-\gamma)^{j+1}\nabla F_{\omega_{n-1}}(\phi_\omega^{(n)}(x, z))\nabla F_{\omega_{n-1}}\left(\phi_\omega^{(n-2)}(x, z)\right)\cdot\ldots\cdot\nabla F_{\omega_{n-j}}\left(\phi_\omega^{(n-1-j)}(x, z)\right) \\
&= I + \sum_{j=1}^{n-1}(-\gamma)^j\nabla F_{\omega_{n-1}}(\phi_\omega^{(n)}(x, z))\nabla F_{\omega_{n-1}}\left(\phi_\omega^{(n-2)}(x, z)\right)\cdot\ldots\cdot\nabla F_{\omega_{n-j+1}}\left(\phi_\omega^{(n-j)}(x, z)\right) \\
&= \sum_{j=0}^{n-1}(-\gamma)^j\nabla F_{\omega_{n-1}}(\phi_\omega^{(n)}(x, z))\nabla F_{\omega_{n-1}}\left(\phi_\omega^{(n-2)}(x, z)\right)\cdot\ldots\cdot\nabla F_{\omega_{n-j+1}}\left(\phi_\omega^{(n-j)}(x, z)\right)
\end{aligned}$$

Thus, in order to compute $\nabla_x G(\omega, x^*) = I - \nabla\phi_\omega^{(n)}(x^*, x^*)$, we first compute $\nabla\phi_\omega^{(n)}(x^*, x^*)$. Since $x^*$ is a stationary point, it is a fixed point of the operator $\phi_\omega^{(j)}(x^*, x^*) = x^*, \forall j \geq 0$. From chain rule, we have that

$$\begin{aligned}
\nabla\phi_\omega^{(n)}(x^*, x^*) &= \nabla_z\phi_\omega^{(n)}(x^*, x^*) + \nabla_x\phi_\omega^{(n)}(x^*, x^*) &(54) \\
&= (-\gamma)^n\prod_{j=1}^{n}\nabla F_{\omega_j}(x^*) + \sum_{i=1}^{n-1}\prod_{j=1}^{i}(-\gamma\nabla F_{\omega_{n-j}}(x^*)) &(55) \\
&= \sum_{i=1}^{n}\prod_{j=1}^{i}(-\gamma\nabla F_{\omega_{n-j}}(x^*)) &(56)
\end{aligned}$$

In order to find the maximum eigenvalue of the operator $\nabla\phi_\omega^{(n)}(x^*, x^*)$, we apply the submultiplicative property of the operator norm to get

$$
\left\| \sum_{i=1}^{n} \prod_{j=1}^{i} (-\gamma\nabla F_{\omega_{n-j}}(x^*)) \right\|_{\text{op}} \leq \sum_{i=1}^{n} \left\| \prod_{j=1}^{i} (-\gamma\nabla F_{\omega_{n-j}}(x^*)) \right\|_{\text{op}}
$$

$$
\leq \sum_{i=1}^{n} \prod_{j=1}^{i} \| -\gamma\nabla F_{\omega_{n-j}}(x^*) \|_{\text{op}}
$$

$$
\leq \sum_{i=1}^{n} \gamma^i \prod_{j=1}^{i} L_{n-j} \tag{57}
$$

$$
\leq \sum_{i=1}^{n} (\gamma L_{max})^i \tag{58}
$$

where $L_i$ is the maximum eigenvalue of $\nabla F_i(x^*)$ and $L_{max}$ is the maximum over all eigenvalues of $\nabla F_i(x^*), \forall i \in [n]$. Since $\nabla G(\omega, x^*) = I - \nabla\phi_\omega^{(n)}(x_k^0, x^*)$, using the submultiplicative property of the operator norm we have that $\|\nabla G(\omega, x^*)\|_{\text{op}} \leq 1 + \|\nabla\phi_\omega^{(n)}(x_k^0, x^*)\|_{\text{op}}$ and thus the maximum eigenvalue of $\nabla G(\omega, x^*)$ is $L_{max}^G = 1 + \sum_{i=1}^{n} (\gamma L_{max})^i$. $\qquad\square$

We, next, provide the theorem establishing that the combination of the two heuristics lead to a refined bias of the order $\mathcal{O}\left(\gamma^3\right)$.

### F.2 HIGHER-ORDER TERMS OF RR₁ BIAS
(LEMMA 4.5)

**Lemma F.3** (Extended version of Lemma 4.5)**.** Let $\lambda = 0$ and Assumptions 2.1–2.4 hold. If `Perturbed SGD-RR₁` is run with $\gamma < \gamma_{max}$, it holds that

$$
\text{bias}(\texttt{Perturbed SGD-RR}_1) = \limsup_{k\to\infty} \| \mathbb{E}[x_k] - x^* \| = C(x^*)\gamma + \mathcal{O}(\gamma^3).
$$

$$
\Updownarrow
$$

$$
\mathbb{E}_{\pi_\gamma}[x] = x_* + \gamma A + \mathcal{O}\left(\gamma^3\right)
$$

where $A = -\frac{1}{2}\nabla_x H(\omega, x^*)^{-1}\nabla^2 H(\omega, x^*) M \int_{\mathbb{R}^d} C(x)\pi_\gamma(dx), C = \mathbb{E}\left[\mathbb{U}_{\omega_{1k}}^{\otimes 2}\right], L_{max}^G = 1 + \sum_{i=1}^{n} L_{max}^i, M = \nabla_x H(\omega, x^*) \otimes I + I \otimes \nabla_x H(\omega, x^*) - \gamma\nabla_x H(\omega, x^*) \otimes \nabla_x H(\omega, x^*)$ and the maximum step size is $\gamma_{max} = \min\left\{ \frac{\mu}{3nL_{max}^2}, \frac{\sqrt{1+6\mu^2 L_{max}^2}-1}{12nL_{max}^2}, \frac{\mu^{3/5}}{8nL_{max}^{3/5}} \right\}$.

*Proof.* From a third order Taylor expansion of $G$ around $x_*$, we have that

$$
H(\omega, x) = \nabla_x H(\omega, x^*)(x - x^*) + \frac{1}{2}\nabla^2 H(\omega, x^*)(x - x^*)^{\otimes 2} + R_1(x), \forall x \in \mathbb{R}^d \tag{59}
$$

where the reminder $R_1(x)$ satisfies $\sup_{x\in\mathbb{R}^d}\left\{ \frac{R_1(x)\|}{\|x-x^*\|^3} \right\} < +\infty$. Taking expectation with respect to the invariant distribution $\pi_\gamma$ and using the fact that $\mathbb{E}_{\pi_\gamma}[H(\omega, x)] = 0$, we get

$$
0 = \mathbb{E}_{\pi_\gamma}\left[ \nabla_x H(\omega, x^*)(x - x^*) + \frac{1}{2}\nabla^2 H(\omega, x^*)(x - x^*)^{\otimes 2} + R_1(x) \right] \tag{60}
$$

From Lemma E.4 and using Holder inequality and the fact that $\sup_{x\in\mathbb{R}^d}\left\{ \frac{R_1(x)\|}{\|x-x^*\|^3} \right\} < +\infty$, we obtain

$$
\nabla_x H(\omega, x^*)\mathbb{E}_{\pi_\gamma}[x - x^*] + \frac{1}{2}\nabla^2 H(\omega, x^*)\int_{\mathbb{R}^d} (x - x^*)^{\otimes 2}\pi_\gamma(dx) = \mathcal{O}\left(\gamma^3\right) \tag{61}
$$

Taking the second order Taylor of $G$ around $x^*$, we have that

$$x_0^1 - x^* = x_0^0 - x^* - \gamma \nabla_x H(\omega, x^*)(x_0^0 - x^*) + \gamma \mathbb{U}_1^k(x_0^0) + \gamma R_2(x_0^0) \tag{62}$$

with $\mathcal{R}_2$ the second order reminder satisfying $\sup_{x \in \mathbb{R}^d} \left\{ \frac{R_2(x)\|}{\|x-x^*\|^2} \right\} < +\infty$. From the second order moment of equation (62), the unbiasedness of the noise $\mathbb{U}_k, \forall i, k \in \mathbb{N}$, and Theorem E.4, we have that

$$
\begin{aligned}
\int_{\mathbb{R}^d} (x - x^*)^{\otimes 2} \pi_\gamma(dx) &= [I - \gamma \nabla_x H(\omega, x^*)] \int_{\mathbb{R}^d} (x - x^*)^{\otimes 2} \pi_\gamma(dx) [I - \gamma \nabla_x H(\omega, x^*)] \\
&\quad + \gamma^2 \int_{\mathbb{R}^d} C(x) \pi_\gamma(dx) + \mathcal{O}\left(\gamma^5\right)
\end{aligned}
$$

Rearranging the terms, we get

$$M \int_{\mathbb{R}^d} (x - x^*)^{\otimes 2} \pi_\gamma(dx) = \gamma \int_{\mathbb{R}^d} C(x) \pi_\gamma(dx) + \mathcal{O}\left(\gamma^3\right) \tag{63}$$

where $M = \nabla_x H(\omega, x^*) \otimes I + I \otimes \nabla_x H(\omega, x^*) - \gamma \nabla_x H(\omega, x^*) \otimes \nabla_x H(\omega, x^*)$.

We, next, show that the operator $M$ is invertible for the selected step size by proving that it is symmetric and positive definite. Let $\lambda_i, \forall i \in [d]$, be the eigenvalues of $\nabla_x H(\omega, x^*)$ with $\{u_i\}_{i \in [d]}$ the corresponding eigenvectors. Note that $I - \frac{\gamma}{2} \nabla_x H(\omega, x^*)$ has eigenvalues $(1 - \frac{\gamma}{2} \lambda_i) > 0$ and thus for $\gamma < \frac{2}{\lambda_{max}(\nabla_x H(\omega, x_*))}$ it is symmetric positive definite on the same basis $\{u_i\}_{i \in [d]}$. Hence we can factor the operator $M$ as

$$
\begin{aligned}
M &= \nabla_x H(\omega, x^*) \otimes I + I \otimes \nabla_x H(\omega, x^*) - \gamma \nabla_x H(\omega, x^*) \otimes \nabla_x H(\omega, x^*) \\
&= \nabla_x H(\omega, x^*) \otimes (I - \frac{\gamma}{2} \nabla_x H(\omega, x^*)) + (I - \frac{\gamma}{2} \nabla_x H(\omega, x^*)) \otimes \nabla_x H(\omega, x^*)
\end{aligned}
$$

Thus, the vectors $u_i \otimes u_j, \forall i, j \in [d]$ diagonalize $M$ with eigenvalues $\mu_{i,j} = \lambda_i(1 - \gamma \lambda_j) + \lambda_j(1 - \gamma \lambda_i), \forall i, j \in [d]$. From Lemma F.2, we have that the maximum eigenvalue of $\nabla_x H(\omega, x^*)$ is $L_{max}^G = 1 + \sum_{i=1}^n (\gamma L_{max})^i$ and hence for $\gamma < 1$ we have that $\gamma L_{max} \leq 1$ and hence $L_{max}^G < \tilde{L}_{max}^G =: 1 + n$. Selecting the stepsize such that $\gamma < \frac{2}{n+1}$, it holds that $\mu_{i,j} > 0, \forall i, j \in [d]$, and thus $M$ is positive definite and invertible. Thus, multiplying (63) with $M^{-1}$ from the left, we get

$$\int_{\mathbb{R}^d} (x - x^*)^{\otimes 2} \pi_\gamma(dx) = \gamma M^{-1} \int_{\mathbb{R}^d} C(x) \pi_\gamma(dx) + \mathcal{O}\left(\gamma^3\right) \tag{64}$$

Substituting (64) into (61) and rearranging the terms, we obtain

$$\nabla_x H(\omega, x^*) \mathbb{E}_{\pi_\gamma}[x - x^*] = -\frac{\gamma}{2} \nabla^2 H(\omega, x^*) M \int_{\mathbb{R}^d} C(x) \pi_\gamma(dx) + \mathcal{O}\left(\gamma^3\right)$$

$$\Rightarrow \mathbb{E}_{\pi_\gamma}[x - x^*] = -\frac{\gamma}{2} \nabla_x H(\omega, x^*)^{-1} \nabla^2 H(\omega, x^*) M \int_{\mathbb{R}^d} C(x) \pi_\gamma(dx) + \mathcal{O}\left(\gamma^3\right) \tag{65}$$

Letting $A = -\frac{1}{2} \nabla_x H(\omega, x^*)^{-1} \nabla^2 H(\omega, x^*) M \int_{\mathbb{R}^d} C(x) \pi_\gamma(dx)$, we obtain

$$\mathbb{E}_{\pi_\gamma}[x] = x_* + \gamma A + \mathcal{O}\left(\gamma^3\right) \tag{66}$$

$\square$

### F.3 PROOF OF BIAS REFINEMENTS OF $RR_1 \oplus RR_2$ (THEOREM 4.6)

**Theorem F.4** (Restatement of Theorem 4.6). Under the assumptions of Lemma 4.5, Algorithm 1 output satisfies

Last-iterate version (line 9): $\quad \| \mathbb{E}[x_k] - x^* \| \leq c(1 - \rho)^k + \mathcal{O}(\gamma^3),$

Averaged-iterate version (line 10): $\quad \left\| \mathbb{E}\left[ \frac{1}{k} \sum_{m=1}^k x_m \right] - x^* \right\| \leq \frac{c/\rho}{k} + \mathcal{O}(\gamma^3).$

where $\rho \in (0, 1)$, $c < \infty$ (cf. Theorem 4.3).

*Proof.* From Lemma F.3, we have that the iterates $x_{\gamma,k}$ of Perturbed SGD–RR$_1$ with step size $\gamma$ satisfy

$$\mathbb{E}_{x_\gamma \sim \pi_\gamma}[x] = x_* + \gamma A + \mathcal{O}\left(\gamma^3\right) \tag{67}$$

Similarly the iterates $(x_{2\gamma,k})_k$ of SGD-RR with step size $2\gamma$ satisfy

$$\mathbb{E}_{x_{2\gamma} \sim \pi_{2\gamma}}[x_{2\gamma}] = x_* + 2\gamma A + \mathcal{O}\left(\gamma^3\right) \tag{68}$$

Thus, from (67), (68) we can compute the Richardson Romberg iterates as

$$\left(\mathbb{E}_{x_\gamma \sim \pi_\gamma}[2x] - \mathbb{E}_{x_{2\gamma} \sim \pi_{2\gamma}}[x_{2\gamma}]\right) = \mathcal{O}\left(\gamma^3\right) \tag{69}$$

Consider the test function $\ell(x) = x$. The function satisfies the assumptions in both Theorem D.7, 4.4. Combining (69) with the rate that the iterates of the method tend to the limiting invariant distribution and the corresponding Central Limit Theorem from Theorems D.7, 4.4, we obtain

$$\| \mathbb{E}[x_k] - x^* \| \leq c(1 - \rho)^k + \mathcal{O}(\gamma^3), \tag{70}$$

$$\left\| \mathbb{E}\left[ \frac{1}{k} \sum_{m=1}^{k} x_m \right] - x^* \right\| \leq \frac{c/\rho}{k} + \mathcal{O}(\gamma^3). \tag{71}$$

where $\rho \in (0, 1), c \in (0, +\infty)$. $\qquad\square$

## G ON EXPERIMENTS

In this section, we provide additional details on the experimental setting used for the conducted experiments. We consider the setup of strongly monotone quadratic min-max problems

$$\min_{x_1 \in \mathbb{R}^d} \max_{x_2 \in \mathbb{R}^d} f(x_1, x_2) = \frac{1}{n} \sum_{i=1}^n \frac{1}{2} x_1^T A_i x_1 + x_1^T B_i x_2 - \frac{1}{2} x_2^T C_i x_2 + \alpha_i^T x_1 - c_i^T x_2.$$

The matrices $A_i$ are sampled by first sampling an orthogonal matrix $P$ and then sampling a diagonal matrix $D_i$ with elements in the diagonal uniformly sampled from the interval $[\mu, L]$. The selected parameters $\mu, L$ correspond to the strong monotonicity parameter in Assumption 2.2 and the Lipschitz parameter of the underlying problem respectively. We acquire the matrices $A_i$ as the product $A_i = PD_iP^T$. We sample the matrices $B_i, C_i$ similarly to sampling the matrices $A_i$ with the only difference that the elements of the diagonal matrices $D_i$ lie in the interval $[0, 0.1]$ and $[\mu, L]$ respectively. The vectors $a_i, c_i$ are follow the normal distribution $\mathcal{N}(\mathbf{0}, I)$. In all experiments, we use $n = 100, d = 100$, while we specify the values of $\mu, L$ in each experiment independently as they differ.

We provide additional experiments on the effect of each heuristic in the convergence of the algorithm. More specifically, we compare the classical with-replacement SGDA algorithm, the $RR_1$ variant, the $RR_2$ variant and the algorithm utilizing both heuristics. We run the experiment for multiple stepsizes $\gamma = \{10^{-3}, 10^{-4}, 10^{-5}\}$ and multiple condition numbers $\kappa = \{1, 5, 10\}$.

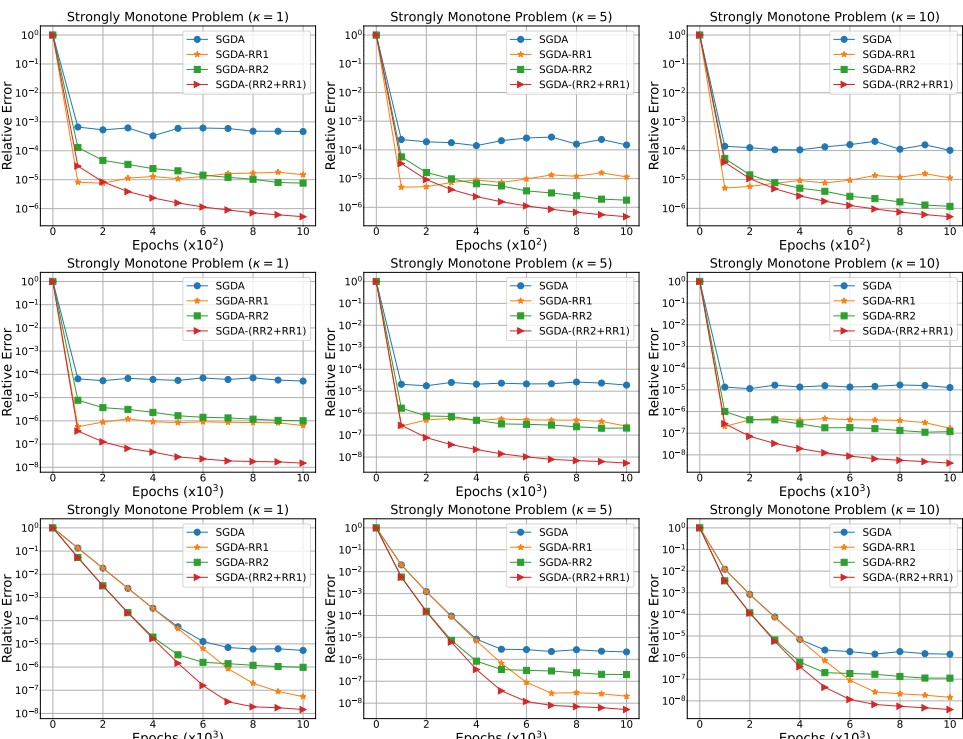

Figure 5: Relative Error of the different variants of SGDA. Each row corresponds to a strongly monotone problem with condition number $\kappa = \{1, 5, 10\}$ and each row corresponds to a different step size $\gamma = \{10^{-3}, 10^{-4}, 10^{-5}\}$. The combination of both heuristics $RR_2 \oplus RR_1$ achieves the smallest relative error in comparison to the other methods.

In Figure 5, we observe that all variants converge linearly to a neighbourhood of the solution. Demonstrably, the variant leveraging both heuristics outperforms the other variants, reducing faster the relative error and validating the theoretical results established so far.

We, next, provide an ablation study on the effect of the proposed heuristic in a variety of common algorithms used in VI and machine learning settings.

**Wasserstein GANs.** We train a Wasserstein GAN (WGAN) (Arjovsky et al., 2017) for learning the mean of a multivariate Gaussian and consider the effects of each heuristic in the training of the GAN. In a Wasserstein GAN, the optimization objective is a two-player zero-sum game between the generator $G(\cdot)$ and the discriminator $D(\cdot)$, given by

$$\inf_\theta \sup_w \mathbb{E}_{x \sim N(v,I)} \left[ \langle w, x \rangle \right] - \mathbb{E}_{z \sim N(0,I)} \left[ \langle w, z + \theta \rangle \right]. \tag{72}$$

The discriminator consists a linear classifier $D(x; w) = \langle w, x \rangle$ of the input $x \in \mathbb{R}^d$, while the generator returns a noisy estimation $G(z; \theta) = z + \theta$ of the learned parameter $\theta \in \mathbb{R}^d$, after sampling a noise vector $z \sim \mathcal{N}(0, I)$. In our experiments, the aim of the generator is to learn the mean $\mu$ of the true Gaussian distribution with mean $\mu = [3, 4]^T$ and covariance $\Sigma = \frac{1}{10} I$.

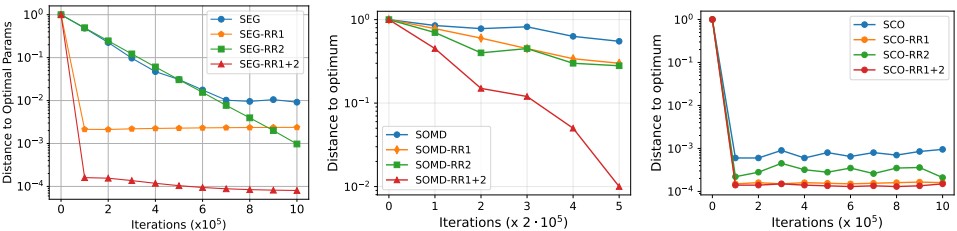

Figure 6: Wasserstein GAN trained with different heuristics on top of a base algorithm. For all base algorithms, the generator trained with the combination of both heuristics $\text{RR}_2 \oplus \text{RR}_1$ converges closer to the optimal parameters than the generator trained with any other algorithmic variant.

We examine the effect of the heuristics in a variety of different training algorithms and report the distance from the generator's optimal parameters for each experiment. Similar to Emmanouilidis et al. (2024), we, first, consider the Stochastic Extragradient (SEG) method as the main algorithmic template for training and use each one of the 4 variants (SEG, SEG-$\text{RR}_1$, SEG-$\text{RR}_2$, SEG-$\text{RR}_2 \oplus \text{RR}_1$) to train a GAN. We use the same constant step size for the generator and discriminator as in Emmanouilidis et al. (2024) and double the step size of the variants that implement Richardson-Romberg extrapolation. Figure 6 shows that the generator trained with SEG-$\text{RR}_2 \oplus \text{RR}_1$ is able to converge closer to the optimal parameters than the generator trained with any other variant and thus the synergy of both heuristics ($\text{RR}_2 \oplus \text{RR}_1$) is beneficial in training.

Following Daskalakis et al. (2017), we train a WGAN with the use of Optimistic Mirror Descent (OMD). Aiming to see the effect of each heuristic even for this algorithm, we use the classical OMD method, the $\text{RR}_1$ variant, the random reshuffling ($\text{RR}_2$) and the combination of both ($\text{RR}_2 \oplus \text{RR}_1$). We let the step size of the generator and the discriminator be $\gamma_G = 0.02, \gamma_D = 0.01$ respectively. As shown in Figure 6, the $\text{RR}_2 \oplus \text{RR}_1$ outperforms all other variants, indicating that the advantages of this heuristic remain apparent even for the OMD algorithm.

Lastly, we test lightweight second order methods common in the literature of VIs (Mescheder et al., 2017; Loizou et al., 2020). More specifically, Stochastic Consensus Optimization (SCO) is an algorithm that can be seen as a combination of the SGDA algorithm and the Stochastic Hamiltonian (SHMD) method (Loizou et al., 2020), where a regularizer $\lambda$ articulates the contribution of SHMD that is being introduced in the update rule. Given that the SCO method is related to the SGDA but requires Jacobian vector products, thus being a lightweight second order method, we have run experiments to examine whether the $\text{RR}_2 \oplus \text{RR}_1$ provides benefits in higher-order methods. According to Figure 6, the generator trained with the heuristic $\text{RR}_2 \oplus \text{RR}_1$ converges closer to the optimal parameters than the generator trained with plain SCO or any other variant.

**On Single-Run Experiments & Variance of Observed Behaviour.** For completeness, we report the variability of our experimental results over single runs, establishing a more refined description of the effect of each heuristic empirically. More specifically, in Figure 7 we plot the mean and standard deviations for each of the 4 variants over 5 runs. As shown in Figure 7, the $\text{RR}_2 \oplus \text{RR}_1$ variant outperforms all other heuristics even in single trials.

**Wall-clock time Comparison of the different heuristics.** We, next, compare the wall-clock time of the different heuristics. All 4 variants have the same per iteration cost in terms of gradient evaluations, since the only difference between the classical SGD algorithm and the $\text{RR}_1$ variant is the

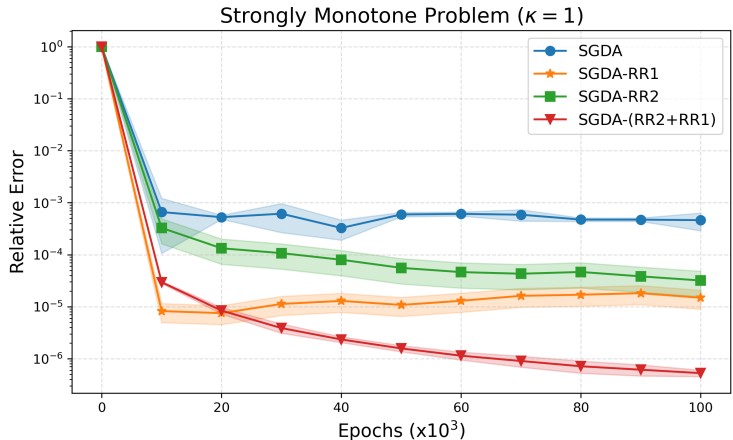

Figure 7: Mean and standard deviation of each heuristic over 5 trials. The combination of both heuristics $RR_2 \oplus RR_1$ achieves the smallest relative error in comparison to the other methods.

way that the mini-batch gradients are sampled, while in $RR_2$ and $RR_2 \oplus RR_1$ the two chains of the Richardson extrapolation can be run in parallel.

Interestingly, SGDA with random reshuffling typically runs faster in wall-clock time than SGDA with with-replacement sampling. The following factors explain why random reshuffling can be run faster, as observed in our experiments:

- $RR_1$ performs only one random operation per epoch. With with-replacement sampling, each iteration requires a random draw $i_t \sim Uniform(1, \ldots, n)$.
- $RR_1$ calls the PRNG once per epoch (through randperm(n) or equivalent), after which all iterations are sequential. This eliminates thousands of PRNG calls and reduces interpreter overhead.

We have reproduced the same experiment as in Figure **??** and have reported the wall-clock time needed for each method. According to table 2, the $RR_2 \oplus RR_1$ heuristic runs in half the time required for the plain SGDA variant and comparable time with respect to random reshuffling. Hence, thanks to parallelization one can obtain the benefits from the synergy of the two heuristics without the need of a higher wall-clock time.

Table 2: Wall-clock time comparison of SGDA variants.

| Method | Time (sec) |
|---|---|
| SGDA | 107.59 |
| SGDA-$RR_1$ | 50.92 |
| SGDA-$RR_2$ | 107.59 |
| SGDA-$RR_2 \oplus RR_1$ | 51.94 |

