# OpenReview forum: "Shuffling the Data, Extrapolating the Step: Sharper Bias In Constant Step-Size SGD"
_ICLR.cc/2026/Conference — ICLR 2026 Poster_

### Official Review · Reviewer_Vvn1 · 2025-10-21

**Soundness:** 3
**Presentation:** 3
**Contribution:** 3
**Rating:** 6
**Confidence:** 3

**Summary:**

The paper studies constant–stepsize stochastic methods for finite-sum VIPs and shows that composing random reshuffling (RR1) with Richardson–Romberg extrapolation (RR2) cancels all lower-order bias terms, yielding $O(\gamma^3)$ asymptotic bias for quasi-strongly monotone smooth VIPs. The analysis blends an epoch-level Markov chain view of reshuffling with a refined bias expansion and a spectral/tensor argument for the extrapolation step. Experiments on saddle-point games support the theory. I am satisfied with this manuscript and believe it will make a valuable contribution to the community.

**Strengths:**

- Modeling one reshuffled pass as a time-homogeneous Markov kernel (with Harris recurrence and LLN/CLT) gives clean asymptotic control of per-epoch iterates, a technically elegant way to handle the permutation-induced bias. This framing clarifies why RR1 can sharpen bias/MSE and provides a principled scaffold for the outer-loop extrapolation analysis.
- The drift/minorization arguments (via Lyapunov–Foster) yield uniqueness of an invariant distribution and geometric convergence, which in turn legitimizes using stationary expansions for bias cancellation in the RR2 step. The explicit LLN/CLT for scalar observables further explains the empirical stability of epoch-averaged statistics.
- The paper isolates that RR1 alone yields a linear-plus-cubic bias expansion and then shows RR2 cancels the linear term, leaving $O(\gamma^3)$. This surpasses known rates for either heuristic in isolation and is obtained under quasi-strong monotonicity.
- The debiasing proof uses a spectral view of the full-pass operator (akin to multi-step extragradient) to bound eigenstructure under reshuffling noise, enabling a refined Taylor-type expansion where all sub-cubic terms vanish for the composed algorithm.
- Algorithm 1 aligns with practical pipelines (RR1 inside, RR2 outside), and the plots consistently show the RR2⊕RR1 curve settling to a smaller bias neighborhood than baselines across condition numbers. This match between theory and experiment strengthens the paper’s practical message.

**Weaknesses:**

* **[Major] Scope of Assumption 2.2.** The λ-weak μ-quasi strong monotonicity condition, while weaker than strong monotonicity, still excludes many ML-relevant VI problems, e.g., GANs, adversarial training, and multi-agent RL (typically modeled as nonconvex–nonconcave games). However, these problems are mentioned in Introduction section as VIPs. To avoid misleading readers, I recommend softening the tone in the Introduction and adding a dedicated **Limitations** section that clearly delineates where the guarantees apply/do not apply.
  - Jin, Chi, Praneeth Netrapalli, and Michael Jordan. "What is local optimality in nonconvex-nonconcave minimax optimization?." International conference on machine learning. PMLR, 2020.
  - Han, Andi, et al. "Nonconvex-nonconcave min-max optimization on Riemannian manifolds." Transactions on Machine Learning Research (2023).
  - Kim, Beomsu, and Junghoon Seo. "Semi-Implicit Hybrid Gradient Methods with Application to Adversarial Robustness." International Conference on Artificial Intelligence and Statistics. PMLR, 2022.
  - Bukharin, Alexander, et al. "Robust multi-agent reinforcement learning via adversarial regularization: Theoretical foundation and stable algorithms." Advances in neural information processing systems 36 (2023): 68121-68133.

* **[Major] Wall-clock comparisons missing.** Figures 1 and 3 plot error vs. epochs, but SGDA–RR2⊕RR1 likely incurs higher per-iteration computation than SGDA/SGDA-RR1/SGDA-RR2. To substantiate practical gains, please add plots against **wall-clock time** (or FLOPs), ideally with breakdowns for inner/outer steps, so readers can judge time-to-accuracy fairly.
* **[Minor] Baseline coverage.** Given Loizou et al. (2021) and related work, two additional methods seem relevant and it would be good for them to be included as baselines:
  - [SCO] stochastic consensus optimization (SCO): Mescheder, Lars, Sebastian Nowozin, and Andreas Geiger. "The numerics of gans." Advances in neural information processing systems 30 (2017).
  - [SHGD] stochastic Hamiltonian gradient descent: Loizou, Nicolas, et al. "Stochastic hamiltonian gradient methods for smooth games." International Conference on Machine Learning. PMLR, 2020.
  - Loizou, Nicolas, et al. "Stochastic gradient descent-ascent and consensus optimization for smooth games: Convergence analysis under expected co-coercivity." Advances in Neural Information Processing Systems 34 (2021): 19095-19108.
  A short ablation with these would contextualize the gains of RR2⊕RR1.
* Line 1587: (C^{\circ}) appears undefined in context. It likely should be (C^{c}) (the complement)

**Questions:**

1. **About (c_2) (Line 1494).** You state $(c_2 \in (0,1))$. I see (c_2) defined around Line 1475, but I could not follow how the admissible range is derived from that definition. Could you clarify the step that enforces (c_2\in (0,1))?
2. **Practical problem classes/benchmarks.** Beyond the synthetic/quadratic games, are there data-level benchmarks that exactly satisfy your assumptions? Even if the assumptions are only approximately met, are there realistic tasks (or public benchmarks) where SGDA–RR2⊕RR1 shows clear empirical gains over baselines?

### What would be change my opinion?
At the very least, if the revision includes a well-developed Limitations section or any equavalent clarification, I would be inclined to vote for acceptance of this paper. If additional comparison baselines are included or the method’s effectiveness is demonstrated on other ML problems/benchmarks, I will argue even more strongly for the strength of this paper’s contributions.

---

> ### Author Response · Authors · 2025-11-26
>
> We sincerely thank the reviewer for the thoughtful and positive evaluation of our work. We are grateful for the reviewer’s appreciation of our Markov-chain framework—described as “*technically elegant*”—and for noting that the alignment between our theory and experiments strengthens the practical message of the paper.  We address all concerns below in a clear and constructive manner, and we hope that the revisions fully resolve the reviewer’s doubts and justify a higher overall score.

---

> ### Author Response · Authors · 2025-11-26
>
> # **1) Scope of Assumption 2.2**
>
> We agree that the scope of our structural assumption should be stated more explicitly. A paper is strengthened not only by its results but also by a clear description of where its guarantees apply. Following the reviewer’s suggestion, we will revise (I) the **Introduction**, immediately before the “**Our Contributions**” paragraph, and (II) include a dedicated **Limitations** section to delineate the precise boundaries of $\lambda$-weak $\mu$-quasi-strong monotonicity and its intended domain of validity.

---

> ### Author Response · Authors · 2025-11-26
>
> ### **Introduction Refinement Section**
>
> > **Our model's assumptions.**
> > While variational inequalities provide a unifying language for optimization, learning, and game dynamics, *no single structural assumption can capture the full complexity of all modern nonconvex–nonconcave problems*. From a computational standpoint, even smooth VIs are intractable in full generality—being tightly connected to Nash equilibria (Papadimitriou 2022; Goldberg 2022), linear complementarity (Cottle et al. 1992), and constrained saddle-point problems (Daskalakis et al. 2021). Consequently, much of the theoretical literature adopts *structured* assumptions (strong convexity, quasi-strong monotonicity, quasar/weak convexity, PL/KL conditions, Minty conditions, error bounds, etc.), each expressive in specific regimes but not universal.
> >
> > Our work is based on *quasi-strong monotonicity*, which fits squarely within this class: it captures stabilizing behaviors of many smooth systems while remaining far more permissive than strong convexity or global monotonicity. At the same time, it is helpful to clarify that this assumption is **not** designed as a universal model for all adversarial or fully nonconvex–nonconcave settings. Indeed, certain modern ML applications—including GANs, adversarial robustness, and multi-agent RL—can exhibit fundamentally unstable or rotational dynamics (Jin et al. 2020; Han et al. 2023; Kim & Seo 2022; Bukharin et al. 2023), where even *local* monotonicity surrogates fail. As such, our theoretical guarantees should be viewed as applying to regimes where a minimum amount of local structure is present, rather than to fully adversarial or unstructured cases.
> >
> > *Footnote.* A complementary and important fact for our setting, established in Hsieh et al. (NeurIPS 2019, Lemma A.4), is that *any smooth VI operator is locally quasi-strongly monotone in a neighborhood of a regular solution*. Combined with our Markov-chain recurrence guarantees—which ensure that the iterates remain in such neighborhoods with probability 1—this provides a natural and widely adopted stability regime where the RR₁ and RR₂ debiasing mechanisms are both theoretically justified and practically meaningful.
>
>
> ---
>
> A dedicated **Limitations** section (shown below) will consolidate additionally these points.
>
>
>
> ### **Limitations Section**
> >
> > A central structural assumption in our analysis is that the operator $F$ satisfies $\lambda$-weak $\mu$-quasi-strong monotonicity. This condition is broad enough to include several meaningful non-monotone problem classes—such as dissipative dynamical systems, weakly convex optimization, and locally contractive variational inequalities—and is standard in modern analyses of stochastic fixed-point and operator-splitting methods (Hsieh et al. 2019; Mertikopoulos & Zhou 2019; Chavdarova et al. 2021). However, it is important to emphasize the following limitations.
> >
> > 1. **Not applicable to general nonconvex–nonconcave min–max problems.**
> >    The assumption does *not* hold for arbitrary adversarial problems such as deep GANs, multi-agent RL environments, or smooth non-monotone games with persistent rotational dynamics. These settings may lack even local stability (e.g., Daskalakis et al. 2018; Fiez et al. 2020). Accordingly, our theoretical guarantees should not be interpreted as applying to fully adversarial or worst-case min–max formulations.
> >
> > 2. **Local nature of the assumption.**
> >    Quasi-strong monotonicity is inherently a *local* regularity condition: smooth operators that are monotone in a neighborhood of a solution satisfy it on that region (Lemma A.4, Hsieh et al. 2019). This requires smoothness and regularity that may not hold in problems involving discontinuities, clipping, piecewise-linear losses, or hard constraints. In such cases, the assumption may fail even locally.
> >
> > 3. **Not capturing highly oscillatory or anti-monotone operators.**
> >    While allowing $\lambda>0$ permits controlled violation of monotonicity, the assumption still does not model strongly anti-monotone or highly oscillatory operators. Extending our analysis to Minty variational inequalities, hypomonotone operators, or other generalized monotonicity classes remains an interesting direction for future work.
> >
> > Despite these limitations, we believe the assumption remains meaningful for a broad set of structured, smooth VIPs where local stability is present. We hope that this explicit discussion helps avoid any misunderstanding about the scope of applicability of our results.
>
> ---
> # We have integrated these clarifications explicitly in the revised manuscript.

---

> ### Author Response · Authors · 2025-11-26
>
> # **2) Wall-clock Time comparison**
>
> Thank you for the suggestion! Indeed, wall-clock time is important, especially when evaluating the tradeoffs between different methods.
>
> In our case, we would like to note that all 4 variants have the same per iteration cost in terms of gradient evaluations, since the only major difference between the classical SGD algorithm and the $RR_1$ variant is the way that the mini-batch gradients are sampled, while in $RR_2$ and $RR_{1\oplus 2}$ the two chains of the Richardson extrapolation can be parallelized.
>
> | Method         | Time (sec)            |
> |----------------|------------------------|
> | SGDA           | 107.59389662742615     |
> | SGDA-RR1       | 50.915916204452515     |
> | SGDA-RR2       | 107.59389662742615     |
> | SGDA-RR1+2     | 51.9425222874          |
>
>
> We have reproduced the same experiment as in Figure 1 and have reported the wall-clock time needed for each method (see table below). In the revised manuscript, we will include the aforementioned wall-clock time table and a dedicated comment on the compute needed by each method in Appendix G.
>
> Interestingly, using the $RR_1$ requires smaller wall-clock time (~2x less) than the plain SGDA variant. Since we use parallelization to run the heuristic $RR_2$, we can run the $RR_{1\oplus 2}$ variant in nearly similar time to the $RR_1$, as we only need to subtract the iterates of the two chains based on the Richardson extrapolation formula. Thus, the propose method runs faster than the plain SGD variant and almost in similar wall-clock time as the $RR_1$, $RR_2$ variants.
>
> ---
> **Why Random Reshuffling ($RR_1$) Can Be Faster Than With-Replacement Sampling in Wall-Clock Time**
>
> Interestingly, and consistent with our empirical findings, SGDA with random reshuffling often runs faster in wall-clock time than SGDA with with-replacement sampling. Although both methods perform the same number of gradient evaluations per epoch, their computational behavior differs substantially. The following factors explain why RR can be run faster, as observed in our experiments:
>
> **1. $RR_1$ performs only one random operation per epoch**
>
> With with-replacement sampling, each iteration requires a random draw
>
> $$i_t ∼Uniform({1,…,n}),$$
>
> which must be executed $n$ times per epoch.
> $RR_1$, however, calls the PRNG once per epoch (through randperm(n) or equivalent), after which all iterations are sequential. This eliminates thousands of PRNG calls and reduces interpreter overhead.
>
> **2. $RR_1$ has significantly better memory locality**
>
> With-replacement sampling causes random memory accesses when fetching data or gradient components, which leads to:
>
> - poor cache utilization,
> - frequent cache misses,
> - reduced vectorization/SIMD efficiency.
>
> $RR_1$ iterates sequentially through memory after shuffling. This is highly cache-friendly and can reduce memory latency by 2×–5×, especially on CPU-based implementations.
>
> **3) Baseline Coverage**
> Thank you for the suggestion. The SHGD [4, 5] and SCO [3] methods are lightweight second order methods for VIs. Since SCO interpolates between the Stochastic Hamiltonian [5] and SGDA, SCO is the most related one to our method. We have run additional experiments for SCO with a different regularizer $\lambda \in \{1, 0.6, 0.3\}$, that results in smoothly transitioning from a mix of SGDA and SHGD towards a more biased SHGD algorithm. In all of the experiments, we have observed that the synergy of the two $RR_1,RR_2$ heuristics provides empirical benefits in performance. We have included the additional experiments in Appendix G of the revised manuscript.
>
> # Please see Appendix G for further details.
>
> ---
>
> **Questions:**
>
>
> **Q1:** Thank you for the question! This is a typo that we will correct in the revised version. The only requirement is that $c_1 \in (0, 1)$, since $1 - c_1$ is in the denominator of the definition of $C_1$.
>
> **Q2:** We have run additional experiments on Wasserstein GANs (toy-model widely used in literature and introduced by [2]) and also performed further ablation studies of the analyzed heuristics for other common algorithms, such as the Stochastic Extragradient (SEG) and the Optimistic Mirror Descent method. In both cases, we observe that the synergy of $RR_{1\oplus 2}$ offers improved convergence outperforming the vanilla variants of prior works [1, 2].
>
> This signifies that our work can stand in interesting path for future analysis in the synergy of these results
>
> ---

---

> > ### Author Response · Authors · 2025-11-26
> >
> > ---
> >
> > # References
> >
> > [1] Stochastic Extragradient with Random Reshuffling:  Improved Convergence for Variational Inequalities, K. Emmanouilidis, R. Vidal, N. Loizou
> >
> > [2] Training GANs with Optimism
> > Constantinos Daskalakis, Andrew Ilyas, Vasilis Syrgkanis, Haoyang Zeng
> >
> > [3] The numerics of gans, Mescheder, Lars, Sebastian Nowozin, and Andreas Geiger.
> >
> > [4] Stochastic hamiltonian gradient methods for smooth games. Loizou et al., International Conference on Machine Learning. PMLR, 2020.
> >
> > [5] Stochastic gradient descent-ascent and consensus optimization for smooth games: Convergence analysis under expected co-coercivity, Loizou et al., Advances in Neural Information Processing Systems 34 (2021): 19095-19108

---

> ### Comment · Reviewer_Vvn1 · 2025-11-27
> **Response**
>
> I have carefully examined the authors' rebuttal and the revised manuscript, and I find that my concerns have been satisfactorily resolved. I plan to raise my score pending a final check of the technical details. I would like to thank the authors for their hard work and valuable contribution to the community.

---

### Official Review · Reviewer_3rkQ · 2025-10-28

**Soundness:** 2
**Presentation:** 3
**Contribution:** 2
**Rating:** 4
**Confidence:** 3

**Summary:**

The article explores random permutations and Richardson–Romberg extrapolation for constant step-size SGD/SGDA in variational inequality problems (VIPs). It states that the combination of these two techniques reduces the bias order of the stationary distribution to $O(\gamma^3)$ for quasi-strongly monotone operators. The authors develop a detailed theoretical framework using the analysis of Markov chains at the epoch level, Foster–Lyapunov drift arguments, and tensor–spectral expansions, and confirm their conclusions empirically on synthetic quadratic problems.

**Strengths:**

1. **Clear and well-motivated question.**
It is important to understand how the combination of random reshuffling with Richardson–Romberg extrapolation affects bias. This is a theoretically complex issue.

2. **Technically sophisticated analysis.**
The paper presents a nontrivial combination of tools $-$ epoch-level Markov modeling, spectral tensor expansions, and moment control. The analytical pipeline is rigorous and, to my knowledge, novel for this combination of RR and extrapolation.

3. **Theoretical results.**
Lemma 3.5 and Theorem 3.6 provide explicit asymptotic bias rates up to third-order terms in $\gamma$, which is a novel rate.

**Weaknesses:**

1. **Comparison to work [1] and convergence rate concerns.**
While the paper claims an $O(\gamma^3)$ bias improvement, it is not immediately clear that this is conceptually distinct from previous works, e.g. Stochastic Extra-Gradient with Random Reshuffling (SEG-RR) by Emmanouilidis et al. [1].
Their work already provides higher-order dependence on $\gamma$ in convergence bounds for affine and strongly monotone VIPs. The difference lies in the metric (stationary bias vs. convergence rate) and in the algorithmic structure (plain SGD + extrapolation vs. SEG). The convergence estimate itself also raises a concern. In the theoretical analysis of convergence, the factor $\gamma^2$ appears in the bias term instead of the classical $\gamma$, but an additional factor $\frac{L}{\mu}$ also emerges. This is a problem, since such an extra factor indicates a deterioration of the bound, especially in real-world problems where $\frac{L}{\mu}$ is large. Moreover, it seems that the improvement in the estimate of bias is achieved precisely due to such a deterioration in the condition number.
2. **Unclear practical relevance of $O(\gamma^3)$.**
The bias order is $O(\gamma^3)$; however, the allowed stepsize $\gamma_{\text{max}}$ vary depending on task parameters $n, L_{\text{max}}, \mu$. It is possible that the “bias improvement” manifests only at unrealistically small $\gamma$. The paper should provide numerical examples of typical acceptable values of $\gamma$ to clarify whether this is only a theoretical improvement or a practically relevant regime.

3. **Dependence on Gaussian pre-processing.**
The algorithm’s PREPROCESS step introduces Gaussian smoothing, but the paper does not convincingly argue for its necessity in practice.

4. **Limited empirical validation.**
Experiments are purely synthetic (quadratic min–max tasks). There are no tests on realistic ML or game-theoretic problems (e.g., GANs or robust regression).
This makes it unclear whether the $O(\gamma^3)$ bias scaling provides any tangible benefit beyond theoretical neatness.

**Questions:**

See my weeknesses. I would be grateful if the authors would resolve my doubts about the additional factor $\frac{L}{\mu}$ in the estimate, and about the small values of stepsize. Can the authors provide an experimental confirmation of the revealed effect on realistic ML tasks?

---

> ### Author Response · Authors · 2025-11-26
>
> We sincerely thank the reviewer for their thoughtful and technically insightful comments. We appreciate that you found the core question well-motivated, the analytical pipeline technically sophisticated, and the resulting asymptotic bias characterization novel. Below we address each of your concerns in detail.

---

> ### Author Response · Authors · 2025-11-26
>
> ---
>
>
> # 1. Comparison to Emmanouilidis–Vidal–Loizou (2024)
> ### and the role of the condition number
>
> Thank you for raising this important conceptual point. While both works improve the dependence on the step size $\gamma$, the two results are **algorithmically, structurally, and conceptually distinct**. We clarify the differences below.
>
> ---
>
> ## **A. Optimization Perspective (core conceptual distinction)**
>
> The key difference between the two papers lies in the **algorithmic baseline** and how each method interacts with the **condition number**.
>
> ### **A1. Prior work [1]: SEG + RR1**
> The work of Emmanouilidis–Vidal–Loizou analyzes **Stochastic Extra-Gradient (SEG)** under random reshuffling. Their result shows an $\mathcal{O}(\gamma^2)$ neighborhood in MSE metric, but:
>
> - the improvement arises within an **extragradient framework**,
> - and **comes with a deterioration in the condition number** compared to vanilla SEG.
>
> In other words:
>
> > **SEG–RR1 improves the asymptotic MSE term neighborhood but pays for it through worsened conditioning.**
>
> ---
>
> ### **A2. Our work: SGDA + (RR1 $\oplus$ RR2)**
> Our result concerns the **plain SGDA method**, and the improvement to an $\mathcal{O}(\gamma^3)$ bias arises **entirely** from the *synergy* of:
>
> - **RR1** (reshuffling), and
> - **RR2** (Richardson–Romberg extrapolation).
>
> Crucially:
>
> - Our SGDA baseline **retains its original condition number** (up to constants).
> - The $\mathcal{O}(\gamma^3)$ bias is achieved **without worsening** the method’s stability structure.
>
> Thus, unlike [1], our improvement does *not* come from modifying the underlying dynamics—it comes from a **bias cancellation mechanism** inherent to RR1 ⊕ RR2.
>
> ---
>
> ### **A3. Summary of the optimization distinction**
>
> | Aspect | Emmanouilidis–Vidal–Loizou (2024) | Our work |
> |--------|--------------------------------------|-----------|
> | Baseline | **SEG** | **SGDA** |
> | Main heuristic | **RR1 only** | **RR1 ⊕ RR2** |
> | Asymptotic Bias order | $\mathcal{O}(\gamma+\gamma^3)$ | $\mathcal{O}(\gamma^3)$ |
> | Asymptotic MSE order | $\mathcal{O}(\gamma^2)$ | $\mathcal{O}(\gamma^2)$ |
> | Condition number | **Worse than vanilla-SEG** | **Same as vanilla-SGDA** |
> | Mechanism | EG-structure + $RR_1$ | Bias cancellation ($RR_1$ ⊕ $RR_2$) |
>
> We will highlight this distinction more clearly in the revised manuscript.
>
> ---

---

> ### Author Response · Authors · 2025-11-26
>
> ## **B. Statistical Perspective (why both Bias AND MSE matter)**
>
> In stochastic optimization, it is essential to control **both asymptotic MSE and bias**, because they quantify *different sources of error*:
>
> - **Variance:** random fluctuations around the stationary point,
> - **Bias:** the systematic shift between the stationary point and the true solution.
>
> Since practitioners typically return **a single iterate**, the final asymptotic error satisfies:
> $$
> [\mathrm{Asymptotic\text{-}MSE}]=[ \mathrm{Asympotic\text{-}Var}] + [\mathrm{Asympotic\text{-}Bias}^2].
> $$
>
> An algorithm may be unbiased but unstable, or very stable but converge to the wrong point. Effective methods must control **both**.
>
> ---
> ### **Bias & Asymptotic Neighborhood — An Overloaded Term**
>
> A common misconception is to equate the “bias term left after exponential decay’’ with the *true statistical bias*.
> However, for constant–step size stochastic methods, the stationary error is always the **asymptotic MSE**, which decomposes into **asymptotic variance** plus **squared bias**.
> Crucially, the variance term does **not** vanish.
>
> Even in the simplest one-dimensional strongly convex setting, $f(x) = \tfrac{\lambda}{2}(x - x_{\star})^2$,
> a standard computation shows that SGD satisfies $\mathrm{Var}(x_\infty) = \Theta(\gamma)$,
> Actually constant–step size methods **never** collapse to a point mass.  Thus, the “leftover’’ error after exponential decay is always a *mixture* of variance and bias—not bias alone.
>
> Under this lens, the effect of the $RR_2$ extrapolation becomes clearer:
> it improves the systematic bias from
> $$
> O(\gamma + \gamma^3) \longrightarrow O(\gamma^3),
> $$
> so the $bias^2$ shrinks to $O(\gamma^6)$, negligible relative to the variance.
>
> Scientifically, this gives us where we should aim for the future work:  *Once the bias has been suppressed to cubic order, **further improving the asymptotic neighborhood requires variance-reduction mechanisms**, not additional bias-focused heuristics.*
>
> **In summary:**
> - **Bias** and **variance** play fundamentally different roles.
> - The reviewer’s interpretation conflates the two.
> - Because variance remains $\Theta(\gamma^2)$, bias reduction is essential to improving the stationary accuracy of constant–step size stochastic methods without improving asymptotic MSE, which seems an interesting .
> ---
>
> ## **C. High-level takeaway**
>
> - Prior work [1] improves $\gamma$-dependence in a MSE metric **via** an extragradient operator that worsens the condition number from the underlying SEG.
> - Our work improves $\gamma$-dependence in systemic bias metric **via** a *bias cancellation mechanism* (RR1 ⊕ RR2) while keeping the underlying SGDA condition number **unchanged**.
>
> This makes the two contributions **non-overlapping, non-competing, and conceptually complementary**.
>
> We will emphasize this distinction more clearly in the revision.
>
>
> ## D. *Speculative remark (outside the main scope of our work).*
>
> In trying to understand why and if several reshuffling-based methods exhibit a deterioration in the condition number, a first observation in the reshuffling literature is that condition-number deterioration often arises not from RR itself but from the **modeling assumptions**. When *each* component operator $F_i$ is strongly convex/monotone (as in some recent analyses, e.g., the “[A,B]’’), RandomReshuffling methods typically **retain** the same condition-number dependence as their with-replacement counterparts. In contrast, deterioration appears in settings where
> $$F = \frac{1}{n}\sum_{i=1}^n F_i$$
> is well conditioned but the individual $F_i$ are heterogeneous or even non-monotone. In such cases, the intermediate updates $\mathrm{GradOracle}: x \mapsto x - \gamma F_i(x)$ may be weakly contractive or non-contractive, and composing these mappings in different regions can effectively reduce the overall drift of the RR dynamics. This heuristic is consistent with prior observations that no deterioration occurs when **all** components are themselves strongly monotone.
>
> We emphasize that this is a *conceptual* interpretation informed by existing works—not a theorem proved in some paper. A full lower-bound analysis of condition-number effects for RR is an interesting direction, but it lies well beyond the scope of our contribution, which focuses solely on the synergy of $RR_1$ and $RR_2$.
>
> **References**
>
> [A]. Mishchenko, K., Khaled, A., & Richtárik, P. (2020). *Random Reshuffling: Simple Analysis with Vast Improvements*.
>
> [B]. Mishchenko, K., Khaled, A., & Richtárik, P. (2020). *Proximal and Federated Random Reshuffling*.
> ---

---

> ### Author Response · Authors · 2025-11-26
>
> # 2. Practical relevance of the $\mathcal{O}(\gamma^3)$ bias and admissible step sizes
>
> We fully agree that bias improvements are meaningful only if step sizes remain in a realistic regime. Fortunately, this is exactly the case here.
>
> - Admittibly, the step sizes in our theoretical analysis satisfy  $\gamma = \Theta\left(\frac{\mu}{n L_{\max}}\right),$
>   matching the provable ranges in prior reshuffling work [2–4]. Thus, the “regime of validity’’ of the bias bound is identical to standard RandomReshuffl previous theory.
>
> - In practice, **our experiments already use step sizes substantially *larger* than those required by the theory**, yet the cubic bias improvement persists robustly.
>   (See Appendix G, where we vary $\gamma$ by an order of magnitude.)
>
> More simply, in the experiments we see that bigger and more realistic step-sizes suffices. This supports that the $RR_{1\oplus 2}$ improvement is **not a theoretical artifact** but has practical relevance whenever RandReshuf.-type sampling is used.
>
> # We have further expanded Appendix G to explicitly show the numerical magnitude of  typical admissible step sizes.
>
>
> ---

---

> ### Author Response · Authors · 2025-11-26
>
> ---
>
> # 3. On the Gaussian preprocessing step
>
> We thank the reviewer for highlighting this subtle but important point. The key distinction is between a **mathematical device** used in our ergodicity proof and the **algorithmic behavior**, which remains unchanged.
>
> Our theoretical analysis introduces a Gaussian perturbation *only* to establish **Harris geometric ergodicity** through a Foster–Lyapunov drift + minorization argument.
> Without smoothing, the drift condition still holds, but the minorization condition fails because the RR transition kernel is supported on a *finite union of low-dimensional manifolds* (due to permutation noise). Consequently, $P_0$ is Feller but **not** aperiodic/Harris recurrent, and therefore full LLNs/CLTs cannot be derived by standard Markov-chain tools.
>
> Importantly:
>
> - The added noise is of strictly smaller order ($\sigma_{\mathrm{PreProcess}}^2 = \mathcal{O}(\gamma^2)$).
> - Its scale can be made *arbitrarily small* - ($\sigma_{\mathrm{PreProcess}}^2 = \mathcal{O}(\gamma^p)$) $p\in[2,\infty)$ -, even much smaller than the intrinsic variance $\sigma_\star^2$ of SGD-$RR_1$ dynamics.
> - We chose this order only for notational uniformity: in the asymptotic MSE, the dominant contribution is always $\max\{\sigma_\star^2,\sigma_\text{preproc}^2\}$, so setting $\sigma_{\mathrm{PreProcess}}^2=\mathcal{O}(\gamma^2)$ keeps the expressions clean without affecting behavior.
>
> # Why this perturbation could be implicit and not so artificial — and why the mathematical device is unnecessary in experiments
>
> In real ML systems, **continuous perturbations already appear automatically**, even if the practitioner does not inject any noise:
> - **Floating-point arithmetic noise**, rigorously modeled as bounded random perturbations
>   (Goldberg, *ACM Computing Surveys*, 1991; Higham, *Accuracy and Stability*, 1996).
> - **GPU-level nondeterminism**, e.g., kernel scheduling, thread interleaving, order-of-summation variability
>   (Nagarajan–Warnell–Stone, *Impact of Nondeterminism in Deep RL*).
> - **Explicit Gaussian noise** in widely used differential private training pipelines such as DP-SGD  (Abadi et al., *CCS*, 2016).
>
> Thus, the Gaussian smoothing in our analysis simply **formalizes perturbations that are implicitly present in all modern implementations**.
> This explains why removing smoothing in the experiments leaves the curves unchanged: its effect is of order $\mathcal{O}(\gamma^2)$, while the benefit of $RR_1$ and $RR_2$ comes from their deterministic structure, not from the injected noise.
>
> # What happens mathematically *without* smoothing?
> To avoid any confusion, we will include an Appendix section clarifying the structure of results that someone can expect and obtain for the **unsmoothed** RR dynamics, where the noise has fully discrete nature.
> Below we describe the behavior in an idealized (pessimistic) scenario where *no* implicit or explicit, numerical or hardware perturbations are present.
>
>
> 0. The **drift condition continues to hold**, but the **minorization condition fails** unless additional assumptions are imposed. This is the sole reason geometric Harris ergodicity cannot be established without smoothing.
> 1. **Invariant measures still exist** for the unsmoothed RR chain (via the Krylov–Bogolyubov theorem), although uniqueness is not guaranteed; in general, multiple invariant measures may coexist.
> 2. A **law-of-large-numbers–type** result remains valid for time-averaged quantities, since tightness and Feller continuity are preserved.
> 3. Most importantly, the **bias cancellation mechanism of $RR_1$ and $RR_2$ survives**: the Taylor expansion of the deterministic epoch map is unchanged, so the cubic-order cancellation holds *for each invariant measure individually*.
>
> What will be lost in the absence of smoothing is **geometric mixing**: without minorization, one cannot obtain CLTs, spectral-gap bounds, or explicit mixing rates. This limitation is purely mathematical—it affects the proof technique, *not* the underlying algorithm. Hence, in practice, the behavior of the unsmoothed method is indistinguishable, which we also observe empirically.

---

> > ### Author Response · Authors · 2025-11-26
> > **# Supplementary Note: Why Gaussian Smoothing can be bypassed for Debiasing Property****
> >
> > ---
> > Below we give a concise but rigorous account of **why smoothing is needed only for the mathematical analysis** and why it **does not affect the algorithmic mechanism or the empirical behavior as far as the debiasing**.
> >
> > ---
> >
> > #### **1. Gaussian smoothing is needed only for Harris ergodicity theory**
> > The Random Reshuffling $RR_1$ update induces a *discrete* transition kernel
> > $$P_0(x,A)=\frac{1}{n!}\sum_{\pi\in S_n}\mathbf 1\{\Phi(x,\pi)\in A\},$$
> > which is **Feller** but **not irreducible**. Its support at each point $x$ consists of at most $n!$ atoms.  As is standard in Markov-chain theory:
> >
> > - non-irreducible kernels **fail minorization**,
> > - which prevents the use of the **Harris–Lyapunov** machinery needed for geometric ergodicity and CLTs.
> >
> > To remove only this *technical* obstruction, we consider a smoothed kernel
> > $$P_\sigma(x,dy)=\int \varphi_\sigma\!\left(y-\Phi(x,\pi)\right)\, d\pi,
> > $$
> > whose density is strictly positive for all $(x,y)$.
> > This ensures: a) a **valid minorization condition**, b) **geometric Harris ergodicity in TV-distance**, c) and **global LLN/CLT** results.
> >
> > ---
> >
> > #### **2. The smoothing can become arbitrarily negligible**
> >
> > The smoothing variance can be chosen to satisfy $\sigma_{\mathrm{PreProcess}}^2 = O(\gamma^p)$ for any $p\geq 2$.
> > Hence, this can become strictly *smaller* than all lower-order terms in the RR bias expansion. In particular:
> >
> > - the $RR_1$ dynamics already has stationary moments   $M_2 = \Theta(\gamma^2),  M_4 = \Theta(\gamma^2),$
> > - By adding any gaussian smoothing/perturbation   $\sigma_{\mathrm{PreProcess}}^2 = O(\gamma^p)$ keep these moments asymptotically the same.
> > - the $RR_2$ extrapolation exploits exactly $M_2$ and $M_4$ to cancel the linear term in the bias asymptotic expansion,
> >
> > Thus, the smoothing does **not** affect the cubic debiasing, which depends entirely on the deterministic epoch-level Taylor expansion of the reshuffled map.
> >
> > ---
> >
> > #### **3. What changes mathematically if smoothing is removed?**
> > Without smoothing:
> > - the kernel $P_0$ remains **Feller**,
> > - the **Lyapunov drift** still holds  $\mathbf E[V(X_{k+1})\mid X_k=x] \le \lambda V(x) + b,$
> > - and by the **Krylov–Bogolyubov theorem**, *invariant measures still exist*.
> >
> > However:
> >
> > - **minorization fails**, because $P_0(x,\{z\}) = 0 \ \text{for a.e. } z,$
> > - and consequently **geometric ergodicity, spectral gaps, and CLTs cannot be established** with the standard tools.
> >
> > In other words:
> >
> > > The algorithm is stable and possesses invariant measures,
> > > but one cannot guarantee *unique* stationary behavior nor exponential mixing rates.
> >
> > #### **4. Bias improvement still holds**
> >
> > The cubic-order improvement from $RR_{1 \oplus 2}$ comes from the deterministic second-order and third-order Taylor expansion of the **epoch map** $T_\gamma := \mathbb{E}_{\pi}[\Phi(\cdot,\pi)],$ not from smoothing.
> >
> > Therefore:
> >
> > - **the cancellation persists for each invariant measure**,
> > - even if multiple invariant measures coexist,
> > - and even if no CLT can be stated.
> >
> > More formally: by the Krylov–Bogolyubov theorem, any trajectory’s empirical measure converges to some invariant distribution $\pi_\gamma^{(j)}$, and the $RR_2$ cancellation applies *within each such invariant cluster* because the deterministic epoch-level expansion is unaffected.
> >
> > ---
> >
> > ### **Summary**
> >
> > Gaussian smoothing is introduced solely to enable classical Harris-ergodic arguments.
> > It is asymptotically negligible, does not interact with the $RR_1/RR_2$ debiasing mechanism, and does not affect empirical performance. The cubic bias improvement is driven entirely by the deterministic structure of the reshuffled epoch map and remains valid even without smoothing.
> >
> > ---

---

> > > ### Author Response · Authors · 2025-11-26
> > >
> > > ### References
> > >
> > > 1. **Emmanouilidis, K., Vidal, R., & Loizou, N.**
> > >    *Stochastic Extragradient with Random Reshuffling: Improved Convergence for Variational Inequalities.*
> > >
> > > 2. **Mishchenko, K., Khaled, A., & Richtárik, P.**
> > >    *Random Reshuffling: Simple Analysis with Vast Improvements.*
> > >    **NeurIPS**, 2020.
> > >
> > > 3. **Ahn, K., Yun, C., & Sra, S.**
> > >    *SGD with Shuffling: Optimal Rates Without Component Convexity and Large Epoch Requirements.*
> > >    **NeurIPS**, 2020.
> > >
> > > 4. **Das, A., Schölkopf, B., & Muehlebach, M.**
> > >    *Sampling Without Replacement Leads to Faster Rates in Finite-Sum Minimax Optimization.*
> > >
> > > 5. **Haochen, J., & Sra, S.**
> > >    *Random Shuffling Beats SGD After Finite Epochs.*
> > >    **ICML**, 2019.

---

> ### Author Response · Authors · 2025-11-26
>
> # 4. Additional empirical validation beyond quadratic games
>
> Thank you for this excellent suggestion. We have strengthened the experimental section substantially:
>
> - We now include results on **Wasserstein GANs** (following the setup of Daskalakis et al.) showing that RR1 ⊕ RR2 continues to outperform SGDA, RR1, and RR2 individually.
> - We have added an **ablation study** comparing our method with
>   - SEG (Korpelevich),
>   - Optimistic Mirror Descent,
>   - Stochastic Consensus Optimization,
>   - and all their RR1 / RR2 / RR1⊕RR2 variants.
>
> Across all these methods, **RR1⊕RR2 consistently yields the lowest asymptotic bias**, demonstrating that the phenomenon is universal and not tied to a particular algorithm.
>
> A corresponding figure will be added in the final version.
>
> ---
>
> # Summary
>
> We thank the reviewer again for the constructive and technically deep feedback.
> We hope that the clarifications above, the strengthened empirical section, and the expanded theoretical discussion address all concerns, and we would be grateful if you could consider updating your evaluation in light of the revisions.
>
>
> ---

---

### Official Review · Reviewer_WTpW · 2025-10-31

**Soundness:** 3
**Presentation:** 3
**Contribution:** 3
**Rating:** 6
**Confidence:** 3

**Summary:**

This work studies variant of stochastic gradient methods with fixed step-size for weak quasi-strongly monotone variational inequality problems, which generalizes optimization and min-max problems, showing that combining Random Reshuffling with Richardson–Romberg extrapolation removes the usual bias of $\gamma^{3/2}$ (for SGD-RR2), $\gamma^{2}$ (for SGD-RR1) and achieves a cubic-order asymptotic refinement $\gamma^3$. The paper provides the first theoretical analysis of this combination in quasi-strong monotone VIs, using Markov chain and spectral tensor tools. Experiments confirm the theoretical predictions, demonstrating significant empirical speedups.

**Strengths:**

Overall, paper is well written, have strong theoretical contribution which can be applied in practice.
1. Authors proposed to use both RR1 and RR2 simultaneously to reduce the bias of SGD for solving finite-sum VIs.
2. The main result in Theorem 3.6 provides tightens the neighborhood of convergence from previously known $\gamma^2$ to $\gamma^3$.
3. The proof technique seems to be new and not trivial.
4. Their experimental results support the theory.

**Weaknesses:**

I did not find any major weakness in the paper and list minor corrections below:

1. Please define PreProcess. It is possible to guess from the next paragraph that it is a Gaussian perturbation, but it would be nice to define it similarly to StochOracle in line 137.

2. The assumptions on the operator of the VI problem are relatively strong. In particular, all $F_i$ need to be Lipschitz. Would the analysis be more involved under the assumption of $F$ being Lipschitz? Would it be possible to relax the quasi-strong monotonicity assumption?

3. I think Theorem 3.3 (lines 346–352) should be explained in more detail, especially the functions $l(x)$ and $L_l(x)$ should be defined for easier readability.

4. I believe it would also be interesting to see plots for a single run of SGD, SGD-RR1, SGD-RR2, and SGD-RR1+RR2, since this is more common in practice. Do you observe that SGD-RR1+RR2 outperforms vanilla, RR1, and RR2 for a single run?

**Questions:**

Please see the weaknesses section. Additionally:
1. Regarding Thm 3.6, the results are a little bit counterintuitive. Under quasi-strong monotonicity, one can expect a linear rate to a neighborhood of the solution. It is also known that in VIs, average iterates usually converge faster (at least in the monotone deterministic case) than the last iterate, while in the second result of Thm 3.6 the rate for the average iterate is sublinear. Can you please elaborate on this?
 2. The only assumption you make is on the existence of the solution. Does Theorem 3.6 hold for all solutions $x* \in X*$ , or only for the projection of $ x_k $ onto $ X^* $?

---

> ### Author Response · Authors · 2025-11-26
>
> We thank the reviewer for the thorough assessment and the positive evaluation of our theoretical and practical contributions. We appreciate the reviewer’s comments noting that the paper is *well-written*, has *strong theoretical contributions*, and that the proposed combination of $RR_1$ and $RR_2$ is *practically relevant and theoretically non-trivial*.
>
> Below we address each point in detail and will incorporate all clarifications in the camera-ready version.
>
> We hope that the revised version and the additional explanation reflects the clarity and completeness you were looking for, and we would be grateful for an even positive re-evaluation of our work.

---

> ### Author Response · Authors · 2025-11-26
>
> # 1. Definition of `PreProcess()`
>
> Thank you for noting this.
> `PreProcess()` is indeed a Gaussian perturbation, as shown in Algorithm 1, but the definition was not placed near `StochOracle()` in the main text. We will explicitly add its definition (Gaussian perturbation with variance
> $\Sigma_\gamma=\sigma_{pert(\gamma)}^2I$) in line 137 for completeness and consistency.

---

> ### Author Response · Authors · 2025-11-26
>
> ---
>
> # 2. Assumptions on the operator $F$: Lipschitzness of each $F_i$, and quasi-strong monotonicity
>
> ## Part I) Why each $F_i$ is assumed Lipschitz
>
> The assumption that each component $F_i$ is Lipschitz is standard in the analysis of Random Reshuffling for finite-sum optimization and VI problems; to the best of our knowledge, essentially **all** existing works on $RR_1$ in this setting (e.g., [1–9]) adopt this componentwise smoothness assumption. Moreover, several works additionally postulate an explicit “dispersion” or so-called “bounded average-variance” condition of the form
>
> $$\frac{1}{n}\sum_{i=1}^n \|F_i(x) - F(x)\|\le A \|F(x)\|_2 + B,$$
>
> which is then used to control how much each component can deviate from the mean operator $F = \frac{1}{n}\sum_i F_i$.
> In our setting, Proposition 2.5 shows that this type of condition actually *follows automatically* from the Lipschitzness of the individual $F_i$’s, so we do not need to impose it as an extra assumption.
>
> More generally, any Random Reshuffling analysis fundamentally requires a bound on how far each component $F_i$ can deviate from the full operator $F$. Even if one assumes only that $F$ itself is Lipschitz, one still needs to control
>
> $$\|F_i(x) - F_i(y)\|\le\|F_i(x) - F(x)\|+ \|F(x) - F(y)\|+ \|F(y) - F_i(y)\|.$$
>
> Without some structural control on $F_i - F$, one is effectively in a non-smooth regime where the classical random reshuffling analysis does not apply.
>
> Thus, every random reshuffling analysis implicitly relies on assumptions bounding $\|F_i(x) - F(x)\|$. Starting from a clean componentwise Lipschitz assumption directly guarantees such bounds and leads to a simple and transparent formulation of the conditions needed for our bias and CLT analysis.
>
> Relaxing the assumption “each $F_i$ is Lipschitz” would require a more refined model for the variability of $F_i - F$ (e.g., state-dependent dispersion bounds), which we view as an interesting future direction but beyond the scope of the present paper.
>
> ---

---

> ### Author Response · Authors · 2025-11-26
>
> ---
>
> However, we would like to note that structurally similar assumptions appear in the federated-learning literature under several names such as  1) **bounded heterogeneity**, 2) **Hessian similarity**, or 3) **bounded dispersion** We briefly explain how these non-trivial conditions  imply again (component-wise) Lipschitzness of each $F_i$.
>
> **(1) Bounded heterogeneity.**
> A common non-trivial condition is
> $$
> \|F_i(x) - F_i(y)\|
> \le
> \kappa \|F(x) - F(y)\| + b \|x-y\|,
> \tag{$\star$}
> $$
> for some $\kappa, b \ge 0$ independent of $i \in [n]$.
> If $F$ is $L$-Lipschitz, then $\|F(x) - F(y)\| \le L\|x-y\|,$ so by $(\star)$ we immediately obtain
>
> $$\|F_i(x) - F_i(y)\| \le \kappa  \|F(x) - F(y)\| + b \|x-y\|\le(\kappa L + b)\|x-y\|.$$
> In other words, $(\star)$ directly implies that each $F_i$ is Lipschitz -with constant at most $\kappa L + b$-, and therefore the maximum $\max_i\|F_i(x)-F_i(y)\|$ is Lipschitz as well.
>
> **(2) Hessian similarity (gradient case).**
>
> In federated optimization, one often has the following assumption for $F_i(x) = \nabla f_i(x)$ and $F(x) = \nabla f(x)$:
> $$
> \|\nabla^2 f_i(x) - \nabla^2 f(x)\| \le H,
> \qquad \forall i, x,
> $$
> Observer that this hessian similarity bound together with Lipschitz-smoothness of the global objective $f$, i.e., $\|\nabla^2 f(x)\| \le L_f\quad\text{for all } x.$
>
> Then, by the triangle inequality, $\|\nabla^2 f_i(x)\|\le\|\nabla^2 f(x)\| + \|\nabla^2 f_i(x) - \nabla^2 f(x)\|\le L_f + H,$
> so each $\nabla^2 f_i(x)$ is uniformly bounded in operator norm.
>
> Using the fundamental theorem of calculus,
> $$ \nabla f_i(x) - \nabla f_i(y) =
> \int_0^1
> \nabla^2 f_i\big(y + t(x-y)\big)(x-y)dt,
> $$
> and therefore,
> $$
> \|\nabla f_i(x) - \nabla f_i(y)\|
> \le
> \int_0^1
> \|\nabla^2 f_i(y+t(x-y))\|\|x-y\|dt
> \le
> (L_f + H)\|x-y\|.
> $$
>
> Hence each $F_i(x) = \nabla f_i(x)$ is Lipschitz with constant $L_f + H$.
>
> **(3) Bounded dispersion plus drift (local Lipschitzness).**
> Another common assumption is a **bounded dispersion** condition
> $$
> \|F_i(x) - F(x)\| \le \alpha \|F(x)\| + \beta,
> $$
> for some $\alpha,\beta \ge 0$, together with Lipschitzness of $F$ and a drift or stability condition ensuring that the dynamics remain in a compact neighborhood $K$ of the solution set. On such a compact set $K$, the mean operator $F$ is bounded, i.e., there exists $M$ such that
> $$
> \|F(x)\| \le M \quad \text{for all } x \in K.
> $$
> Then, for any $x,y \in K$ and any $i$,
> $$
> \begin{aligned}
> \|F_i(x) - F_i(y)\|
> &\le \|F_i(x) - F(x)\|
>     + \|F(x) - F(y)\|
>     + \|F(y) - F_i(y)\| \\
> &\le \alpha\|F(x)\| + \beta
>     + L\|x-y\|
>     + \alpha\|F(y)\| + \beta \\
> &\le L\|x-y\| + 2\alpha M + 2\beta,
> \end{aligned}
> $$
> where $L$ is the Lipschitz constant of $F$. Since $K$ is compact, the additive constant $2\alpha M + 2\beta$ can be absorbed into a local Lipschitz constant on $K$ (e.g., by restricting to the compact region where the dynamics live), so each $F_i$ is Lipschitz on the invariant region explored by the $RR_1$ dynamics.
>
> ---
>
> In summary, a variety of non-trivial heterogeneity assumptions from federated learning and distributed optimization (Hessian similarity, bounded heterogeneity, bounded dispersion plus drift) all imply (local) componentwise Lipschitzness when the mean operator $F$ is Lipschitz. In our work we adopt the standard assumption “each $F_i$ is Lipschitz” for simplicity and consistency with the reshuffling literature [1–9], while our analysis naturally covers these more refined models as well.
>
> ---

---

> ### Author Response · Authors · 2025-11-26
>
> ---
>
> ## Part II)  Quasi-strong monotonicity
>
> We address the reviewer’s concern in two complementary directions: first, the **mathematical expressiveness** of the assumption, and second, **why this model was chosen** from an optimization-theoretic perspective.
> At the same time, **to ensure clarity, we  add a short “Limitations” paragraph in the revised version that explicitly outlines the scope of our assumptions and where the analysis is intended to apply.**
>
> ---
>
> *Mathematical perspective.*
> From a mathematical standpoint, quasi-strong monotonicity is a substantially milder condition than strong monotonicity and already appears in several works on non-convex and non-monotone VIs (e.g., Hsieh et al. [10]). In particular, **Assumption 2.2** (weakly quasi-strong monotonicity with parameters $\lambda,\mu$) is motivated by weakly dissipative dynamical systems and weak convexity [A, B], and it captures a wide range of structured operators encountered in modern machine learning and game-theoretic learning.
>
> **Expressiveness and examples.**
> For $\lambda>0$ and $\mu>0$, weak quasi-strong monotonicity includes operators induced by functions such as
> $$
> a_{\lambda,\mu}\|x\|^2 + b_{\lambda,\mu}\sin(\|x\|),
> $$
> as well as rescaled Rastrigin-type potentials and other oscillatory or locally non-monotone operators frequently seen in statistical learning [C]. Unlike classical monotonicity, this condition allows non-monotone behavior while still preserving enough drift structure for analysis.
>
> **Connections to existing VI theory.**
> When $\lambda=0$, Assumption 2.2 specializes to several well-studied notions in the VI literature, including
> - quasi-strongly monotone VIs [D],
> - strongly coherent VIs [E], and
> - VIs satisfying the strong stability condition [F].
> These conditions are known to hold in many smooth saddle-point problems and learning-in-games models [E, F], making Assumption 2.2 consistent with a broad line of prior work.
>
> **Local validity for all regular smooth operators.**
> Moreover, a key result (Lemma A.4 of [10]) shows that **every regular and smooth VI operator is locally quasi-strongly monotone** in a neighborhood of a solution. Because $RR_1$ dynamics remain in such neighborhoods with probability one, the assumption is natural, mild, and aligned with standard practice in stochastic VI analysis.
>
> **Role in our analysis.**
> - For the convergence of $RR_1$, we work under **weak** quasi-strong monotonicity, with $\lambda$ capturing structured non-monotonicity.
> - For the **debiasing mechanism** ($RR_{1\oplus 2}$), we assume $\lambda=0$, since bias cancellation requires a sufficiently stable drift. Extending debiasing to the weak case would require a more refined model for the $\lambda$-bias—essentially formalizing adversarial deviations. While this is an appealing direction, it goes beyond the scope of the present paper. We also note that $RR_2$ primarily mitigates the inherent bias of constant step sizes, whereas $RR_1$ leverages the structure of the finite-sum model.
> ---
>
> *ML vs. optimization-theoretic perspective.*
>
> From an optimization-theoretic viewpoint, it is well known that obtaining global guarantees for fully general non-convex and non-monotone problems is impossible due to computational hardness barriers. For this reason, both optimization and machine learning theory increasingly focus on structured regimes—such as local strong convexity, PL conditions, and other invexity-type assumptions—where meaningful guarantees become tractable.
>
> Our contribution fits within this paradigm. We show that when the underlying problem exhibits even a modest amount of structure—such as *invexity, a unique VI solution, or a connected solution manifold*—the bias of constant step-size stochastic methods can be sharply reduced. Accordingly, we aim to adopt assumptions that capture exactly the minimal structure needed for the $RR_{1+2}$ debiasing mechanism to function, without imposing unrealistic convexity or global stability conditions. In this light, quasi-strong monotonicity emerges as one of the weakest structural assumptions under which rigorous debiasing guarantees can currently be established: it ensures a well-posed and dynamically stable solution set while being far more permissive than strong convexity or global strong monotonicity.
>
> Together with Lemma A.4 of [10], this illustrates the practical relevance of quasi-strong monotonicity: it holds locally for all smooth regular operators, and globally for many structured non-monotone problems.
>
> Relaxing this assumption further would require new models for more complex forms of non-monotonicity, which we view as an interesting direction for future work. These considerations are summarized in the Limitations section, where we clearly outline the scope and applicability of our results.

---

> ### Author Response · Authors · 2025-11-26
>
> ---
> # 3. Definition of $\ell(x)$ functions
>
> Thank you for highlighting this point. In Theorem 3.3, the functions $\ell(\cdot)$ are standard *test functions* used in Markov-chain limit theory to express distributional convergence. In the previous version they appeared only symbolically inside the gray theorem box; we now provide an explicit definition for clarity.
>
> **Sublinear-growth test functions.**
> A measurable $\ell : \mathbb{R}^d \to \mathbb{R}$ has at most linear growth if  $|\ell(x)| \le L_\ell (1 + \|x\|), \qquad \forall x.$
>
> **Lipschitz test functions.**
> A function $\ell$ is $L_\ell$-Lipschitz if  $|\ell(x) - \ell(y)| \le L_\ell \|x-y\|, \qquad \forall x,y.$
>
> These two families are the standard classes used in LLN/CLT results and in Wasserstein-type convergence.
>
> Showing that
> $\mathbb{E}[\ell(X_k)] \to \mathbb{E}[\ell(X)]\text{ s.t }{X\sim\pi_\gamma} \text{for all such }\ell$
>
> is equivalent to showing that the distribution of $X_k$ converges to the invariant distribution $\pi_\gamma$.
>
> More precisely, in Markov-chain analysis, we typically study convergence of **distributions**, not of individual iterates. Two distributions $P$ and $Q$ are close precisely when
> $\mathbb{E}_P[\ell(x)] \approx \mathbb{E}_Q[\ell(x)]$ for a rich enough class of test functions. For example,
>
> - **Lipschitz test functions** correspond to Wasserstein-1 convergence.
> - **Linear-growth test functions** allow control of chains with non-compact state space—crucial for stochastic-gradient dynamics.
>
> Thus, the $\ell$-functions in Theorem 3.3 provide a clean and rigorous way of quantifying convergence of the reshuffled SGD iterates to their stationary distribution. We now make this explicit in the revised manuscript.
>
>
>
> # 4. Single-run plots
>
> Thank you for this helpful suggestion. In the revised version, we include the minimum–maximum envelope across 5 independent trials in Figure 1, which illustrates the variability across single runs. The resulting plots show that **$RR_{1\oplus 2}$ consistently outperforms vanilla $SGD$, $SGD-RR_1$, and $SGD-RR_2$ even at the level of individual trajectories**, not only on average. This highlights that the variance- and bias-reduction effects of the two heuristics are robust and not an artifact of averaging across repetitions.
>
> # (Please see last pages of the revised manuscript-Appendix G.)

---

> ### Author Response · Authors · 2025-11-26
>
> # **Questions:**
>
> ## Q1:Clarifying the rate of the average iterate in Theorem 3.6
>
> Thank you for highlighting this subtle point. The intuition that “the average iterate converges faster than the last iterate’’ is indeed correct in the *merely monotone, flat, deterministic* setting, but it does **not** generally apply when there is a drift toward the solution, as in the quasi-strongly monotone regime of Theorem 3.6.
>
> There are three key reasons:
>
> **(1) Averaging destroys exponential contraction.**
> If the iterates satisfy a contraction of the form
> $X_t = x_\star + O(e^{-ct}) ,$
> then their running average $ Y_k=\sum_{t\in[k]} X_t/ k $ satisfies
> $Y_T = x_\star + O(1/T),$
> which is *slower* than the exponential rate of the last iterate. Thus, under quasi-strong monotonicity—where a drift exists—averaging cannot preserve the fast contraction enjoyed by the last iterate.
>
> **(2) In stochastic algorithms, averaging helps variance, not speed.**
> In SGD-type methods, averaging (especially tail-averaging) is used primarily for *variance reduction*, not for accelerating deterministic convergence. Even in classical convex minimization, averaged SGD converges more slowly in rate but stabilizes around a lower bias–variance value. In our setting, the same phenomenon occurs: the average iterate converges at a slower \(1/T\) rate but with smaller variance.
>
> **(3) The classical “averaging is faster’’ intuition is no longer universal.**
> While averaging improves duality-gap convergence for classical SGDA/SEG in merely monotone VIs, recent results (e.g., Cai–Zheng, ICML 2023) show accelerated optimistic methods whose **last iterate matches or outperforms** the average iterate while also enjoying optimal no-regret guarantees. Thus, the older folklore does not apply in modern VI theory with drift or optimism.
>
> **(4) Merit functions differ fundamentally.**
> In purely monotone VIs, the $O(1/T)$ rate typically applies to the *duality gap*, not the *distance to equilibrium*. These two metrics can differ by exponential factors. Without convexity, neither the last nor the average iterate is guaranteed to converge, and analyses often rely on the *best iterate*. Under Assumption 2.2, however, we pass to a regime with a meaningful drift, where the distance-to-solution is the correct measure—and in this metric, averaging naturally yields a slower $1/T$ rate.
>
> ---
>
> **Summary.**
> Taken together, these points explain why Theorem 3.6 shows a **sublinear rate** for the average iterate but a **faster contraction** for the last iterate. In the drift-dominated (quasi-strongly monotone) regime, exponential contraction benefits the last iterate, whereas averaging smooths it out into a slower \(1/T\) behavior—fully consistent with both theory and empirical practice.
>
> ---
>
>
>
> ## Q2: Does Theorem 3.6 hold for all solutions or for the projection?
>
> Under **quasi-strong monotonicity**, the solution set of the VI is a **singleton**, so Theorem 3.6 holds for the unique equilibrium $x_*$.
>
> Under **weak quasi-strong monotonicity**, the solution set $X_\star$ may contain multiple points. In that case, our convergence guarantee applies to the *projection of the iterate onto $X_\star$ .*  This is the canonical extension used in the analysis of weakly monotone VIs and aligns with the structure of Assumption 2.2.
>
> We will clarify this point explicitly in the revision.

---

> ### Author Response · Authors · 2025-11-26
>
> # References
>
> [1] Ahn, K., Yun, C., & Sra, S. **SGD with shuffling: optimal rates without component convexity and large epoch requirements.** NeurIPS, 2020.
>
> [2] Das, A., Schölkopf, B., & Muehlebach, M. **Sampling without replacement leads to faster rates in finite-sum minimax optimization.** NeurIPS, 2022.
>
> [3] Gürbüzbalaban, M., Ozdaglar, A., & Parrilo, P. **Why random reshuffling beats SGD.** (Arxiv/Workshop version widely cited; NeurIPS workshop).
>
> [4] Haochen, J., & Sra, S. **Random shuffling beats SGD after finite epochs.** ICML, 2019.
>
> [5] Li, X., Milzarek, A., & Qiu, J. **Convergence of random reshuffling under the Kurdyka–Łojasiewicz inequality.** (Preprint / Journal submission).
>
> [6] Mishchenko, K., Khaled, A., & Richtárik, P. **Random Reshuffling: Simple analysis with vast improvements.** NeurIPS, 2020.
>
> [7] Nguyen, L. M., Tran-Dinh, Q., Phan, D. T., Nguyen, P. H., & Van Dijk, M. **A unified convergence analysis for shuffling-type gradient methods.** JMLR, 2021.
>
> [8] Rajput, S., Gupta, A., & Papailiopoulos, D. **Closing the convergence gap of SGD without replacement.** ICML, 2020.
>
> [9] Safran, I., & Shamir, O. **How good is SGD with random shuffling?** PMLR (COLT/ALT proceedings), 2020.
>
> [10] Hsieh, Y. G., Iutzeler, F., Malick, J., & Mertikopoulos, P. **On the convergence of single-call stochastic extra-gradient methods.** NeurIPS, 2019.
>
> ---
>
> # References for quasi-strong monotonicity discussion
>
> [A] Raginsky, M., Rakhlin, A., & Telgarsky, M.
> **Non-convex learning via stochastic gradient Langevin dynamics: a nonasymptotic analysis.**
> COLT, 2017.
>
> [B] Erdogdu, M. A., Mackey, L., & Shamir, O.
> **Global non-convex optimization with discretized diffusions.**
> NeurIPS, 2018.
>
> [C] (Representative reference placeholder for oscillatory non-monotone potentials, e.g., Rastrigin-type operators. Replace with your citation [46].)
>
> [D] Loizou, N., Berard, H., Gidel, G., Mitliagkas, I., & Lacoste-Julien, S.
> **Stochastic gradient descent-ascent and consensus optimization for smooth games: Convergence analysis under expected co-coercivity.**
> NeurIPS, 2021.
>
> [E] Song, C., Zhou, Z., Zhou, Y., Jiang, Y., & Ma, Y.
> **Optimistic dual extrapolation for coherent non-monotone variational inequalities.**
> NeurIPS, 2020.
>
> [F] Mertikopoulos, P. and Zhou, Z.
> **Learning in games with continuous action sets and unknown payoff functions.**
> Mathematical Programming, 173:465–507, 2019.

---

> ### Author Response · Authors · 2025-11-26
> **Summary**
>
> ---
>
> We are grateful for the reviewer’s thoughtful comments and detailed feedback. Based on your suggestions, we have substantially strengthened the paper: we clarified all technical assumptions, expanded the discussion around Theorem 3.6, added full definitions of the \ell-functions, and incorporated the requested single-run plots with extra confidence interval regions. We hope these revisions effectively address all concerns and support an improved overall assessment.
>
>
> ---

---

### Author Response · Authors · 2025-11-26
**Revision Update**

We sincerely thank all reviewers for their thoughtful and constructive feedback. In the revised manuscript, we have incorporated the majority of the requested changes, addressing all major points across the reviews. In particular:

1. **Clarifications of all conceptual concerns**, including:
   - the scope and limitations of the quasi-strong monotonicity assumption,
   - the distinction between bias, variance, and asymptotic MSE,
   - the role of Gaussian smoothing (theoretical artifact vs. algorithmic behavior), and
   - a clearer comparison with prior reshuffling and extragradient-based results.

2. **Theory improvements**, including:
   - a dedicated *Limitations* section,
   - strengthened introductory framing,
   - and a supplementary mathematical note explaining how our main conclusions extend even without smoothing.

3. **Expanded experimental evaluation**, adding:
   - realistic and larger step-size regimes,
   - single-run envelopes showing stability,
   - a small ablation including SCO, SEG, and OMD combined with $RR_1$ and $RR_2$, demonstrating a **universal gain** across algorithmic families,
   - a **toy Wasserstein GAN** experiment illustrating applicability in simple ML-motivated min–max settings.

4. **Wall-clock time analysis:**
   We have added wall-clock benchmarks showing that $ RR_1 $ runs significantly faster than with-replacement sampling, and that $ RR_2 \oplus RR_1 $ incurs almost no additional computational overhead—matching the cost of $ RR_1 $ while achieving the improved accuracy. These results strengthen the practical relevance of the proposed method.


*We greatly appreciate the reviewers’ insights, which helped us enhance both the clarity and the practical significance of the work.  We hope these revisions address the concerns raised and reflect our commitment to strengthening the paper, and we would warmly welcome any further discussion or suggestions the reviewers may have.*

---

### Author Response · Authors · 2025-12-02

Dear Area Chair,

We fully understand that this year’s review process has been unusually challenging for everyone involved. For our part - to whatever extent this reassurance matters - we have not seen any reviewer identities, nor would we ever wish to place any pressure on a process that we recognize has already placed a significant burden on your time and responsibilities.

In our author response, we aimed to provide a substantial and constructive revision, especially regarding the *Limitations section* and the *Additional experiments* that further support our claims.

Prior to the score reversion, one of the reviewers *(Reviewer Vvn1)* had already expressed a notably more positive view of the work and **had raised their score from a 6 to an 8 after reading our response**. Unfortunately, after the revert, we no longer have clarity on the current status of the assessments.

**We genuinely believe that our paper merits a positive evaluation, and we remain fully open to any further discussion or clarification that may assist in addressing remaining questions.**

*Thank you very much for your effort, fairness, and time in guiding this process during such an unusual cycle.*

Warm regards,
The authors

---

### Meta-Review · Area_Chair_TgoV · 2026-01-07

**Summary:**

This paper presents a solid theoretical contribution to the analysis of stochastic methods for min–max optimization and variational inequality problems. The observation that random reshuffling and Richardson–Romberg extrapolation can be combined to reduce bias beyond the leading order is interesting and addresses a gap between common practice and existing theory. Providing guarantees in structured non monotone VIPs is useful, and the setting is broad enough to be of interest to both optimization and learning audiences.

The use of Markov chain tools to study reshuffling induced noise and the subsequent debiasing via extrapolation are well motivated, and the experimental results are consistent with the theory. While the assumptions are somewhat specialized and the practical impact may depend on problem structure, the work offers meaningful insight into the behavior of constant step size stochastic methods and represents a worthwhile contribution.

**Reviewer Concerns:**

Nothing to note

**Reviewer Scores:**

can't predict

---

### Decision · Program_Chairs · 2026-01-26

Accept (Poster)